# Single Index Bandits: Generalized Linear Contextual Bandits with Unknown Reward Functions

**Yue Kang**[†*], **Mingshuo Liu**[‡], **Bongsoo Yi**[§], **Jing Lyu**[‡],
**Zhi Zhang**[¶], **Doudou Zhou**[∥], **Yao Li**[§]

[†]Microsoft    [‡]University of California, Davis    [§]University of North Carolina at Chapel Hill
[¶]University of California, Los Angeles    [∥]National University of Singapore
yuekang@microsoft.com
{mshliu,jjlyu}@ucdavis.edu
{bongsoo,yaoli}@unc.edu
zzh237@g.ucla.edu
ddzhou@nus.edu.sg

## Abstract

Generalized linear bandits have been extensively studied due to their broad applicability in real-world online decision-making problems. However, these methods typically assume that the expected reward function is known to the users, an assumption that is often unrealistic in practice. Misspecification of this link function can lead to the failure of all existing algorithms. In this work, we address this critical limitation by introducing a new problem of generalized linear bandits with unknown reward functions, also known as single index bandits. We first consider the case where the unknown reward function is monotonically increasing, and propose two novel and efficient algorithms, STOR and ESTOR, that achieve decent regrets under standard assumptions. Notably, our ESTOR can obtain the nearly optimal regret bound $\tilde{O}_T(\sqrt{T})$[1] in terms of the time horizon $T$. We then extend our methods to the high-dimensional sparse setting and show that the same regret rate can be attained with the sparsity index. Next, we introduce GSTOR, an algorithm that is agnostic to general reward functions, and establish regret bounds under a Gaussian design assumption. Finally, we validate the efficiency and effectiveness of our algorithms through experiments on both synthetic and real-world datasets.

## 1 Introduction

The multi-armed bandit models sequential decision-making with partial feedback and has found broad applicability in areas such as online recommendation (Li et al., 2010), clinical trials (Villar et al., 2015), precision medicine (Lu et al., 2021), and hyperparameter learning (Ding et al., 2022; Kang et al., 2024b). To incorporate the side information (contexts) of arms, one widely adopted extension is the linear contextual bandit model (Abbasi-Yadkori et al., 2011; 2012), which assumes the expected reward is a linear function of the observed feature with some unknown parameter. However, many real-world applications involve inherently non-linear reward structures, such as binary clicks or discrete counts, rendering linear models inadequate. Generalized linear bandits (GLBs) (Filippi et al., 2010; Li et al., 2017) were then naturally introduced to address this limitation by applying a non-linear (inverse) link function to the linear predictor, enabling compatibility with a broader range of reward types. While GLBs maintain much of the traceability and theoretical guarantees of linear models, they fundamentally rely on the assumption that the link function is known in advance. In practice, however, this assumption is often unrealistic, as the underlying parametric form is typically unknown and impossible to identify. It has been shown that model misspecification can severely

---

[*]Yue Kang is the corresponding author.

[1]$\tilde{O}$ hides polylogarithmic factors. A subscript $T$ on asymptotic notations (e.g., $O_T$) indicates that the bound is expressed only in terms of $T$, with dependence on other problem parameters suppressed.

degrade the practical performance (Ghosh et al., 2017; Bogunovic & Krause, 2021), and even minor deviations can lead to linear regret for all existing GLB methods (Lattimore & Szepesvári, 2020). This fundamental limitation motivates the development of a more robust, agnostic approach.

To address this critical limitation, we introduce the single index bandit (SIB) problem, where the expected reward is an unknown function of the linear predictor. To the best of our knowledge, this is the first work to advance GLBs without assuming access to the reward function. Our setting is inspired by single index models (SIMs) from statistical learning (Härdle, 2004), which extend generalized linear models (GLMs) by removing the need for a known link function. Despite their flexibility and effectiveness in offline settings, SIMs have never been explored in the bandit literature, primarily due to the substantial theoretical challenges they pose in online learning. We propose a family of SIB algorithms that achieve strong theoretical guarantees and demonstrate great empirical performance in this work, all without knowing the form of the reward function. Notably, although GLBs have been extensively studied in recent years, none of the existing results can be applied or extended to our setting: Both Upper Confidence Bound (UCB)-based methods (Filippi et al., 2010; Li et al., 2017) and Thompson Sampling (TS)-based approaches (Agrawal & Goyal, 2013; Kveton et al., 2020) require solving a (quasi-)maximum likelihood estimator for the unknown parameter, which essentially relies on explicit knowledge of the reward function. Another class of GLB algorithms improves computational efficiency via second-order online optimization (Jun et al., 2017; Xue et al., 2024), while these methods also presume a known link function for online Newton updates at each round. More fundamentally, all existing GLB analyses depend on concentration bounds for some vector-valued martingales involving the noise terms across the time horizon $T$, whose construction inherently requires the explicit form of the reward function, and these analytical techniques collapse in the agnostic setting. On the other hand, another line of research considers the general contextual bandit problem under the realizability assumption, i.e., the expected reward belongs to some known function class $\mathcal{F}$ (Foster et al., 2018; Foster & Rakhlin, 2020; Simchi-Levi & Xu, 2022), and these approaches critically assume access to a powerful square-loss regression oracle with strong non-asymptotic guarantees. However, such an oracle is impossible to realize in single index models: most existing SIM estimators are based on maximum likelihood methods and provide only asymptotic guarantees (Han, 1987; Carroll et al., 1997; Härdle, 2004), while the few results with finite-sample rates require highly restrictive distributional assumptions (e.g., i.i.d. Gaussian covariates) (Neykov et al., 2016; Plan & Vershynin, 2016). These conditions are fundamentally incompatible with all contextual bandit algorithms under the realizability assumption, where the full reward function and its parameter must be learned jointly from adaptively collected data rather than from a fixed and restrictive sampling strategy. A detailed explanation of this intrinsic incompatibility is provided in Appendix B. In conclusion, the single index bandit setting we introduce lies beyond the reach of all existing approaches, and tackling this challenging problem necessitates the development of entirely new algorithmic techniques and theoretical tools.

To overcome the aforementioned challenges, we introduce a novel estimator based on Stein's method, which offers a fundamentally different approach to estimating the unknown parameter. While Stein's method has recently been applied in low-rank matrix bandits (Kang et al., 2022), it remains unexplored in the context of linear or generalized linear bandits. We first leverage this tool to study the case where the unknown reward function is continuously differentiable and monotonically increasing, and propose two efficient algorithms that achieve strong performance under mild conditions. Notably, GLMs with the canonical form also induce monotonically increasing reward functions (McCullagh, 2019), which indicates that this setting strictly encompasses the classic GLB framework. Furthermore, we extend our framework to the sparse high-dimensional regime and generalize our approach to handle arbitrary (non-monotonic) reward functions. The contributions are summarized as follows:

- We propose a novel and efficient estimator based on Stein's method that requires only the existence of the second moment of the noise and attains the minimax error rate. Moreover, our estimator is highly efficient: it requires no optimization, and can be computed in $O(nd)$ time and $O(d)$ space from $n$ contextual samples in $\mathbb{R}^d$. We further show that this estimator extends naturally to the sparse high-dimensional regime with the same error bound in terms of the sparsity index. These results also constitute a notable advancement in the estimation of SIMs as well.

- As a warm-up, we introduce STOR, an Explore-then-Commit (EtC) style algorithm, and show that it achieves a regret bound of order $\tilde{O}_T(T^{\frac{2}{3}})$. Building on this, we propose a new algorithm named

ESTOR that leverages a novel epoch-based schedule, and prove that it achieves a nearly-optimal regret bound of order $\tilde{O}_T(T^{\frac{1}{2}})$ under our significantly more challenging problem setting.

- Next, we extend our results to the high-dimensional setting and show that the same regret bounds hold when the ambient dimension is replaced by the sparsity level of the true parameter, making our approach well-suited for modern applications with sparse high-dimensional feature spaces.

- Furthermore, we propose GSTOR, a new method that extends STOR to the general case where the reward functions are not necessarily monotonic. We introduce a double-exploration-then-exploit methodology that combines our Stein's method-based estimator with a kernel regression procedure to approximate the unknown link function. Under a standard Gaussian design assumption, our GSTOR achieves the regret of order $\tilde{O}_T(T^{\frac{3}{4}})$.

- Finally, we validate our theoretical results through extensive experiments on both synthetic and real-world datasets, illustrating the effectiveness of our methods across diverse reward structures.

Due to space constraints, we defer the full related work with discussion to Appendix B. There, we provide a comprehensive overview of prior research on generalized linear bandits, single index models, and contextual bandits under the realizability assumption. We further explain why all existing approaches are fundamentally unable to address the single index bandit setting in detail, and illustrate that the SIM/SIB is significantly more challenging than the GLM/GLB.

## 2 PRELIMINARIES

We present the setting of single index bandit (SIB) problem and introduce the assumptions used in our analysis. Denote $T$ as the time horizon and $\mathcal{X}_t$ as the arm set available to the agent at each round $t \in [T]$, where $\mathcal{X}_t = \{x_{t,a} \in \mathbb{R}^d : a \in [K]\}$ consists of $K$ ($K \geq 3$) feature vectors sampled i.i.d. from some $d$-dimensional continuous distribution $\mathcal{D}$ with the multivariate density $p(\cdot)$. At time $t$, the agent chooses an arm $x_t \in \mathcal{X}_t$ and observes the associated stochastic reward $y_t$ such that $y_t = f(x_t^\top \theta_*) + \eta_t$, where $\theta_* \in \mathbb{R}^d$ is an unknown parameter vector and $\eta_t$ is the zero-mean noise with finite variance $\sigma^2$. Note that our analysis only requires a bounded variance assumption, which is strictly weaker than the sub-Gaussian noise condition imposed in most GLB literature (Lattimore & Szepesvári, 2020). Additionally, unlike the GLB setting where the reward function $f(\cdot)$ is known to the agent, we consider a more challenging problem in which $f(\cdot)$ is an unknown, continuously differentiable function. This agnostic relaxation significantly increases the difficulty of the problem and invalidates the theoretical guarantees of all existing GLB methods, as discussed in Section 1 and Appendix B. Furthermore, we let $\|\theta_*\|_1 = 1$ for model identifiability. Define $x_{t,*} := \arg\max_{x \in \mathcal{X}_t} x^\top \theta_*$ as the feature vector of the optimal arm at round $t$, and the goal is to minimize the cumulative regret $R_T$ across the time horizon $T$ defined as,

$$R_T = \sum_{t=1}^{T} f(x_{t,*}^\top \theta_*) - f(x_t^\top \theta_*).$$

Next, we present the following key definition of the score function used in Stein's method.

**Definition 2.1.** Let $p : \mathbb{R}^d \to \mathbb{R}$ be a probability density function defined on $\mathbb{R}^d$. The score function $S^p : \mathbb{R}^d \to \mathbb{R}^d$ associated with the density (pdf) $p(\cdot)$ is given by:

$$S^p(x) = -\nabla_x \log(p(x)) = -\nabla_x p(x)/p(x), \quad x \in \mathbb{R}^d,$$

and $S_i^p(x)$, $i \in [d]$ is defined as the $i$th entry of the score function vector $S^p(x) \in \mathbb{R}^d$.

We will omit the subscript $x$ in $\nabla$ and the superscript $p$ in $S$ when the underlying distribution is clear from context. With this definition, we then introduce two standard assumptions.

**Assumption 2.2.** There exists some constant $M > 0$ such that $\mathbb{E}(S_j(X)^2) \leq M$ for all $j \in [d]$ where $X$ is a random variable sampled from the distribution $\mathcal{D}$.

**Assumption 2.3.** There exists some value $L > 0$ such that $\|x_{t,i}\|_\infty \leq L$ for all $i \in [d], t \in [T]$. And the unknown reward function $f(\cdot)$ and its derivative $f'(\cdot)$ have constant upper bounds $L_f, L_{f'} > 0$, i.e. $|f(x)| \leq L_f, |f'(x)| \leq L_{f'}$, for any $|x| \leq L$.

---

**Algorithm 1** Stein's Oracle Single Index Bandit (STOR)

---

**Input:** $T$, the probability rate $\delta$, parameters $T_1, \lambda, \tau$
1: **for** $t = 1$ **to** $T_1$ **do**
2:     Pull an arm $x_t \in \mathcal{X}_t$ uniformly randomly and observe the stochastic reward $y_t$.
3: **end for**
4: Obtain the estimator $\hat{\theta}$ with $\{x_i, y_i\}_{i=1}^{T_1}$ based on Eqn. (1).
5: **for** $t = T_1 + 1$ **to** $T$ **do**
6:     Choose the arm $x_t = \arg\max_{x \in \mathcal{X}_t} x^\top \hat{\theta}$, break ties arbitrary.
7: **end for**

---

Assumption 2.2 is mild and applies to a broad class of distributions, including many non-sub-Gaussian or even non-zero-mean cases, which enables us to handle previously intractable cases. Furthermore, it is less restrictive than the conditions in the state-of-the-art literature on Stein's method and robust estimation literature (Fan et al., 2019; 2023), which typically require at least a finite fourth moment of the score function. Assumption 2.3 is also standard and widely used in existing linear contextual bandit algorithms. While some existing methods assume both the arm vector $x$ and the parameter $\theta_*$ have an $l_2$-norm bound of $S$ (Abbasi-Yadkori et al., 2011; 2012; Chu et al., 2011; Ding et al., 2021; Li et al., 2017), which may appear different from our Assumption 2.3, both formulations ensure that the inner product $x^\top \theta_*$ is bounded by $L$, maintaining the same fundamental constraint. Moreover, our main results remain valid under both settings, and we adopt our assumption because it is commonly used in sparse linear bandit literature (Hao et al., 2020; Jang et al., 2022), allowing us to maintain consistency in our discussion of both linear and sparse linear bandits. A more detailed discussion on this matter is provided in Appendix I. Additionally, we allow $L$ to remain at a constant scale up to logarithmic factors, i.e. $L = \tilde{O}(1)$, while ensuring our results remain valid. This flexibility accommodates a broad range of distribution $\mathcal{D}$, including any sub-Gaussian or sub-Exponential distribution, and more technical details are provided in Appendix I.

## 3 METHODS

In this section, we introduce our SIB algorithms, beginning with the case of monotonically increasing reward functions. We then extend our approach to the high-dimensional regime and ultimately generalize to non-monotonic settings. We first utilize a novel Stein's-method-based estimator for the unknown parameter $\theta^*$ under general continuously differentiable reward functions, and show that it achieves the optimal error rate under mild assumptions. We further extend this estimator to the sparse high-dimensional setting by incorporating an $l_1$ regularizer, and demonstrate that the optimal error rate is well preserved in terms of the sparsity level of the true parameter.

### 3.1 SINGLE INDEX MODEL ESTIMATOR

Suppose we collect $n$ pairs of samples $\{x_i, y_i\}_{i=1}^n$ where the feature vector $x_i$ is i.i.d. sampled from the $d$-dimensional distribution $\mathcal{D}$. We propose the following minimization problem in Eqn. (1).

$$\hat{\theta} = \arg\min_{\theta \in \Theta} L(\theta) + \lambda \|\theta\|_1, \text{ where } L(\theta) = \|\theta\|_2^2 - \frac{2}{n} \sum_{i=1}^n \phi_\tau(y_i \cdot S(x_i))^\top \theta, \quad (1)$$

where we denote $\phi_\tau(\cdot) : \mathbb{R}^d \to \mathbb{R}^d$ as the elementwise truncation function such that

$$\phi_\tau(v) = [\text{sign}(v_j) \cdot (|v_j| \wedge \tau)]_{j=1}^d, \quad \forall v = [v_j]_{j=1}^d \in \mathbb{R}^d,$$

for some parameter $\tau > 0$. $\|\theta\|_1$ is the $l_1$ regularizer designed for sparse high-dimensional setting (Plan & Vershynin, 2016). Notably, instead of directly using the term $y_i \cdot S(x_i)$, we employ a careful truncation to control the variance of the estimator while managing bias. The truncation is governed by a parameter $\tau$, which explicitly balances bias induced by discarding extreme values against variance reduction to achieve optimal statistical guarantee. While this thresholding technique has been widely adopted in statistical and online learning to handle heavy-tailed rewards (Fan et al., 2021; Bubeck et al., 2013; Kang et al., 2024a), we demonstrate its novel applicability to mitigate the ambiguity from misspecification of the reward function $f(\cdot)$ in this work. Next, we establish the

following statistical guarantees for the estimator in Eqn. (1). Notably, our $\ell_1$ and $\ell_2$ error bounds match the minimax rates of classical linear regression Wainwright (2019) up to logarithmic terms, and since linear regression is a special case of our setting, these bounds are nearly optimal.

**Theorem 3.1.** (Bound of SIM) *For any single index model defined in Section 2 with samples $x_1, \ldots, x_n$ drawn from some distribution $\mathcal{D}$. We denote $\mu_* := \mathbb{E}(f'(X^\top \theta_*))$, $X \sim \mathcal{D}$ and assume $\mu_* \neq 0$. Under Assumption 2.2, 2.3, by solving the optimization problem in Eqn.* (1) *with $\tau = \sqrt{3(\sigma^2 + L_f^2)Mn/\log(2d/\delta)}$ and $\lambda = 0$, with probability at least $(1 - \delta)$ it holds that:*

$$\left\| \hat{\theta} - \mu_* \theta_* \right\|_2 \leq \left( \frac{2\sqrt{3}}{3} + \sqrt{2} \right) \cdot \sqrt{\frac{dM(\sigma^2 + L_f^2)\log(2d/\delta)}{n}} = \tilde{O}\left( \sqrt{\frac{d}{n}} \right),$$

$$\left\| \hat{\theta} - \mu_* \theta_* \right\|_1 \leq \left( \frac{2\sqrt{3}}{3} + \sqrt{2} \right) d \cdot \sqrt{\frac{M(\sigma^2 + L_f^2)\log(2d/\delta)}{n}} = \tilde{O}\left( \frac{d}{\sqrt{n}} \right).$$

**Proof sketch of Theorem 3.1:** We begin by applying Stein's identity to show that $\mathbb{E}[y_i S(x_i)] = \mu_* \theta_*$, which motivates our estimator. To analyze its accuracy, we upper bound the optimization gap $L(\hat{\theta}) - L(\mu_* \theta_*)$ by bounding the gradient norm $\|\nabla L(\mu_* \theta_*)\|_\infty$. We control the heavy-tailed nature of $y_i$ via a truncation function $\phi_\tau(y)$, which limits the influence of extreme values while introducing a controllable bias. We show that the bias term from truncation is small (via Taylor expansion) and the variance is concentrated using Bernstein's inequality. Finally, by applying first-order convexity arguments, we obtain high-probability bounds on both $\|\hat{\theta} - \theta_*\|_2$ and $\|\hat{\theta} - \theta_*\|_1$ that match the minimax rate, all without requiring knowledge of the reward function $f(\cdot)$.

The full proof of Theorem 3.1 is deferred to Appendix D. Notably, our estimator is highly computationally efficient. In the low-dimensional regime, the loss in Eqn. (1) reduces to a simple quadratic form $L(\theta)$ with $\lambda = 0$, and the solution admits a closed-form expression as $\sum_{i=1}^{n} \phi_\tau(y_i S(x_i))/n$. This solution requires no iterative optimization and is obtained by a single averaging operation over the samples, resulting in only $O(nd)$ time and $O(d)$ space complexity. This is remarkably more efficient than solving the MLE used by most GLB algorithms, despite our setting being significantly harder due to the unknown reward function.

While our method assumes that data is sampled from a fixed distribution $\mathcal{D}$, rather than being adversarially chosen, which may appear strong at first glance, we emphasize that this is a reasonable condition in the study of SIMs and is still milder than the stronger assumptions commonly imposed in prior work (see Appendix B). Note that the SIM/SIB setting is fundamentally more difficult than GLM/GLB, as it involves jointly learning both the parameter and the unknown link function. To make the problem tractable, previous studies on SIMs typically impose additional distributional assumptions, such as assuming the context vector is drawn from a standard Gaussian and the link function satisfies certain regularity conditions (Neykov et al., 2016; Thrampoulidis et al., 2015; Plan & Vershynin, 2016). In contrast, our approach avoids these assumptions and aligns with prior bandit work that also employs Stein's method (Kang et al., 2022; Wang et al., 2025). Conclusively, we view the assumption on Stein's score as a reasonable and necessary compromise for addressing a significantly more realistic and challenging setting than GLBs. A detailed review of existing SIM works is given in Appendix B. And we leave studying SIBs with adversarially chosen arms as a challenging future direction.

## 3.2 WARM-UP: SIMPLE ALGORITHM AGNOSTIC TO INCREASING REWARD FUNCTIONS

We first consider the case where the unknown reward function is continuously differentiable and monotonically increasing. Notably, the reward function of any GLM in the canonical form is also monotonically increasing, implying that our setting still encompasses the existing GLB literature.

As a warm-up, we present our efficient STein's ORacle single index bandit (STOR) algorithm, which leverages the Explore-then-Commit (EtC) methodology. EtC has been widely adopted in contextual bandit literature, particularly in the high-dimensional regime (Li et al., 2022; Jang et al., 2022; Hao et al., 2020) due to its simplicity. Our STOR method is shown in Algorithm 1. Specifically, STOR begins with an exploration phase of length $T_1$, where it randomly selects actions for a fixed number

---

**Algorithm 2** Epoched Stein's Oracle Single Index Bandit (ESTOR)

---

**Input:** $T$, epoch schedule $e_i = (2^i - 1)T_0$, the probability rate $\delta$, parameters $T_0, \{\lambda_i, \tau_i\}_{i=2}$
**Initialization:** Set the density function $p_1 = p$ corresponding to the distribution $\mathcal{D}$.
  1: **for** $t = 1$ **to** $e_1$ **do**
  2:    Pull an arm $x_t \in \mathcal{X}_t$ uniformly randomly and observe the stochastic reward $y_t$.
  3: **end for**
  4: **for** epoch $i = 2, 3, \ldots$ **do**
  5:    Obtain the estimator $\hat{\theta}_i$ with $\{x_j, y_j\}_{j=e_{i-2}+1}^{e_{i-1}}$ based on Eqn. (1), where we set $\lambda = \lambda_i$, $\tau = \tau_i$ and $S(\cdot)$ as the score function associated with the density $p_{i-1}$.
  6:    **for** $t = e_{i-1} + 1$ **to** $\min\{e_i, T\}$ **do**
  7:       Choose the arm $x_t = \arg\max_{x \in \mathcal{X}_t} x^\top \hat{\theta}_i$, break ties arbitrary.
  8:    **end for**
  9:    Update the density function $p_i(x) = K \cdot p(x) \cdot F_i(x^\top \hat{\theta}_i)^{K-1}$, where $F_i(\cdot)$ is the cumulative distribution function of the random variable $X^\top \hat{\theta}_i$ with $X \sim \mathcal{D}$.
 10: **end for**

---

of rounds to estimate the direction of the unknown parameter $\theta_*$. It then enters the exploitation phase and greedily selects actions that are expected to maximize the estimated reward.

**Theorem 3.2.** *For any single index bandit with an increasing function, we set $T_1 = (dT)^{\frac{2}{3}} \ln (2d/\delta)^{\frac{1}{3}}$, $\tau = \sqrt{3T_1/\ln(2d/\delta)}$ and $\lambda = 0$ in Algorithm 1. Then under Assumption 2.2, 2.3, we have the following regret bound with probability at least $1 - \delta$:*

$$R_T = O\left(d^{\frac{2}{3}} T^{\frac{2}{3}} \left(\ln (d/\delta)\right)^{\frac{1}{3}}\right) = \tilde{O}\left(d^{\frac{2}{3}} T^{\frac{2}{3}}\right).$$

Although our Algorithm 1 provides the first solution to SIBs with monotonically increasing functions, its regret bound does not match the optimal rate of order $\tilde{O}_T(\sqrt{T})$ for GLBs (Lattimore & Szepesvári, 2020). This is due to the intrinsic suboptimality of the EtC approach, which incurs additional regret from a fixed exploration-exploitation split. Remarkably, we resolve this challenge in the next subsection by proposing an improved and novel algorithm, ESTOR, which achieves the minimax-optimal regret bound of order $\tilde{O}_T(\sqrt{T})$ up to logarithmic factors.

### 3.3 IMPROVED ALGORITHM AGNOSTIC TO INCREASING REWARD FUNCTIONS

Our Epoched STein's ORacle single index bandit (ESTOR) algorithm is presented in Algorithm 2. It employs a carefully designed epoch schedule with an epoch-specific hyperparameter configuration. Specifically, ESTOR uses exponentially increasing epoch lengths, and at the beginning of each epoch, it leverages data collected from the preceding epoch to compute an updated estimate of the unknown parameter (line 5). Actions are selected greedily according to this most recent estimate within each epoch (line 7). At the end of each epoch $i$, ESTOR updates the distribution $p_i$ for arm selection based on the updated estimator $\hat{\theta}_i$ (line 9), which serves as the basis for our novel parameter estimator in the subsequent epoch. Although epoch scheduling and greedy selection have been studied individually, their integration with a solid theoretical guarantee remains largely unexplored in the literature.

Furthermore, while traditional epoch-based techniques like the doubling trick (Auer et al., 1995) focus on adapting to unknown time horizons, our motivation differs fundamentally: we carefully select epoch lengths and utilize epoch-specific hyperparameters that gradually evolve, effectively balancing exploration and exploitation over time. Shorter initial epochs facilitate rapid exploration and coarse estimation, while progressively longer epochs ensure later stages accumulate sufficiently many samples, allowing our parameter estimates to become increasingly precise with diminishing error. By coordinating these components, ESTOR is able to overcome the inherent suboptimality of standard EtC approaches. We now introduce the following mild assumption for our ESTOR to support our regret analysis:

**Assumption 3.3.** There exists some constant $C > 0$ such that $\mathbb{E}(p_v(X^\top v)^2) \le C$ for all $v \in \mathbb{R}^d$ with $\|v\|_\infty = 1$ where $X$ is sampled from the distribution $\mathcal{D}$ and $p_v(\cdot)$ is the density of $X^\top v$.

*Remark* 3.4. Assumption 3.3 is quite mild and applies to a broad class of distributions. For example, if the density function $p_v(\cdot)$ is bounded, then its second moment is automatically finite and hence

Assumption 3.3 holds. This scenario encompasses many standard cases, including $X$ sampled from any multivariate normal distribution. Furthermore, if the entries of $X$ follow the common sub-Gaussian property, then $X^\top v$ is also sub-Gaussian with some parameter. This implies its density function exhibits exponentially decaying tails and thus is well-controlled and integrable in tails. Overall, these observations suggest that Assumption 3.3 is easily satisfied in practice, and a detailed discussion is deferred to Appendix J. Notably, our Algorithm 2 and its regret analysis (Theorem 3.5) rely only on the validity of this assumption, without requiring knowledge of the value $C$.

Now we will present the regret bound of our ESTOR shown in Algorihtm 2:

**Theorem 3.5.** *For any single index bandit with an increasing function, we set* $\lambda = 0, \tau_i = \sqrt{3(e_{i-1} - e_{i-2})/\log(2d\log_2(T)/\delta)}$*, and* $T_0$ *to any value s.t.* $T_0 \leq d\sqrt{K^3 T \log(2d\log_2(T)/\delta)}$ *in Algorithm 2. Then under Assumption 2.2, 2.3, 3.3, our Algorithm 2 can attain following regret bound with probability at least* $1 - \delta$:

$$R_T = O\left(dK^{\frac{3}{2}}\sqrt{C \cdot T \cdot \log(d\log_2(T)/\delta)}\right) = \tilde{O}\left(dK^{\frac{3}{2}}\sqrt{T}\right) = \tilde{O}_T(\sqrt{T}).$$

**Proof sketch of Theorem 3.5** : We begin by characterizing the sampling distribution over arms induced by greedy selection within each epoch of ESTOR. Leveraging this structure, we apply Stein's identity to relate $\mathbb{E}[y_i S(x_i)]$ to $\theta_*$ and bound the estimation error of $\hat{\theta}_i$ using samples from the previous epoch, supported by a novel moment analysis of the score function under the induced distribution. We then show that the per-round regret within each epoch is upper bounded by the estimation error of $\hat{\theta}_i$, and that due to the exponential growth of epoch lengths, these error terms decay geometrically across epochs. Summing over all epochs yields the final regret bound of $\tilde{O}(dK^{3/2}\sqrt{T})$.

*Remark* 3.6. The regret bound derived in Theorem 3.5 exhibits a worst-case dependency on $K$ of order $K^{3/2}$ under Assumption 3.3, based on a conservative bounding technique. However, we can show that under many common scenarios, including when the arm $x_{t,i}$ is sampled from a $d$-dimensional normal distribution, i.e. $\mathcal{N}(\mu_X, \Sigma_X)$, the dependency on $K$ improves to $O(\sqrt{\log(K)})$ when $K$ is large. It is noteworthy that this order of $K$ matches the lower bound deduced in Li et al. (2019) for the simpler linear contextual bandit problem, despite our study addressing a significantly more challenging setting due to the unknown reward function. A detailed discussion is presented in Appendix K.

It is essential to highlight that $T$ is the dominant term as the bandit literature primarily focuses on the order of $T$ (Ding et al., 2021; Lattimore & Szepesvári, 2020), and our Algorithm 2 achieves a nearly optimal dependence on $T$ as $\tilde{O}_T(\sqrt{T})$. Furthermore, ESTOR is highly efficient in both time and memory complexity. Since the estimator is computed via a simple average over each epoch and the algorithm performs greedy action selection at each round, the total time complexity is $O(dT)$. Moreover, ESTOR only maintains the current parameter estimate and another vector containing the sufficient statistics for the truncated mean, yielding a total storage complexity of $O(d)$. Beyond its strong theoretical guarantee, ESTOR stands out as one of the most efficient algorithms (Ding et al., 2021) in terms of both computation and space among GLB methods, despite addressing a substantially more difficult setting with unknown reward functions.

## 3.4 EXTENSION TO SPARSE HIGH-DIMENSIONAL BANDITS AGNOSTIC TO INCREASING REWARD FUNCTIONS

With high-dimensional sparse data becoming increasingly prevalent in modern applications, a growing body of work has focused on sparse linear bandits, *a.k.a.* LASSO bandits (Jang et al., 2022; Hao et al., 2020), where each arm is represented by a high-dimensional feature vector, but the underlying parameter $\theta_*$ is assumed to be sparse with at most $s \ll d$ nonzero entries. To extend our methods to this sparse high-dimensional regime, we incorporate $l_1$-regularization into our loss function defined in Eqn. (1), i.e. $\lambda > 0$. By carefully choosing the regularization parameter $\lambda$, we establish Theorem 3.7. Notably, we achieve the same optimal error bound as in the low-dimensional case (Theorem 3.1), with the ambient dimension $d$ replaced by the sparsity index $s$, without the knowledge of $s$.

**Theorem 3.7.** (Bound of Sparse SIM) *For any sparse single index model where* $\theta_*$ *has* $s$ *nonzero entries, and samples* $x_1, \ldots, x_n$ *are drawn from some distribution* $\mathcal{D}$. *We denote*

$\mu_* := \mathbb{E}(f'(X^\top \theta_*))$, $X \sim \mathcal{D}$ and assume $\mu_* \neq 0$. *Under Assumption 2.2, 2.3, by solving the optimization problem in Eqn.* (1) *with* $\tau = \sqrt{3(\sigma^2 + L_f^2)Mn/\log(2d/\delta)}$, *and*

$$\lambda = 11 \cdot \sqrt{\frac{M(\sigma^2 + L_f^2)\log(2d/\delta)}{n}},$$

*with probability at least* $(1 - \delta)$ *it holds that:*

$$\left\| \hat{\theta} - \mu_* \theta_* \right\|_2 \leq \frac{3}{4}\lambda\sqrt{s} = \tilde{O}\left(\sqrt{\frac{s}{n}}\right), \quad \left\| \hat{\theta} - \mu_* \theta_* \right\|_1 \leq 3\lambda s = \tilde{O}\left(\frac{s}{\sqrt{n}}\right).$$

Remarkably, our efficient STOR and ESTOR algorithms can be directly applied to the sparse high-dimensional SIB setting by simply replacing the original estimator with the solution to the $l_1$-regularized loss function in Eqn. (1) at each epoch, i.e. line 4 of Algorithm 1 and line 5 of Algorithm 2. Both methods not only retain their optimal computational and memory efficiency, but also inherit the regret bounds established in Theorem 3.2 and Theorem 3.5, with the ambient dimension $d$ replaced by the sparsity index $r$, all without requiring the value of $r$. This fact is formalized in Corollary 3.8.

**Corollary 3.8.** *For any sparse single index bandit with an increasing function:*

1. *we set* $T_1 = (sT)^{\frac{2}{3}}\log(2d/\delta)^{\frac{1}{3}}$, $\tau = \sqrt{3(\sigma^2 + L_f^2)MT_1/\log(2d/\delta)}$ *and* $\lambda = 11\sqrt{(\sigma^2 + L_f^2)M\log(2d/\delta)/n}$ *in Algorithm 1. Then under Assumption 2.2, 2.3, we have the following regret bound with probability at least* $1 - \delta$:

$$R_T = O\left(s^{\frac{2}{3}}T^{\frac{2}{3}}(\log(d/\delta))^{\frac{1}{3}}\right) = \tilde{O}\left(s^{\frac{2}{3}}T^{\frac{2}{3}}\right).$$

2. *we set* $\tau_i = \sqrt{3(\sigma^2 + L_f^2)(CK^3 + MK)(e_{i-1} - e_{i-2})/\log(2d\log_2(T)/\delta)}$, $\lambda_i = 11\sqrt{(\sigma^2 + L_f^2)(CK^3 + MK)\log(2d/\delta)/(e_{i-1} - e_{i-2})}$ *and* $T_0$ *to any value s.t.* $T_0 \leq d\sqrt{K^3 T\log(2d\log_2(T)/\delta)}$ *in Algorithm 2. Then under Assumption 2.2, 2.3, 3.3, our Algorithm 2 can attain following regret bound with probability at least* $1 - \delta$:

$$R_T = O\left(sK^{\frac{3}{2}}\sqrt{T \cdot \log(d\log_2(T)/\delta)}\right) = \tilde{O}\left(sK^{\frac{3}{2}}\sqrt{T}\right) = \tilde{O}_T(\sqrt{T}).$$

Notably, both our SIM estimator and the ESTOR algorithm do not rely on the value of the sparsity level s in either theory or implementation under the sparse high-dimensional setting. For STOR, the value of $s$ is used solely to set the exploration phase length $T_1$. Even if we remove this dependence and set $T_1$ independently of $s$, STOR can still achieve a regret bound of order $\tilde{O}(sT^{2/3})$, where the exponent of $s$ increases from $2/3$ to $1$.

## 3.5 EXTENSION TO ARBITRARY REWARD FUNCTIONS

Finally, we study the SIB problem with arbitrary continuously differentiable reward functions. As this problem is very complicated, and modern offline works on general SIMs still rely on restrictive distributional assumptions (Härdle, 2004), we make a similar choice in this initial exploration of the online setting by assuming that each feature vector is drawn from a Gaussian distribution, i.e. $\mathcal{D} = N(\mu_X, \Sigma_X)$, for some positive definite matrix $\Sigma_X$. Similar to the identifiability assumption in Section 2, we make a slightly modified assumption purely for analytical convenience with $\|\Sigma_X^{1/2}\theta_*\|_1 = 1$. Additionally, for the general SIBs, we have to assume $\mu_* = \mathbb{E}(f'(X^\top \theta_*)) > 0$, $X \sim D$ to avoid further identifiability issues, e.g., $f(x) = f(-x)$ implies that $\theta_*$ and $-\theta_*$ are indistinguishable.

We first compute the estimator $\hat{\theta}$ based on Eqn. (1) with $\lambda = 0$ in the low-dimensional setting, followed by a normalization step $\hat{\theta}_0 = \hat{\theta}/\|\Sigma_X^{1/2}\hat{\theta}\|_1$ to scale the estimator onto a fixed and stable manifold. We then propose a novel approach to estimate the unknown link function $f(\cdot)$ via kernel

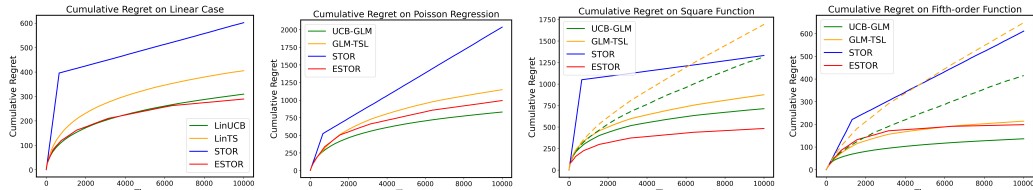

Figure 1: Plots of regrets of STOR, ESTOR, and the baseline methods under linear (1) and generalized linear (2)-(4) scenarios. Misspecified models are shown as dashed lines in (3) and (4).

regression (Härdle, 2004): given another independent set of samples $\{y_i, x_i\}_{i=1}^m$, the predictor $\hat{f}(\cdot)$ is given by,

$$\hat{f}(z) = \begin{cases} \frac{\sum_{i=1}^m y_i K_h(z - x_i^\top \hat{\theta}_0)}{\sum_{i=1}^m K_h(z - x_i^\top \hat{\theta}_0)}, & \text{if } |z - \mu_X^\top \hat{\theta}_0| \leq W, \\ 0, & \text{otherwise,} \end{cases} \quad \forall z \in \mathbb{R}, \text{ where } K_h(v) = \frac{1}{h} \mathbb{1}_{\{|v| \leq h\}}, \quad (2)$$

and $W$ and $h$ are the bandwidth and truncation parameters for the uniform kernel, respectively. We adopt the convention $0/0 = 0$. To handle this fully agnostic case with arbitrary reward functions, we propose a new algorithm, General STein's ORacle single index bandit (GSTOR), which adopts a double exploration-then-commit strategy. Due to space limit, the full algorithm is presented in Algorithm 3 in Appendix L. Specifically, GSTOR uses two separate exploration phases: the first estimates the parameter via Eqn. (1) using $T_1$ samples $\{y_i, x_i\}_{i=1}^{T_1}$, and the second approximates the reward function based on the kernel regression via Eqn. 2 with an additional $T_1$ independent samples $\{y_i, x_i\}_{i=T_1+1}^{2T_1}$. Remarkably, we prove that this design yields a prediction error bound $\tilde{O}(\sqrt{d}\,T_1^{-1/3})$, as formalized in Appendix L.3, through a novel integration of kernel smoothing analysis and perturbation arguments. Following the double exploration phases, GSTOR selects arms greedily based on the estimated parameter and the approximated reward function. The expected regret bound of GSTOR is given in Theorem 3.9, with its detailed proof deferred to Appendix L.

**Theorem 3.9.** *For any SIB problem, we set* $T_1 = d^{\frac{3}{8}} T^{\frac{3}{4}}$, $\tau = \sqrt{3T_1 / \log(2d/\delta)}$, $W = 2\log(T_1)$, $h = T_1^{\frac{1}{3}}$ *and* $\lambda = 0$ *in Algorithm 3. Then under the Gaussian design with assumptions (i).* $|f(x)| \leq L_f, |f'(x)| \leq L_{f'}, \forall x$ *(part of Assumption 2.3); (ii).* $d^{15} = O(T^2)$, *we have* $\mathbb{E}(R_T) = O(d^{\frac{3}{8}} T^{\frac{3}{4}})$.

Our GSTOR provides the first solution to SIBs with general reward functions and achieves a sublinear regret bound. Importantly, it remains unclear whether a $\tilde{O}(\sqrt{T})$ regret bound is achievable in this general setting. Even in the simpler Lipschitz bandit setting, which optimizes an unknown Lipschitz function without contextual features, the minimax regret bound is known to exceed $\sqrt{T}$. Since our setting introduces further complexity by requiring simultaneous estimation of both the continuously differentiable reward function and the parameter, it suggests that achieving $\sqrt{T}$ regret should be fundamentally unattainable without imposing additional structural assumptions.

## 4 EXPERIMENTS

In this section, we empirically validate that our proposed methods, especially ESTOR, perform efficiently across diverse link functions through simulations and real-world experiments.

For the simulation studies, we consider four types of increasing link functions and their corresponding models: (1). $f(x) = x$ with Gaussian noise $N(0, 0.25)$ (linear); (2). $f(x) = \exp(x)$ with outcomes sampled from the Poisson regression model (Poisson); (3). $f(x) = \text{sign}(x) \cdot x^2 + 2x$ with Gaussian noise $N(0, 0.25)$ (square); (4). $f(x) = x^5$ with Gaussian noise $N(0, 0.25)$ (fifth). For the linear case (1), we use LinUCB (Li et al., 2010) and LinTS (Agrawal & Goyal, 2013) as baselines. For the generalized linear cases (2)–(4), we compare our methods against the widely used UCB-GLM (Li et al., 2017) and GLM-TSL (Kveton et al., 2020) algorithms. We set the time horizon to $T = 10,000$, with detailed experimental configurations deferred to Appendix M due to space constraints. Average regrets over 20 repetitions are displayed in Figure 1. Notably, for cases (3) and (4), we further evaluate UCB-GLM and GLM-TSL by fitting them under one reward model and deploying them under the other to assess robustness under misspecification. Solid lines represent correctly specified models, while dashed lines represent misspecified ones.

From the results in Figure 1, we observe that ESTOR consistently achieves comparable performance to the best baselines across all four types of link functions, demonstrating its high efficiency. In contrast, STOR exhibits a clear exploration-then-exploitation pattern in its regret curve, which initially grows and then flattens out. Moreover, when UCB-GLM and GLM-TSL are fitted under an incorrect underlying reward function, both methods suffer significant performance degradation and slower convergence rates. This observation further highlights the vulnerability of traditional GLB algorithms to model misspecification, underscoring the importance of designing agnostic approaches like ours that can maintain robustness across diverse reward structures. Furthermore, we report the average running time of different methods from Figure 1 in Table 1 (Appendix M). Aligned with our time complexity analysis in Subsection 3.3, ESTOR and STOR exhibit substantial computational advantages, performing hundreds of times faster than UCB-GLM and thousands of times faster than GLM-TSL. This fact demonstrates their practical scalability for real applications. We also include the performance of GSTOR across the four cases in Table 2 in Appendix M. While GSTOR is designed for general reward functions and is thus generally less efficient than STOR in these monotone settings, it consistently outperforms GLB-based algorithms under model misspecification, demonstrating its robustness across a broader range of reward functions.

Due to space constraints, we defer the simulations on the sparse high-dimensional case introduced in Subsection 3.4, as well as two real-world experiments on the Forest Cover Type dataset (Blackard, 1998) and the Yahoo news dataset (Chu et al., 2009) involving the GSTOR method presented in Subsection 3.5, to Appendix M. According to Table 3, all our proposed methods consistently outperform state-of-the-art GLB methods. This stems from the fact that the underlying link function is typically unknown in real-world applications, and GLB methods are vulnerable to model misspecification. These results firmly underscore the practical value of our agnostic approach in reality.

## 5 Conclusion

In this work, we introduce a new problem of generalized linear bandits with unknown reward functions, *a.k.a.* single index bandits. We develop a family of efficient algorithms, starting with the case of monotonic reward functions and extending to sparse high-dimensional and non-monotonic settings with solid regret analysis. Central to our approach is a novel Stein's-method-based estimator achieving the optimal error rate under mild assumptions. We validate the effectiveness of our algorithms through comprehensive simulations and real experiments. Our work lays the foundation for various promising directions in future research on the SIB problem.

**Limitation and Future Work:** Our analysis assumes the arm set is drawn i.i.d. from some fixed distribution, rather than adversarially selected as in parts of the GLB literature (Lattimore & Szepesvári, 2020). Moreover, regret analysis of GSTOR relies on the Gaussian design assumption, which is standard in studies of SIMs and kernel regression due to the inherent estimation difficulty (Carroll et al., 1997). Extending these results to more general settings remains a challenging open direction, even in the offline statistical learning context.

## Acknowledgements

We appreciate the constructive feedback from the anonymous reviewers and area chair. Bongsoo Yi and Yao Li were supported in part by the National Science Foundation under grants DMS-2152289 and DMS-2134107. Doudou Zhou was supported by the NUS Start-Up Grant (A-0009985-00-00) and the MOE AcRF Tier 1 Grant (A-8003569-00-00).

ETHICS STATEMENT

We strictly adhere to the ICLR Code of Ethics. This study uses only public and simulated data, involves no human subjects, and raises no foreseeable ethical concerns.

REPRODUCIBILITY STATEMENT

We have taken necessary steps to ensure the reproducibility of our results. All algorithmic details are provided in the main paper, including clear pseudocode for our proposed methods. Complete theoretical proofs of all main results are included in the Appendix. Experimental details such as dataset descriptions and preprocessing steps are presented in the Appendix. In addition, we submit the source codes as part of the supplementary material for reproducibility.

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

# Appendix

## A   THE USE OF LARGE LANGUAGE MODELS

We used a large language model (LLM) minimally for minor sentence rewriting and language polishing. The LLM was not used for any research-related tasks, and all scientific contributions were developed solely by the authors.

## B   RELATED WORK AND DISCUSSION

**Generalized Linear Bandits:**   Filippi et al. (2010) first introduced the GLB problem (Lattimore & Szepesvári, 2020) by extending the linear bandit framework (Abbasi-Yadkori et al., 2011; Dani et al., 2008) and proposed GLM-UCB with a regret bound of order $\tilde{O}(d\sqrt{T})$, and Li et al. (2017) subsequently developed improved UCB-style algorithms with tighter regret bounds. A parallel line of work explores TS approaches for GLBs. Chapelle & Li (2011) proposed Laplace-TS under the Gaussian design, and then Abeille & Lazaric (2017); Russo & Van Roy (2014) showed that TS can be analyzed similarly to UCB under general distribution by bounding the information ratio. More recently, Kveton et al. (2020) demonstrated that their randomized TS-based methods achieve nearly optimal regret bounds, up to logarithmic factors. To improve the scalability of GLBs, Zhang et al. (2016) and Jun et al. (2017) proposed algorithms based on the online Newton step, achieving $\tilde{O}(\sqrt{T})$ regret with improved time and space efficiency. Ding et al. (2021) introduced another efficient GLB algorithm that combines stochastic gradient descent with the TS framework to enhance computational performance. Recently, Rajaraman et al. (2024) briefly examined a closely related GLB problem with an unknown link function as part of a broader study. However, they only consider more restrictive assumptions, including fixed action sets, Gaussian noise, and strong structural constraints on the reward function. Their method is also computationally intensive and lacks empirical validation. In contrast, our work offers a more general and efficient framework that supports non-monotone link functions, handles high-dimensional sparsity, and achieves optimal statistical and computational guarantees. The very recent work by Arya & Song (2025) focuses on a related but different setting, a batched multi-armed bandit with covariates, using a single index modeling approach combined with dynamic binning and arm elimination. This work represents a concurrent and independent effort. To the best of our knowledge, our work provides the first comprehensive treatment of the single index bandit problem under mild assumptions.

**Single Index Models:**   The single index model (SIM) has been extensively studied in the low-dimensional setting (Han, 1987; Härdle, 2004; Carroll et al., 1997), where most approaches estimated the unknown parameter via (quasi-)maximum likelihood estimation and establish asymptotic guarantees using central limit theorems. For non-asymptotic bounds, seminal works such as Thrampoulidis et al. (2015); Na et al. (2019); Plan & Vershynin (2016); Neykov et al. (2016) employed traditional regression techniques such as $l_1$-regularization. These results show that, under standard Gaussian covariates and certain conditions on the link function, the estimator can achieve the same error rates as our Theorem 3.1 and Theorem 3.7. However, these guarantees crucially rely on restrictive distributional assumptions such as standard Gaussian covariates. More recently, Yang et al. (2017) relaxed these distributional assumptions by proposing an efficient estimator, and Fan et al. (2023) further extended it with a regularization-free approach based on overparameterization that achieves optimal non-asymptotic rates, but still assumes that each entry of the context vector is i.i.d. from a known distribution and that the noise has finite fourth moment. In summary, due to the inherent challenges of SIM estimation, prior statistical literature imposes strong assumptions on the data distribution, whereas our work requires milder conditions.

Since part of our work focus on the case where the unknown link function is monotonically increasing, we also examine a line of recent work on statistical estimation in monotone SIMs. Balabdaoui et al. (2019a) propose a least-squares estimator for monotone SIMs, establishing a convergence rate of $n^{-1/3}$ under a suite of strong assumptions on the noise and covariate distributions (see assumptions A1–A6 therein). To improve upon this, Balabdaoui et al. (2019b) propose a score-based estimator with the same $n^{-1/3}$ rate under milder assumptions, and achieves the optimal $n^{-1/2}$ rate when the link function is piecewise constant. More recently, Dai et al. (2022) extend this line to the sparse,

high-dimensional setting, but their projection-based estimator still only guarantees a $n^{-1/3}$ rate under nontrivial distributional assumptions. Collectively, these results underscore the statistical and algorithmic challenges of efficient monotone SIM estimation even under nontrivial assumptions, and motivate our pursuit of a computationally cheap and distributionally robust method with optimal convergence guarantees.

**Contextual Bandits under the Realizability Assumption:** The general contextual bandit problem under the realizability assumption was initiated by Agarwal et al. (2012). A substantial line of work builds on this assumption by reducing the problem to solving either offline or online square-loss regression oracles (Foster et al., 2018; Foster & Rakhlin, 2020; Simchi-Levi & Xu, 2022; Foster & Krishnamurthy, 2021; Zhu & Mineiro, 2022; Zhang et al., 2023; Pacchiano, 2024; Ye et al., 2025). However, as discussed in Section 1, such regression oracles are infeasible under SIMs due to the composite and nonparametric structure of the reward function. Beyond this limitation, the regret bounds of modern works (Pacchiano, 2024; Ye et al., 2025) rely on some complexity measure of the function class, such as the eluder dimension (Russo & Van Roy, 2014). However, in our proposed SIBs with unknown reward functions, the eluder dimension becomes unbounded, making these results inapplicable again.

To illustrate this point, we use FALCON+ (Simchi-Levi & Xu, 2022) as a representative state-of-the-art example to explain why contextual bandit algorithms under the realizability assumption fail under the SIB setting in detail. FALCON+ critically relies on their Assumption 2, which assumes access to an offline oracle that can estimate the full reward function, comprising both the unknown parameter vector and the unknown reward function, with a provable error bound, using data collected via randomized sampling only from the previous epoch (i.e., line 6 of the algorithm). To achieve the optimal $\sqrt{T}$ regret bound, this oracle must guarantee an estimation rate of the optimal order $n^{-1/2}$. However, no existing method for single index models comes close to satisfying this requirement. As discussed in our review of SIM and monotone SIM literature above, existing estimators rely on restrictive distributional assumptions (e.g., i.i.d. Gaussian features). Moreover, the randomized sampling scheme in FALCON+ produces covariate distributions that lie far outside the regimes where existing estimators have any theoretical guarantee, let alone achieve the optimal rate of $n^{-1/2}$. As a result, although FALCON+ is conceptually insightful, its reliance on an idealized regression oracle renders it inapplicable to the SIB setting with any provable theoretical guarantees. Furthermore, existing methods are computationally expensive as the least squares solver leads to infinite-dimensional and non-convex optimization problems under SIMs Fan et al. (2023). These challenges highlight that existing methods are fundamentally inadequate for the SIB setting, necessitating the development of a completely new solution.

## C  SUMMARY OF TECHNICAL NOVELTY

In this section, we summarize the technical contributions of our work in detail. We start from utilizing the novel Stein's-method-based estimator in Eqn. (1), which enables us to estimate the parameter $\theta_*$ even when the reward function $f(\cdot)$ is unknown. Specifically, Stein's identity implies that $\mathbb{E}[f(X^\top \theta_*) \cdot S(X)] = \mu_* \theta_*$ for some scalar $\mu_* \neq 0$, under mild regularity conditions. Thus, taking a (truncated) average of $y_i \cdot S(x_i)$ yields an estimator that is directionally aligned with $\theta_*$. The truncation function $\phi_\tau$ is essential to control the variance of the estimator in the presence of heavy-tailed rewards, as we show through the bias-variance trade-off in the proof. Note our estimator is computationally efficient, does not require knowledge of the reward function, and crucially, achieves the optimal estimation rate under mild assumptions. Moreover, we extend this estimator to the high-dimensional sparse regime with a simple $\ell_1$ regularizer, and it still achieves the minimax-optimal rate without requiring knowledge of the sparsity level $s$ in Theorem 3.7.

The proof of Theorem 3.1 follows a classical comparison argument. We first apply Stein's identity to show that $\mathbb{E}[y_i S(x_i)] = \mu_* \theta_*$, which motivates our estimator. We then analyze the optimization gap between the empirical loss $L(\hat{\theta})$ and $L(\mu_* \theta_*)$ by bounding the gradient norm $|\nabla L(\mu_* \theta_*)|_\infty$, where the bias is controlled by truncation and the concentration is handled via Bernstein's inequality. Finally, applying a first-order Taylor expansion and standard convexity arguments yields the $\ell_2$ and $\ell_1$ estimation bounds. Notably, the estimator admits a closed-form solution in the low-dimensional setting, achieving $\tilde{O}(\sqrt{d/n})$ rate without requiring knowledge of $f(\cdot)$. For Theorem 3.7, we adapt

this argument to the sparse high-dimensional regime by adding an $\ell_1$-penalty to the loss and leveraging subgradient optimality conditions. While the core Stein's-method-based structure remains intact, we use a support-splitting analysis to decouple estimation over the true support versus its complement. This enables us to obtain sparsity-adaptive rates of $\tilde{O}(\sqrt{s/n})$ in $\ell_2$ norm and $\tilde{O}(s/\sqrt{n})$ in $\ell_1$ norm, again without requiring prior knowledge of the link function.

Since our estimator, grounded in Stein's identity, avoids the need to explicitly estimate the unknown reward function, a central advantage of our STOR and ESTOR emerges in the monotonic setting: when the link function is assumed to be monotone increasing, the relative ordering of $f(x^\top \theta)$ is preserved by $x^\top \hat{\theta}$. This allows the algorithm to perform greedy action selection based solely on $x^\top \hat{\theta}$, without requiring knowledge of the explicit form of $f(\cdot)$. As a result, our method achieves substantial gains in sample efficiency by sidestepping the additional exploration cost typically associated with function estimation. Importantly, this setting subsumes the entire class of generalized linear bandits (GLB) with canonical exponential family link functions, which are inherently monotone (McCullagh, 2019). In this way, our estimator leverages structural properties of the reward function class to implicitly handle function uncertainty, enabling principled decision-making without direct function recovery.

In particular, our ESTOR achieves the minimax-optimal regret bound of $\tilde{O}(\sqrt{T})$ without requiring explicit knowledge of the link function. Explore-then-Commit (EtC) algorithms are known to be suboptimal in regret minimization, as parameter estimation is performed exclusively using data from the initial exploration phase, ignoring informative samples collected thereafter. ESTOR is an epoch-based variant of STOR with exponentially growing epoch lengths. In ESTOR, the parameter estimate at the start of each epoch is computed using samples from the preceding epoch, followed by greedy selection in the current epoch. This design ensures continual refinement of the estimate and overcomes the inefficiencies of fixed-sample EtC schemes. Notably, the SIM estimator integrates naturally into this framework: greedy selection induces a tractable sampling distribution within each epoch, allowing Stein's identity to be applied for efficient parameter estimation. For the proof of Theorem 3.5, we characterize this distribution and bound the estimation error at each epoch. Since epoch lengths grow exponentially, the error terms decay geometrically across epochs. Summing the regret over all epochs yields a cumulative regret of $\tilde{O}(dK^{3/2}\sqrt{T})$.

Finally, in the general setting with arbitrary differentiable reward functions, we introduce GSTOR, which employs a double exploration strategy. In the first stage, we estimate the parameter $\theta_*$ using our Stein-based estimator. In the second stage, we estimate the unknown reward function $f(\cdot)$ via kernel regression over the scalar projections $x^\top \hat{\theta}$. We then apply an Explore-then-Commit (EtC) strategy using the estimated $\hat{f}$ to identify the best arm. This two-phase design decouples parameter and function estimation, enabling our SIB framework in a more general class of problems.

## D    PROOF OF THEOREM 3.1

### D.1    USEFUL LEMMAS

**Lemma D.1.** *(Generalized Stein's Lemma, Diaconis et al. (2004)) For a d-dimensional continuous random variable $X \in \mathbb{R}^d$ with continuously differentiable density function $p : \mathbb{R}^d \to \mathbb{R}$, and any continuously differentiable function $f : \mathbb{R}^d \to \mathbb{R}$. Denote $S(X) : \mathbb{R}^d \to \mathbb{R}^d$ as the score function associated with $X$, i.e. $S(X) = -\nabla_X p(X)/p(X)$. If the expected values of both $\nabla f(X)$ and $f(X) \cdot S(X)$ in terms of the density $p$ exist, then it holds that*

$$\mathbb{E}[f(X) \cdot S(X)] = \mathbb{E}[\nabla f(X)].$$

**Lemma D.2.** *(Bernstein's Inequality, Wainwright (2019) Proposition 2.14) Let $X_1, \ldots, X_n$ be real-valued random variables such that $X_i \leq b$ almost surely for all $i = 1, \ldots, n$, then for any $t > 0$ we have that*

$$\mathbb{P}\left(\left|\sum_{i=1}^n (X_i - \mathbb{E}(X_i))\right| \geq \sqrt{2t} \cdot \sqrt{\sum_{i=1}^n \mathbb{E}(X_i^2)} + \frac{bt}{3}\right) \leq 2e^{-t}.$$

**Lemma D.3.** *Assume we have $\mu_* = \mathbb{E}(f'(X^\top \theta_*)) \neq 0$ where $X$ denotes the random vector drawn from the density $p(\cdot)$. By setting*

$$\tau = \sqrt{\frac{3n(\sigma^2 + S_f^2)M}{\log(2d/\delta)}},$$

*in the single index model setting to solve $\hat{\theta}$ according to Eqn. (1) with $\lambda = 0$. Then with probability at least $1 - \delta$ ($0 < \delta < 1$), it holds that,*

$$\|\nabla L(\mu_* \theta_*)\|_\infty \leq \left(\frac{4\sqrt{3}}{3} + 2\sqrt{2}\right)\sqrt{\frac{M(\sigma^2 + S_f^2)\log(2d/\delta)}{n}}.$$

*Proof.* Recall that we have

$$L(\theta) = \langle \theta, \theta \rangle - \frac{2}{n}\sum_{i=1}^n \langle \phi_\tau(y_i \cdot S(x_i)), \theta \rangle,$$

where $\phi_\tau(y_i \cdot S(x_i)) = \text{sign}(y_i S(x_i)) \cdot (|y_i S(x_i)| \wedge \tau)$ and the operation happens on the vector elementwisely. According to Lemma D.1, it holds that for any $i \in [n]$

$$\mathbb{E}(y_i S(x_i)) = \mathbb{E}(f(x_i^\top \theta_*)S(x_i)) = \mathbb{E}(f'(x_i^\top \theta_*)) \cdot \theta_* = \mu_* \theta_*.$$

Therefore, it holds that

$$\nabla L(\mu_* \theta_*) = 2\mu_* \theta_* - \frac{2}{n}\sum_{i=1}^n \phi_\tau(y_i \cdot S(x_i))$$

$$= 2\mathbb{E}(y_1 S(x_1)) - \frac{2}{n}\sum_{i=1}^n \phi_\tau(y_i \cdot S(x_i)).$$

Based on the above equation, we have that

$$\|\nabla L(\mu_* \theta_*)\|_\infty = \left\|2\mathbb{E}(y_1 S(x_1)) - \frac{2}{n}\sum_{i=1}^n \phi_\tau(y_i \cdot S(x_i))\right\|_\infty$$

$$\leq \underbrace{\|2\mathbb{E}(y_1 S(x_1)) - 2\mathbb{E}(\phi_\tau(y_1 \cdot S(x_1)))\|_\infty}_{:= \alpha_1} + \underbrace{\left\|2\mathbb{E}(\phi_\tau(y_1 \cdot S(x_1))) - \frac{2}{n}\sum_{i=1}^n \phi_\tau(y_i \cdot S(x_i))\right\|_\infty}_{:= \alpha_2}.$$

To bound $\alpha_1$, we first note that for each index $j \in [d]$, it holds that

$$|2\mathbb{E}(y_1 S_j(x_1)) - 2\mathbb{E}(\phi_\tau(y_1 \cdot S_j(x_1)))| \leq 2 \cdot \mathbb{E}\left(|y_1 S_j(x_1)| \mathbb{1}_{\{|y_1 S_j(x_1)| > \tau\}}\right).$$

And then we can easily bound the above value by

$$\mathbb{E}\left(|y_1 S_j(x_1)| \mathbb{1}_{\{|y_1 S_j(x_1)| > \tau\}}\right)^2 \overset{(i)}{\leq} \mathbb{E}\left(y_1^2 S_j(x_1)^2\right) \cdot \mathbb{E}\left(\mathbb{1}_{\{|y_1 S_j(x_1)| > \tau\}}^2\right)$$

$$\leq \mathbb{E}\left(y_1^2 S_j(x_1)^2\right) \cdot \mathbb{P}\left(|y_1 S_j(x_1)| > \tau\right)$$

$$\overset{(ii)}{\leq} \left[\mathbb{E}\left(f(x_1^\top \theta_*)^2 S_j(x_1)^2\right) + \mathbb{E}\left(\eta_1^2\right)\mathbb{E}\left(S_j(x_1)^2\right)\right] \cdot \mathbb{P}\left(|y_1 S_j(x_1)| > \tau\right)$$

$$\overset{(iii)}{\leq} \left(\sigma^2 + L_f^2\right) \cdot M \cdot \frac{\mathbb{E}|y_1 S_j(x_1)|^2}{\tau^2} = \left(\sigma^2 + L_f^2\right)^2 \cdot \frac{M^2}{\tau^2},$$

where we have the inequality (i) due to Holder's inequality, and we can deduce the inequality (ii) based on the fact that the white noise $\eta_1$ is independent with the arm $x_1$. The inequality (iii) comes from Chebyshev's inequality. Since the above result holds for all index $j \in [d]$, which indicates that

$$\alpha_1 \leq \frac{2(\sigma^2 + L_f^2)M}{\tau}. \tag{3}$$

On the other hand, since we have that $\{y_i S(x_i)\}_{i=1}^n$ are i.i.d. samples, and for any $i \in [n], j \in [d]$ $|\phi_\tau(y_i S_j(x_i))| \leq \tau$, $\mathrm{Var}\left(\phi_\tau(y_i \cdot S_j(x_i))\right) \leq \mathbb{E}\left(\phi_\tau(y_i S_j(x_i))^2\right) \leq \mathbb{E}\left(y_i^2 S_j(x_i)^2\right) \leq (\sigma^2 + L_f^2)M$, then based on Bernstein's inequality in Lemma D.2, we have that for any $j \in [d]$,

$$\mathbb{P}\left(2\left|\frac{1}{n}\sum_{i=1}^n \phi_\tau(y_i \cdot S_j(x_i)) - \mathbb{E}\left(\phi_\tau(y_1 \cdot S_j(x_1))\right)\right| \geq 2\sqrt{\frac{2(\sigma^2 + L_f^2)M\log(2/\delta)}{n}} + \frac{2\tau\log(2/\delta)}{3n}\right)$$
$$\leq \delta. \qquad (4)$$

Taking union bound over $j \in [d]$ in the above Eqn. (4) yields that

$$\mathbb{P}\left(\alpha_2 \geq 2\sqrt{\frac{2(\sigma^2 + L_f^2)M\log(2d/\delta)}{n}} + \frac{2\tau\log(2d/\delta)}{3n}\right) \leq \delta. \qquad (5)$$

Combining the results in Eqn. (3) and Eqn. (5), with probability at least $1 - \delta$, it holds that,

$$\|\nabla L(\mu_\star \theta_*)\|_\infty \leq \frac{2(\sigma^2 + L_f^2)M}{\tau} + 2\sqrt{\frac{2(\sigma^2 + L_f^2)M\log(2d/\delta)}{n}} + \frac{2\tau\log(2d/\delta)}{3n}. \qquad (6)$$

By taking

$$\tau = \sqrt{\frac{3n(\sigma^2 + S_f^2)M}{\log(2d/\delta)}},$$

into Eqn. (6), we finally have that with probability at least $1 - \delta$,

$$\|\nabla L(\mu_* \theta_*)\|_\infty \leq \left(\frac{4\sqrt{3}}{3} + 2\sqrt{2}\right)\sqrt{\frac{M(\sigma^2 + S_f^2)\log(2d/\delta)}{n}}.$$

$\square$

### D.2 PROOF OF THEOREM 3.1

*Proof.* The proof is straightforward since our loss function $L(\theta)$ is a quadratic function with $\lambda = 0$. We have $L(\hat{\theta}) \leq L(\mu_* \theta_*)$ due to the choice of $\hat{\theta}$, then based on the nature of quadratic functions, it holds that

$$L(\hat{\theta}) - L(\mu_* \theta_*) = \nabla L(\mu_* \theta_*)^\top \left(\hat{\theta} - \mu_* \theta_*\right) + 2\left\|\hat{\theta} - \mu_* \theta_*\right\|_2^2.$$

Therefore, it holds that

$$2\left\|\hat{\theta} - \mu_* \theta_*\right\|_2^2 \leq \nabla L(\mu_* \theta_*)^\top \left(\hat{\theta} - \mu_* \theta_*\right)$$
$$\leq \|\nabla L(\mu_* \theta_*)\|_\infty \left\|\hat{\theta} - \mu_* \theta_*\right\|_1 \leq \|\nabla L(\mu_* \theta_*)\|_\infty \left\|\hat{\theta} - \mu_* \theta_*\right\|_2 \cdot \sqrt{d},$$

and the last two inequalities are based on Holder's inequality and Cauchy-Schwarz inequality respectively. Then by using the results in Lemma D.3, we have that with probability of at least $1 - \delta$,

$$2\left\|\hat{\theta} - \mu_* \theta_*\right\|_2^2 \leq \left(\frac{4\sqrt{3}}{3} + 2\sqrt{2}\right)\sqrt{\frac{M(\sigma^2 + S_f^2)\log(2d/\delta)}{n}} \cdot \left\|\hat{\theta} - \mu_* \theta_*\right\|_2 \cdot \sqrt{d},$$

which is identical to

$$\left\|\hat{\theta} - \mu_* \theta_*\right\|_2 \leq \left(\frac{2\sqrt{3}}{3} + \sqrt{2}\right)\sqrt{\frac{dM(\sigma^2 + S_f^2)\log(2d/\delta)}{n}}.$$

Finally, we can deduce the $l_1$-norm bound:

$$\left\|\hat{\theta} - \mu_* \theta_*\right\|_1 \leq \sqrt{d} \cdot \left\|\hat{\theta} - \mu_* \theta_*\right\|_2 \leq \left(\frac{2\sqrt{3}}{3} + \sqrt{2}\right)d \cdot \sqrt{\frac{M(\sigma^2 + S_f^2)\log(2d/\delta)}{n}}.$$

$\square$

# E    PROOF OF THEOREM 3.2

## E.1    USEFUL LEMMAS

**Lemma E.1.** *Let $f$ be a continuously differentiable and non-decreasing function on an interval $I \subset \mathbb{R}$. Let $X$ be a continuous random variable whose support is (or at least covers) the entire interval $I$. If*

$$\mathbb{E}[f'(X)] = 0,$$

*then $f$ is constant on $I$.*

*Proof.* First, because $f$ is non-decreasing, we have $f'(x)$ is a nonnegative function on $I$. Next, by hypothesis,

$$\mathbb{E}[f'(X)] = 0.$$

In terms of integration against the density of $X$, this means

$$\int_I f'(x)\, dP_X(x) = 0.$$

Since $f'(x) \geq 0$ for all $x \in I$, the integral (or expectation) of a nonnegative function being zero forces

$$f'(x) = 0 \quad \text{for almost every } x \in I.$$

Because $f$ is *continuously differentiable*, $f'(x)$ is actually a *continuous* function on $I$. A continuous function that is zero almost everywhere on an interval must be zero *everywhere* on that interval. (Otherwise, if $f'(x_0) \neq 0$ for some $x_0$, continuity would force $f'(x)$ to be nonzero on an entire neighborhood around $x_0$, contradicting the fact that $f'(x) = 0$ a.e.) Consequently,

$$f'(x) = 0 \quad \text{for all } x \in I.$$

Finally, a function whose derivative is identically zero on an interval is necessarily a constant function on that interval. Therefore, $f$ must be constant on $I$. $\qquad\square$

## E.2    PROOF OF THEOREM 3.2

*Proof.* On the one hand, if we have $\mathbb{E}\big(f'(X^\top \theta_*)\big) = 0$, then based on Lemma E.1 we know that $f(\cdot)$ is constant on the support of $X^\top \theta_*$. Therefore, the expected reward for all possible arms are constant, which indicates that cumulative regret $R_T = 0$, and Theorem 3.2 naturally holds.

On the other hand, if we have $\mathbb{E}(f'(X^\top \theta_*)) \neq 0$, then we can utilize the results from Theorem 3.1. Denote $\mu^* = \mathbb{E}(f'(X^\top \theta_*))$ as in Theorem 3.1. Then for any $t > T_1$, we have that

$$
\begin{aligned}
f(x_{t,*}^\top \theta_*) - f(x_t^\top \theta_*) &= f(x_{t,*}^\top \theta_*) - f\left(x_{t,*}^\top \frac{\hat{\theta}}{\mu^*}\right) + f\left(x_{t,*}^\top \frac{\hat{\theta}}{\mu^*}\right) - f(x_t^\top \theta_*) \\
&\leq f(x_{t,*}^\top \theta_*) - f\left(x_{t,*}^\top \frac{\hat{\theta}}{\mu^*}\right) + f\left(x_t^\top \frac{\hat{\theta}}{\mu^*}\right) - f(x_t^\top \theta_*) \\
&\leq \left| L_{f'} \cdot x_{t,*}^\top \left(\theta_* - \frac{\hat{\theta}}{\mu^*}\right) \right| + \left| L_{f'} \cdot x_t^\top \left(\theta_* - \frac{\hat{\theta}}{\mu^*}\right) \right| \\
&\leq L_{f'} \cdot \|x_{t,*}\|_\infty \left\|\theta_* - \frac{\hat{\theta}}{\mu^*}\right\|_1 + L_{f'} \cdot \|x_t\|_\infty \left\|\theta_* - \frac{\hat{\theta}}{\mu^*}\right\|_1 \qquad (7) \\
&\leq \frac{L_{f'}}{\mu^*} \cdot \|x_{t,*}\|_\infty \left\|\hat{\theta} - \mu\theta_*\right\|_1 + \frac{L_{f'}}{\mu^*} \cdot \|x_t\|_\infty \left\|\hat{\theta} - \mu\theta_*\right\|_1 \\
&\lesssim d\sqrt{\frac{M \log(2d/\delta)}{T_1}},
\end{aligned}
$$

with probability at least $1 - \delta$ where the last inequality comes from the $l_1$-norm error bound proved in Theorem 3.1. Therefore, based on the choice of $T_1$ we have that:

$$R_T = \sum_{t=1}^{T_1} \left[ f(x_{t,*}^\top \theta_*) - f(x_t^\top \theta_*) \right] + \sum_{t=T_1+1}^{T} \left[ f(x_{t,*}^\top \theta_*) - f(x_t^\top \theta_*) \right]$$

$$\lesssim 2L_f T_1 + d \sqrt{\frac{\log (2d/\delta)}{T_1}} \cdot T = O\left( d^{\frac{2}{3}} T^{\frac{2}{3}} \left( \log (2d/\delta) \right)^{\frac{1}{3}} \right) = \tilde{O}\left( d^{\frac{2}{3}} T^{\frac{2}{3}} \right).$$

$\square$

*Remark* E.2. Our final regret bound includes an additional multiplicative factor of $1/\mu_*$ according to Eqn. 7 if $\mu_*$ is not zero. For special case where $\mu_* = 0$ we have shown in Lemma E.1 that the regret is indeed zero. When $\mu_* > 0$, this factor is a constant and does not affect the order of the final regret bound. As a special case, when the derivative of the unknown reward function is assumed to be lower bounded by some constant $c > 0$, the final regret bound in the worst case inherits an additional $1/c$ multiplier without knowing the value of $c$. This fact also holds for Theorem 3.5 and Corollary 3.8.

# F  PROOF OF THEOREM 3.5

## F.1  USEFUL LEMMAS

**Lemma F.1.** *For all $a > b > 0$, the following inequality holds:*
$$\sqrt{a} - \sqrt{b} \le \sqrt{2a - 2b}.$$

*Proof.* Dividing both sides by $\sqrt{b} > 0$ and setting $x = \sqrt{a/b} > 1$, the inequality reduces to
$$x - 1 \le \sqrt{2(x^2 - 1)}.$$
Squaring both sides (which is valid since both sides are nonnegative for $x > 1$) yields
$$(x - 1)^2 \le 2(x^2 - 1).$$
Expanding and rearranging gives
$$x^2 - 2x + 1 \le 2x^2 - 2 \quad \Longleftrightarrow \quad x^2 + 2x \ge 3.$$
Since $x > 1$, the inequality $x^2 + 2x \ge 3$ is always satisfied. Thus the original inequality holds. $\square$

**Lemma F.2.** *Let $x_1, \ldots, x_K$ be i.i.d. continuous random vectors in $\mathbb{R}^d$ drawn from a common distribution $D_0$ with probability density function $p(x)$. Fix an arbitrary vector $\theta \in \mathbb{R}^d$, and define*
$$x^* = \arg \max_{1 \le j \le K} \left( x_j^\top \theta \right).$$
*Then the density function of $x^*$ is*
$$p_\theta(x) = K \, p(x) \left( F_0(x^\top \theta) \right)^{K-1},$$
*where $p_0(\cdot)$ and $F_0(\cdot)$ are the density and cumulative distribution functions of $x^\top \theta$ with $x \sim D_0$, i.e.*
$$F_0(m) = \mathbb{P}\left( x^\top \theta \le m \right) \quad \text{and} \quad p_0(m) = \frac{d}{dm} F_0(m).$$
*Furthermore, the score function $S^{p_\theta}(x) = -\nabla_x \log p_\theta(x)$ can be written as*
$$S^{p_\theta}(x) = S^p(x) - (K - 1) \frac{p_0\left( x^\top \theta \right)}{F_0\left( x^\top \theta \right)} \theta.$$

*Proof.* Let $x_1, \ldots, x_K$ be i.i.d. samples from distribution $D$. For each realization, define
$$x^* = \arg \max_{1 \le j \le K} \left( x_j^\top \theta \right).$$
We compute the density of $x^*$. We have that
$$\mathbb{P}(x^* \in dx) = \sum_{i=1}^{K} \mathbb{P}\left( x^* \in dx, \, x^* = x_i \right).$$
Since $x_i$ is drawn from some continuous distribution, the event $\{x^* \in dx\}$ occurs exactly when:

1. Exactly one of the $x_j$'s lies in $\mathrm{d}x$. Since $x_j \sim p(\cdot)$ and there are $K$ such vectors, the probability contribution is

$$K\, p(x)\, \mathrm{d}x.$$

2. Given $x_j \in \mathrm{d}x$, the remaining $K-1$ vectors satisfy $x_i^\top \theta \le x_j^\top \theta$. By independence,

$$\mathbb{P}\big(x_i^\top \theta \le x_j^\top \theta\big) \;=\; F_0\big(x_j^\top \theta\big),$$

so

$$\mathbb{P}\Big(\big(x_i^\top \theta \le x_j^\top \theta\big) \text{ for all } i \ne j\Big) \;=\; \big(F_0(x^\top \theta)\big)^{K-1}.$$

Hence,

$$\mathbb{P}\big(x^* \in \mathrm{d}x\big) \;=\; K\, p(x)\, \big(F_0(x^\top \theta)\big)^{K-1}\, \mathrm{d}x,$$

which establishes

$$p_\theta(x) \;=\; K\, p(x)\, \big(F_0(x^\top \theta)\big)^{K-1}.$$

For the score function $S^{p_\theta}(\cdot)$, taking logarithms yields

$$\log p_\theta(x) \;=\; \log K \;+\; \log p(x) \;+\; (K-1)\, \log F_0(x^\top \theta).$$

Then

$$\nabla_x \log p_\theta(x) \;=\; \frac{\nabla_x p(x)}{p(x)} \;+\; (K-1)\, \frac{1}{F_0(x^\top \theta)}\, \nabla_x F_0(x^\top \theta).$$

Moreover, by the chain rule, $\nabla_x F_0(x^\top \theta) = p_0(x^\top \theta)\, \theta$. Thus,

$$\nabla_x \log p_\theta(x) \;=\; \frac{\nabla_x p(x)}{p(x)} \;+\; (K-1)\, \frac{p_0(x^\top \theta)}{F_0(x^\top \theta)}\, \theta,$$

and so

$$S^{p_\theta}(x) = -\frac{\nabla_x p(x)}{p(x)} - (K-1)\, \frac{p_0(x^\top \theta)}{F_0(x^\top \theta)}\, \theta = S^p(x) - (K-1)\, \frac{p_0(x^\top \theta)}{F_0(x^\top \theta)}\, \theta.$$

This completes the proof. $\qquad\square$

**Lemma F.3.** *Let $X$ be a continuous random variable with probability density function $p_X$ and cumulative distribution function $F_X$. For any constant $c > 0$, define the random variable $Y = cX$ with density $p_Y$ and CDF $F_Y$. Then,*

$$\mathbb{E}\big[p_Y(Y)^2\big] = \frac{1}{c^2}\, \mathbb{E}\big[p_X(X)^2\big].$$

*Proof.* Since $Y = cX$ has density

$$p_Y(y) = \frac{1}{c}\, p_X\left(\frac{y}{c}\right)$$

evaluating at $y = cX$ yields:

$$p_Y(cX) = \frac{1}{c}\, p_X(X).$$

Taking the expectation with respect to $X$ on both sides completes the proof:

$$\mathbb{E}\big[p_Y(Y)^2\big] = \mathbb{E}\big[p_Y(cX)^2\big] = \frac{1}{c^2}\, \mathbb{E}\big[p_X(X)^2\big].$$

$\qquad\square$

F.2    PROOF OF THEOREM 3.5

*Proof.* Based on our epoch schedule $e_i = (2^i - 1)T_0, i \geq 0$, we can easily verify the length of each epoch denoted as $\{\kappa_i\}_{i=1}$ satisfying that $\kappa_i = e_i - e_{i-1} = 2^{i-1}T_0$. Therefore, this result indicates that our epoch length follows an exponential growth pattern, specifically doubling each time. We denote $\mu_i = \mathbb{E}_{X \sim p_i}(f'(X^\top \theta_*))$. Since we know that the support of $p_i(\cdot)$ is identical to that of the original $p(\cdot)$, then if $\mu_i = 0$ for any $i = 1, 2, \ldots$, then based on Lemma E.1 we know that $f(\cdot)$ is contant on the support of $X^\top \theta_*$ with $X \sim p(\cdot)$. Therefore, the expected reward for all arms is fixed and we have the cumulative regret bound $R_T = 0$. This indacates that our Theorem 3.5 simply holds. So for the rest of the proof, we will focus on the case that $\mu_i \neq 0$ for any $i = 1, 2, \ldots$.

At the beginning, we assume that the time horizon $T$ exactly matches the end of some epoch $H > 0$, i.e. $e_H = T$. Hence we have that $(2^H - 1) \cdot T_0 = T$.

Based on Lemma F.2, we have for $i \geq 2$, $p_i(x) = K \cdot p(x) \cdot F_i(x^\top \hat{\theta}_i)^{K-1}$ is the actually the density function of $y = \arg\max_{y_1, \ldots, y_K} y^\top \hat{\theta}_i$ where $y_1, \ldots, y_K$ are randomly sampled from $\mathcal{D}$. Since we assume the arm set $\mathcal{X}_t, t = 1, 2, \ldots$ consisting of $K$ random samples drawn from $\mathcal{D}$, we can deduce that all the chosen arms at epoch $i$ ($\kappa_i$) follows the distribution with density function $p_i(x)$, i.e.

$$\{x_i\}_{i=e_{i-1}+1}^{e_i} \sim p_i, \quad i = 1, 2, \ldots$$

hold, and this indicates that we can use our Stein's-method-based Theorem 3.1 to bound the error of $\hat{\theta}_i$ at each epoch. For $p_1$, we know $\mathbb{E}(S_j^{p_1}(X)) \leq M$ for all $j \in [d]$ with $X \sim p_1$ according to Assumption 2.2. To bound the second moment of the score function for $p_i$, $i > 1$. Based on Lemma F.2, we know that for $i > 1$

$$p_i(x) = K \cdot p(x) \cdot F_i(x^\top \hat{\theta}_i)^{K-1}, S^{p_i}(x) = S^p(x) - (K-1) \frac{p_i(x^\top \hat{\theta}_i)}{F_i(x^\top \hat{\theta}_i)} \hat{\theta}_i.$$

Therefore, for $X \sim p_i$ it holds that

$$\left\| \mathbb{E}(S^p(X))^2 \right\|_\infty = \left\| \int \frac{\nabla_x p(x)^2}{p(x)^2} \cdot K p(x) F_i(x^\top \hat{\theta}_i)^{K-1} dx \right\|_\infty$$

$$\leq K \left\| \int \frac{\nabla_x p(x)^2}{p(x)^2} \cdot p(x) dx \right\|_\infty \leq KM,$$

based on Assumption 2.3. On the other hand, we have that for $X \sim p_i$ and under $K > 3$,

$$\mathbb{E}\left( (K-1)^2 \frac{p_i(X^\top \hat{\theta}_i)^2}{F_i(X^\top \hat{\theta}_i)^2} \cdot \left\| \hat{\theta}_i \right\|_\infty^2 \right) = (K-1)^2 \left\| \hat{\theta}_i \right\|_\infty^2 \int K \frac{p_i(x^\top \hat{\theta}_i)^2}{F_i(x^\top \hat{\theta}_i)^2} \cdot p(x) F_i(x^\top \hat{\theta}_i)^{K-1} dx$$

$$= K(K-1)^2 \left\| \hat{\theta}_i \right\|_\infty^2 \int p_i(x^\top \hat{\theta}_i)^2 \cdot p(x) \cdot F_i(x^\top \hat{\theta}_i)^{K-3} dx$$

$$\leq K^3 \left\| \hat{\theta}_i \right\|_\infty^2 \cdot \int p_i(x^\top \hat{\theta}_i)^2 \cdot p(x) dx \tag{8}$$

$$= K^3 \left\| \hat{\theta}_i \right\|_\infty^2 \cdot \mathbb{E}_{Y \sim p}\left( p_i(Y^\top \hat{\theta}_i)^2 \right).$$

Based on Lemma F.3, we can conclude that $\left\| \hat{\theta}_i \right\|_\infty^2 \cdot \mathbb{E}_{Y \sim p}\left( p_i(Y^\top \hat{\theta}_i)^2 \right)$ is invariant to the scale of $\hat{\theta}_i$, and hence based on Assumption 3.3, we know this term is actually bounded by some constant $C$. Therefore, it holds that

$$\mathbb{E}\left( (K-1)^2 \frac{p_i(X^\top \hat{\theta}_i)^2}{F_i(X^\top \hat{\theta}_i)^2} \cdot \left\| \hat{\theta}_i \right\|_\infty^2 \right) \leq K^3 C.$$

Consequently, for $X \sim p_i(\cdot)$, we have that

$$\left\| \mathbb{E}(S^{p_i}(X)^2) \right\|_\infty \leq 2 \left\| \mathbb{E}(S^p(X))^2 \right\|_\infty + 2\mathbb{E}\left( (K-1)^2 \frac{p_i(X^\top \hat{\theta}_i)^2}{F_i(X^\top \hat{\theta}_i)^2} \cdot \left\| \hat{\theta}_i \right\|_\infty^2 \right)$$

$$\leq KM + K^3 C := M_0,$$

Furthermore, at epoch $i > 1$, based on Theorem 3.1 we have that

$$\left\|\hat{\theta}_i - \theta_*\right\|_1 \lesssim d\sqrt{\frac{M_0 \log(2d/\delta)}{\kappa_{i-1}}},$$

with probability at least $1 - \delta$. Taking the union we have that for all epoch $i = 2, \ldots, H$, we have

$$\left\|\hat{\theta}_i - \theta_*\right\|_1 \lesssim d\sqrt{\frac{M_0 \log(2d\log_2(T)/\delta)}{\kappa_{i-1}}}$$

holds simultaneously with probability at least $1 - \delta$. Then at some time step $t$ in epoch $i$, $i > 1$, we have that

$$
\begin{aligned}
f(x_{t,*}^\top \theta_*) - f(x_t^\top \theta_*) &= f(x_{t,*}^\top \theta_*) - f\left(x_{t,*}^\top \frac{\hat{\theta}}{\mu^*}\right) + f\left(x_{t,*}^\top \frac{\hat{\theta}}{\mu^*}\right) - f(x_t^\top \theta_*) \\
&\leq f(x_{t,*}^\top \theta_*) - f\left(x_{t,*}^\top \frac{\hat{\theta}}{\mu^*}\right) + f\left(x_t^\top \frac{\hat{\theta}}{\mu^*}\right) - f(x_t^\top \theta_*) \\
&\leq \left| L_{f'} \cdot x_{t,*}^\top \left(\theta_* - \frac{\hat{\theta}}{\mu^*}\right) \right| + \left| L_{f'} \cdot x_t^\top \left(\theta_* - \frac{\hat{\theta}}{\mu^*}\right) \right| \\
&\leq L_{f'} \cdot \|x_{t,*}\|_\infty \left\|\theta_* - \frac{\hat{\theta}}{\mu^*}\right\|_1 + L_{f'} \cdot \|x_t\|_\infty \left\|\theta_* - \frac{\hat{\theta}}{\mu^*}\right\|_1 \\
&\leq \frac{L_{f'}}{\mu^*} \cdot \|x_{t,*}\|_\infty \left\|\mu^*\theta_* - \hat{\theta}\right\|_1 + \frac{L_{f'}}{\mu^*} \cdot \|x_t\|_\infty \left\|\mu^*\theta_* - \hat{\theta}\right\|_1 \\
&\lesssim d\sqrt{\frac{M_0 \log(2d\log_2(T)/\delta)}{\kappa_{i-1}}},
\end{aligned}
$$

Therefore, with probability at least $1 - \delta$,

$$
\begin{aligned}
R_T &\lesssim 2L_f\kappa_1 + \sum_{m=2}^H d\sqrt{\frac{M_0 \log(2d\log_2(T)/\delta)}{\kappa_{m-1}}} \cdot \kappa_m \\
&= 2L_fT_0 + \sum_{m=2}^H d\sqrt{M_0 \log(2d\log_2(T)/\delta)} 2\sqrt{\kappa_{m-1}} \\
&= 2L_fT_0 + 2d\sqrt{M_0 \log(2d\log_2(T)/\delta)} \cdot \sqrt{T_0} \cdot \frac{(\sqrt{2})^{H-1} - 1}{\sqrt{2} - 1} \\
&\overset{(i)}{\leq} 2L_fT_0 + (2+\sqrt{2})d\sqrt{M_0 \log(2d\log_2(T)/\delta)} \cdot \sqrt{(2^H - 2)T_0} \\
&\leq 2L_fT_0 + (2+\sqrt{2})d\sqrt{(MK + CK^3)\log(2d\log_2(T)/\delta)} \cdot \sqrt{T} \\
&= O\left(T_0 + d\sqrt{CK^3 \log(d\log_2(T)/\delta)T}\right),
\end{aligned}
$$

where inequality (i) comes from Lemma F.1. Finally, based on the choice of $T_0$, we have that

$$R_T = O\left(dK^{\frac{3}{2}}\sqrt{C \cdot T \cdot \log(d\log_2(T)/\delta)}\right) = \tilde{O}\left(dK^{\frac{3}{2}}\sqrt{T}\right).$$

On the other hand, if the time horizon $T$ does not match the end of an epoch, i.e. we have some $H > 0$ such that $e_H < T < e_{H+1}$. Since we have that $e_i = (2^i - 1)T_0, i \geq 0$, which indicates that $e_i > 2e_{i+1}, i \geq 0$. Therefore, it holds that

$$e_{H+1} < 2e_H < 2T.$$

Therefore, it holds that

$$
\begin{aligned}
R_T \leq R_{e_{H+1}} &= O\left(dK^{\frac{3}{2}}\sqrt{e_{H+1} \cdot \log(d\log_2(e_{H+1})/\delta)}\right) \leq O\left(dK^{\frac{3}{2}}\sqrt{2T \cdot \log(d\log_2(2T)/\delta)}\right) \\
&= O\left(dK^{\frac{3}{2}}\sqrt{T \cdot \log(d\log_2(T)/\delta)}\right).
\end{aligned}
$$

And this concludes our proof. □

Note that the final regret bound of order $\tilde{O}_T(\sqrt{T})$ holds as long as $C$ does not scale with $T$, which naturally occurs as $T$ grows large. Consequently, Assumption 3.3 is not required to establish this nearly-optimal regret bound, and the final regret bound with $C$ becomes

$$R_T \leq O\left(dK^{\frac{3}{2}}\sqrt{C \cdot T \cdot \log(d \log_2 (T)/\delta)}\right) = \tilde{O}\left(dK^{\frac{3}{2}}\sqrt{C \cdot T}\right).$$

However, as we emphasize in the main paper, assuming that $C$ is of constant scale is a very mild requirement, and in fact is satisfied by most practical distributions. For simplicity, we adopt this assumption here, which we consider both natural and reasonable.

## G   PROOF OF THEOREM 3.7

The proof of Theorem 3.7 builds upon our previous proof of Theorem 3.1 and also relies on Lemmas D.1–D.3, which we introduced in Appendix D. Notably, Lemma D.3 naturally extends to the high-dimensional single index model. However, to eliminate the dependency on $d$ in the final estimation bound, we incorporate an $l_1$-norm penalization in the loss function (Eqn. (1)) and introduce a novel technical approach.

*Proof.* Since $\hat{\theta}$ minimizes the loss function in Eqn. (1), based on the property of sub-gradient it holds that

$$\nabla L(\hat{\theta}) + \lambda\epsilon = 0, \quad \text{where } \epsilon \in \partial \left\|\hat{\theta}\right\|_1.$$

Therefore, based on a widely known result (Boyd & Vandenberghe, 2004) on the $l_1$ norm, we have for any $j \in [d]$,

$$\epsilon_j \begin{cases} = \text{sign}(\hat{\theta}_j), & \text{if } j \in \text{supp}(\hat{\theta}), \\ \in [-1,1], & \text{if } j \notin \text{supp}(\hat{\theta}), \end{cases}$$

where we denote $\text{supp}(\hat{\theta})$ as the support of $\hat{\theta}$, i.e. $\text{supp}(\hat{\theta}) = \{j \in [d] : \hat{\theta}_j \neq 0\}$. For some set $V \subseteq [d]$ and vector $v \in \mathbb{R}^d$, we use $v_V$ to denote a $d$-dimensional vector whose $j$th entry is equal to $v_j$ if $j \in V$ and 0 otherwise. For simplicity, we denote $U := \text{supp}(\theta_*)$ and $\beta = \hat{\theta} - \mu_*\theta_*$ in the following proof, and we know that $\epsilon = \epsilon_U + \epsilon_{U^c}$ and the cardinality of $U$ is $s$. Since $L(\theta)$ is a quadratic function for $\theta \in \mathbb{R}^d$, we have that

$$2\|\beta\|_2^2 = \left(\nabla L(\hat{\theta}) - \nabla L(\mu_*\theta_*)\right)^\top \beta = (-\lambda\epsilon - \nabla L(\mu_*\theta_*))^\top \beta$$
$$\leq (-\lambda\epsilon_U - \lambda\epsilon_{U^c})^\top \beta + \|\nabla L(\mu_*\theta_*)\|_\infty \|\beta\|_1. \quad (9)$$

Due to the fact that $\|\epsilon_U\|_\infty \leq 1$, it holds that

$$-\lambda\epsilon_U^\top\beta = -\lambda\epsilon_U^\top\beta_U \leq \lambda\|\beta_U\|_1.$$

And based on the definitions above, we have that

$$-\lambda\epsilon_{U^c}^\top\beta = -\lambda\epsilon_{U^c}^\top(\hat{\theta} - \mu_*\theta_*) = -\lambda\epsilon_{U^c}^\top\hat{\theta} = -\lambda\left\|\hat{\theta}_{U^c}\right\|_1 = -\lambda\|\beta_{U^c}\|_1.$$

By combining the above results with Eqn. (9), it holds that

$$2\|\beta\|_2^2 \leq -\lambda\|\beta_{U^c}\|_1 + \lambda\|\beta_U\|_1 + \|\nabla L(\mu_*\theta_*)\|_\infty \|\beta\|_1.$$

If we have $\lambda \geq 2\|\nabla L(\mu_*\theta_*)\|_\infty$, then it holds that

$$2\|\beta\|_2^2 \leq -\lambda\|\beta_{U^c}\|_1 + \lambda\|\beta_U\|_1 + \frac{\lambda}{2}\|\beta\|_1$$
$$\leq -\lambda\|\beta_{U^c}\|_1 + \lambda\|\beta_U\|_1 + \frac{\lambda}{2}(\|\beta_U\|_1 + \|\beta_{U^c}\|_1)$$
$$\leq -\frac{\lambda}{2}\|\beta_{U^c}\|_1 + \frac{3\lambda}{2}\|\beta_U\|_1 \quad (10)$$
$$\leq \frac{3\lambda}{2}\|\beta_U\|_1 \overset{(i)}{\leq} \frac{3\lambda}{2}\|\beta_U\|_2 \cdot \sqrt{s} \leq \frac{3\lambda}{2}\sqrt{s}\|\beta\|_2,$$

and the inequality (i) is due to Cauchy-Schwarz inequality and $|U| = s$. Therefore, we have

$$\|\beta\|_2 \leq \frac{3\lambda}{4}\sqrt{s} \qquad (11)$$

Moreover, due to the Eqn. (10), we have that

$$-\frac{\lambda}{2}\|\beta_{U^c}\|_1 + \frac{3\lambda}{2}\|\beta_U\|_1 \geq 0,$$

which indicates that $\|\beta_{U^c}\|_1 \leq \|\beta_U\|_1$. Therefore, with Eqn. (11), it holds that

$$\|\beta\|_1 = \|\beta_{U^c}\|_1 + \|\beta_U\|_1 \leq 4\|\beta_U\|_1 \leq 4\|\beta_U\|_2 \cdot \sqrt{s} \leq 4\sqrt{s}\|\beta\|_2 \leq 3\lambda s. \qquad (12)$$

According to Lemma D.3, by taking

$$\tau = \sqrt{\frac{3n(\sigma^2 + S_f^2)M}{\log(2d/\delta)}},$$

we have that with probability at least $1 - \delta$,

$$\|\nabla L(\mu_*\theta_*)\|_\infty \leq \left(\frac{4\sqrt{3}}{3} + 2\sqrt{2}\right)\sqrt{\frac{M(\sigma^2 + S_f^2)\log(2d/\delta)}{n}}.$$

Therefore, by setting the same value for $\tau$ and taking

$$\lambda = 11 \cdot \sqrt{\frac{M(\sigma^2 + L_f^2)\log(2d/\delta)}{n}} \geq \left(\frac{8\sqrt{3}}{3} + 4\sqrt{2}\right) \cdot \sqrt{\frac{M(\sigma^2 + L_f^2)\log(2d/\delta)}{n}},$$

then with probability at least $1 - \delta$ we have $\lambda \geq 2\|\nabla L(\mu_*\theta_*)\|_\infty$. Finally, based on Eqn. (11) and (12), we can deduce that

$$\left\|\hat{\theta} - \mu_*\theta_*\right\|_2 \leq \frac{3}{4}\lambda\sqrt{s} = \tilde{O}\left(\sqrt{\frac{s}{n}}\right), \quad \left\|\hat{\theta} - \mu_*\theta_*\right\|_1 \leq 3\lambda s = \tilde{O}\left(\frac{s}{\sqrt{n}}\right).$$

$\square$

## H    PROOF OF COROLLARY 3.8

*Proof.* Corollary 3.8 consists two parts, where the first part is the regret bound of Algorithm 1 and the second part is the regret bound of Algorithm 2. We will omit the detailed proof here since they are a simple combination of our deduced results above. Specifically, the proof of the first part is a combination of results in Theorem 3.7 and the proof of Themrem 3.2, and the proof of the second part is a combination of the results in Theorem 3.7 and the proof procedure of Theorem 3.5. And compared with the estimation error deduced under the low-dimensional case under Theorem 3.1, the estimation bound under the high-dimensional case under Theorem 3.7 depends on the sparsity index $s$ instead of the dimension $d$ in terms of the non-logarithmic factors. Therefore, the final regret bound will also be adjusted by replacing $d$ by $s$ in the non-logarithmic terms. In other words, if we ignore the logarithmic factors in the final regret bound, then the regret bounds will simply replace $d$ by $s$. And all the proof procedure can be identically reused under the high-dimensional sparse case. Therefore, we will omit the detailed proof due to redundancy.

For STOR, the value of $s$ is used only to set the exploration phase length $T_1$. Even if we remove this dependence and set $T_1$ free of $s$, we can still prove a regret bound of order $\tilde{O}(sT^{2/3})$, where the exponent of $s$ changes from $2/3$ to $1$. $\square$

## I    EXPLANATIONS ON ASSUMPTION 2.3

### I.1    EQUIVALENCE BETWEEN $\|x_{t,i}\|_\infty \leq L$ SETTING AND $\|x_{t,i}\|_2 \leq L$ SETTING

As we mentioned in Section 2 under Assumption 2.3, our main results hold regardless of the assumption on types of the norms. Specifically, we have the following two types of assumptions:

- Condition I: $\|\theta_*\|_2 = 1$, $\|x_{t,i}\|_2 \le L$, $\forall t \in [T], i \in [K]$ for some $L > 0$.

- Condition II: $\|\theta_*\|_1 = 1$, $\|x_{t,i}\|_\infty \le L$, $\forall t \in [T], i \in [K]$ for some $L > 0$.

Note we set the $l_2$ ($l_1$)-norm as some constant for $\theta_*$ due to the identifiability of the single index model. The former one is more commonly used in the contextual linear bandit literature (Abbasi-Yadkori et al., 2011; Filippi et al., 2010), while we use the latter one in this work. As we explained, both of these two assumptions are mainly used to ensure the inner product $x_{t,i}^\top \theta_*$ can be bounded by $L$ based on Holder's inequality, and hence they are identical. And we use the latter one merely to keep consistent with the sparse linear bandit case in Section 3.4 where $l_1$ norm is commonly assumed to be bounded. Furthermore, we will claim here our main Theorem 3.2, Theorem 3.5 and Theorem L.2 still hold: First, the estimation bounds presented in Theorem 3.1 and Theorem L.1 remain valid irrespective of the assumptions. In particular, the proof of Theorem 3.1 does not depend on the magnitude of $\theta_*$, ensuring that its bounds hold under both Condition I and Condition II. Moreover, the final results of Theorem L.1 differ at most by a constant factor under these conditions, indicating that the bounds remain valid as well. Second, for the regret bounds established in this work, such as Theorem 3.2, Theorem 3.5, and Theorem L.2, we show that the same conclusions hold. As an illustrative example, we examine the proof of Theorem 3.2 in Appendix E.2, as all regret analyses follow a similar way on leveraging the estimation bound of the parameter. In its proof, we use the estimation bound from 3.1 with Holder's inequality in Eqn. (7). Under condition II in our work, we have that

$$\|x_{t,i}\|_\infty \cdot \left\|\theta_* - \frac{\hat{\theta}}{\mu}\right\|_1 \le \frac{S}{\mu} \cdot \left(\frac{2\sqrt{3}}{3} + \sqrt{2}\right) d \cdot \sqrt{\frac{M(\sigma^2 + L_f^2)\log(2d/\delta)}{n}},$$

where $M$ can be considered as a constant since each entry of $x_{t,i}$ is in a constant scale. Specifically, for a Gaussian random variable $\mathcal{N}(\mu_0, \sigma_0^2)$, we can calculate that $M = 1/\sigma_0^2$. On the other hand, if we have Condition I, then we should rewrite the above equation with

$$\|x_{t,i}\|_2 \cdot \left\|\theta_* - \frac{\hat{\theta}}{\mu}\right\|_2 \le \frac{S}{\mu} \cdot \left(\frac{2\sqrt{3}}{3} + \sqrt{2}\right) \cdot \sqrt{\frac{dM(\sigma^2 + L_f^2)\log(2d/\delta)}{n}}.$$

Although the bound seems to improve by a multiplier of $\sqrt{d}$ explicitly, but here the value of $M$ may not be in a constant scale. Specifically, since we have $\|x_{t,i}\|_2$ is bounded by some constant $L$, then each entry will be bounded by the order $1/\sqrt{d}$ in magnitude on average. Assume the entry follows $\mathcal{N}(\mu_0, \sigma_0^2)$ with $\sigma_0^2 = \Omega(1/\sqrt{d})$, then it holds that $M = \Omega(d)$. Therefore, with $M = \Omega(d)$, we actually obtain the same bound of order $\tilde{O}(d\sqrt{\log(d/\delta)/n})$ under Condition II. This identical argument can be used in the proof of all other Theorems with regret bounds, and hence we can conclude that the regret bounds are the same under Condition I and Condition II.

## I.2 DETAILS OF $L = \tilde{O}(1)$

As we mentioned in the paragraph right after Assumption 2.3, we can actually let $L$ be in a constant scale up to some logarithmic terms, i.e. $L = \tilde{O}(1)$, and our main theorems in this work will still hold. This enables us to work with a wider range of distributions such as any sub-Gaussian or sub-Exponential distributions, and we will explain why this holds. Since sub-Gaussian distribution is a specific case of sub-Exponential, we will use the following Lemma I.1 to illustrate that $L = \tilde{O}(1)$ with arbitrary high probability under sub-exponential $\mathcal{D}$, and hence it will not contribute to the final regret bound after ignoring all logarithmic terms. We also assume zero mean for simplicity in the following lemma since the final bound will only differ by a constant mean shift.

**Lemma I.1.** *Let $X_1, \ldots, X_n$ be i.i.d. zero-mean sub-Exponential random variables. Specifcially, suppose there exist positive constants $\alpha$ and $\nu$ such that*

$$\mathbb{E}\left[e^{\lambda X_1}\right] \le \exp\left(\frac{\nu^2 \lambda^2}{2}\right) \quad \text{for all } |\lambda| < \frac{1}{\alpha}.$$

*Then there is a constant $c = c(\alpha, \nu)$ for which, for every $0 < \delta < 1$,*

$$\mathbb{P}\left(\max_{1 \le i \le n} |X_i| \ge c \log\left(\frac{n}{\delta}\right)\right) \le \delta.$$

*Proof.* By a standard sub-exponential tail estimate (Vershynin, 2018), there exist constants $C_1 > 0$ depending only on $\alpha$ and $\nu$ such that for all $t \geq 0$,

$$\mathbb{P}\big(|X_1| \geq t\big) \leq 2 \exp\Big(-\tfrac{t}{C_1}\Big).$$

A union bound then shows

$$\mathbb{P}\Big(\max_{1 \leq i \leq n} |X_i| \geq t\Big) \leq 2n \exp\Big(-\tfrac{t}{C_1}\Big).$$

Choosing $t = c \log\big(\tfrac{n}{\delta}\big)$ with a sufficiently large $c$ absorbs $n$ and $C_1$ inside the exponential, making the above probability at most $\delta$. It holds that

$$\mathbb{P}\Big(\max_{1 \leq i \leq n} |X_i| \geq c \log\big(\tfrac{n}{\delta}\big)\Big) \leq \delta,$$

as claimed. $\qquad\square$

This result indicates that with assuming $\mathcal{D}$ is any sub-Exponential distribution, we will get the same final regret bounds up to logarithmic factors with high probability.

## J   EXPLANATIONS OF REMARK 3.4

In this section, we show that Assumption 3.3 is not restrictive and holds for many common distributions. In particular, to support Remark 3.4, we demonstrate that if the random vector $X \in \mathbb{R}^d$ is drawn from some multivariate normal distribution, then Assumption 3.3 is satisfied. Firstly, based on Lemma F.3, we know that the value of $\mathbb{E}\big(p_v(X^\top v)^2\big) \cdot \|v\|_\infty^2$ is fixed regardless of the scale of $v$. In other words, to prove Assumption 3.3, it is equivalent to show that for any $v$ with $\|v\|_2 = 1$, we have

$$\mathbb{E}\Big[p_v(X^\top v)^2\Big] \cdot \|v\|_\infty^2 = \|v\|_\infty^2 \cdot \int p_v(x^\top v)^2 p(x)\, dx \leq C.$$

**Lemma J.1.** *Let $X \in \mathbb{R}^d$ be a random vector sampled from some d-dimensioanl multivariate normal distribution with expected value $\mu_X \in \mathbb{R}^d$ and covariance matrix $\Sigma_X \in \mathbb{R}^{d \times d}$. And $v \in \mathbb{R}^d$ is an arbitrary vector with $\|v\|_2 > 0$. Then we have Assumption 3.3 hold:*

$$\mathbb{E}\big(p_v(X^\top v)^2\big) \cdot \|v\|_\infty^2 \leq C \quad \text{for some constant } C > 0,$$

*where $p_v(\cdot)$ is the density of $X^\top v$.*

*Proof.* Since $X \sim \mathcal{N}(\mu_X, \Sigma_X)$, the univariate random variable $Y := X^\top v$ is itself normally distributed: $Y \sim \mathcal{N}(\mu_X^\top v,\ v^\top \Sigma_X v)$. Denote $\sigma^2 := v^\top \Sigma_X v > 0$. Then the density of $Y$ is

$$p_v(t) = \frac{1}{\sqrt{2\pi}\,\sigma} \exp\Big(-\tfrac{(t-\mu_X^\top v)^2}{2\,\sigma^2}\Big).$$

We must show that $\mathbb{E}[p_v(Y)^2]$ is finite. Since for every real $t$ we have

$$p_v(t) \leq \frac{1}{\sqrt{2\pi}\,\sigma} \implies p_v(t)^2 \leq \frac{1}{2\pi\,\sigma^2}.$$

Hence

$$\big|p_v(Y)^2\big| \leq \frac{1}{2\pi\,\sigma^2}.$$

Taking expectations on both sides yields

$$\mathbb{E}\big[p_v(Y)^2\big] \leq \frac{1}{2\pi\,\sigma^2} < \infty.$$

Since we have that $\|v\|_\infty \leq \|v\|_2 = 1$, we can take $C = \frac{1}{2\pi\,\sigma^2}$, proving the claim. $\qquad\square$

Based on the above proof, we can further conclude that Assumption 3.3 is satisfied once the density function of $X^\top v$ is a bounded function, i.e. $p_v(\cdot) \leq C$ for some $C > 0$, then we naturally have that $\mathbb{E}((p_v(X^\top v)^2)) \leq C^2$. Note this finiteness holds for most commonly-used distributions, such as Gamma distribution $\Gamma(k, \theta)$ with $k > 1$, Laplace distribution, uniform distribution, etc.

Finally, If each entry of $X$ is i.i.d. sub-Gaussian and $v$ is a unit vector, then $Y = X^\top v$ is itself sub-Gaussian with the same tail parameter. In particular, this guarantees tail decay of the form

$$\mathbb{P}(|Y| > r) \leq \exp(-c\, r^2) \text{ for some } c > 0.$$

It is noteworthy that the square density $p_v(y)^2$ inherits rapid decay (dominated by $e^{-2cy^2}$) at infinity ensuring its integrability. Near $y = 0$, standard Sub-Gaussian distributions also typically avoid pathological density spikes, so $\left[p_v(y)\right]^2$ remains well controlled. Consequently, one expects $\mathbb{E}\left[p_v(X^\top v)^2\right]$ to be finite for a broad class of sub-Gaussian families. While exotic sub-Gaussian distributions with unbounded and non-integrable squared densities could theoretically violate this, such cases are very atypical in applied settings.

## K $\quad K$ DEPENDENCY OF THEOREM 3.5 IN REMARK 3.6

Note that we remove the term $F_i(x^\top \hat\theta_i)^{K-1}$ directly in the above proof in Eqn. 8 since it must be less than 1. However, this is a very conservative step since $F_i(x^\top \hat\theta_i)^{K-1}$ is exponentially small and quickly converges to 0 under a large value of $K$. Therefore, our proof holds under worst cases, whereas the dependence on $K$ could be significantly improved under lots of common settings. For example, if $x_{t,i}$ is sampled from some $d$-dimensional normal distribution, then we can prove that

$$\mathbb{E}\left[K \cdot p(X)^2 \cdot F(X)^{K-3}\right] = \Theta\left(\frac{\log K}{K^2}\right) \quad \text{as } K \to \infty,$$

and this fact indicates that $M_0$ is in the order of $\log(K)$ when $K$ is large. Therefore, the final regret bound of Algorithm 2 exhibits an order of $O(\sqrt{M_0}) = O(\sqrt{\log(K)})$, as established in our proof in Appendix F.2 by setting $T_0$ to be any value such that $T_0 \leq d\sqrt{T \log(K) \log(2d \log_2(T)/\delta)}$.

In Lemma K.1, we provide a proof sketch under the standard normal assumption for simplicity, while noting that the asymptotic order remains unchanged for any normal distribution, regardless of its specific mean or variance.

**Lemma K.1.** *Let $X \sim \mathcal{N}(0,1)$ be a standard normal random variable with probability density function $p(x) = \frac{1}{\sqrt{2\pi}}e^{-x^2/2}$ and cumulative distribution function (CDF) $F(x) = \Phi(x)$. For $K > 3$, the expectation*

$$\mathbb{E}\left[K \cdot p(X)^2 \cdot F(X)^{K-3}\right]$$

*satisfies the asymptotic relation:*

$$\mathbb{E}\left[K \cdot p(X)^2 \cdot F(X)^{K-3}\right] = \Theta\left(\frac{\log K}{K^2}\right) \quad \text{as } K \to \infty.$$

*Proof.* The expectation can be expressed as:

$$I_K = K \int_{-\infty}^{\infty} p(x)^3 \Phi(x)^{K-3} \, dx.$$

Under the substitution $u = \Phi(x)$, we transform the integral as:

$$I_K = K \int_0^1 p\left(\Phi^{-1}(u)\right)^2 u^{K-3} \, du.$$

To analyze the asymptotic behavior as $K \to \infty$, we first note that the dominant contribution to the integral arises from values of $u$ near 1. For $u \in [0, 1-\delta]$ with fixed $\delta \in (0,1)$, the term $u^{K-3}$ decays exponentially as $(1-\delta)^{K-3}$, and the squared density $p\left(\Phi^{-1}(u)\right)^2$ is bounded by $\frac{1}{2\pi}$. Thus, the integral over $[0, 1-\delta]$ satisfies:

$$K \int_0^{1-\delta} p\left(\Phi^{-1}(u)\right)^2 u^{K-3} \, du = O\left(K(1-\delta)^K\right),$$

which is exponentially negligible compared to any polynomial decay as $K$ goes to infinity.

For the dominant region $u \in [1 - \delta, 1]$, let $u = 1 - t$ with $t \in [0, \delta]$. The inverse CDF satisfies the asymptotic expansion as $t \to 0^+$:

$$\Phi^{-1}(1 - t) = \sqrt{2 \log(1/t)} - \frac{\log(4\pi \log(1/t))}{2\sqrt{2 \log(1/t)}} + O\left(\frac{1}{\log(1/t)}\right).$$

The density at this point is:

$$p\left(\Phi^{-1}(1 - t)\right) = \frac{1}{\sqrt{2\pi}} e^{-\frac{1}{2}(2 \log(1/t) - \log(4\pi \log(1/t)) + \mathcal{O}(1))} = \frac{t}{\sqrt{2\pi}} \sqrt{4\pi \log(1/t)} \left(1 + O\left(\frac{1}{\log(1/t)}\right)\right),$$

which simplifies to:

$$p\left(\Phi^{-1}(1 - t)\right) = t\sqrt{2 \log(1/t)} \left(1 + O\left(\frac{1}{\log(1/t)}\right)\right).$$

Squaring this gives:

$$p\left(\Phi^{-1}(1 - t)\right)^2 = 2t^2 \log(1/t) \left(1 + O\left(\frac{1}{\log(1/t)}\right)\right).$$

Substituting $(1 - t)^{K-3} \approx e^{-(K-3)t}$ and extending the upper limit to $\infty$ (with exponentially small error), we have:

$$I_K \sim 2K \int_0^\infty t^2 \log(1/t) e^{-(K-3)t} \, dt,$$

where $\sim$ denotes asymptotic equivalence (exact leading term precision including constants). Using the substitution $s = (K - 3)t$, the integral becomes:

$$I_K = \frac{2K}{(K-3)^3} \int_0^\infty s^2 \log\left(\frac{K-3}{s}\right) e^{-s} \, ds.$$

Expanding $\log\left(\frac{K-3}{s}\right) = \log K - \log s + \mathcal{O}(1/K)$, the dominant term is:

$$\frac{2K \log K}{(K-3)^3} \int_0^\infty s^2 e^{-s} \, ds = \frac{4K \log K}{(K-3)^3}.$$

Since $\int_0^\infty s^2 e^{-s} ds = 2$, we obtain:

$$I_K \sim \frac{4 \log K}{K^2} = \Theta\left(\frac{\log K}{K^2}\right) \quad \text{as } K \to \infty.$$

Combining the negligible contribution from $[0, 1 - \delta]$ and the dominant term from $[1 - \delta, 1]$, we conclude:

$$\mathbb{E}\left[K \cdot p(X)^2 \cdot F(X)^{K-3}\right] = \Theta\left(\frac{\log K}{K^2}\right) \quad \text{as } K \to \infty.$$

$\square$

## L    DETAILS AND THEORY OF GSTOR

### L.1    DETAILS OF GSTOR

In this part, we present the pseudocode of our proposed GSTOR in Algorithm 3. As mentioned in Section 3.5, our GSTOR adopts a double exploration-then-commit strategy. Specifically, our algorithm uses $T_1$ random samples to estimate the parameter $\theta_*$ and obtain the estimator $\hat{\theta}$ (line 4). Afterward, we normalize the estimator and obtain $\hat{\theta}_0$ (line 5). Furthermore, we choose another independent set of $T_1$ samples. We leverage $\hat{\theta}_0$ and the kernel regression to approximate the unknown function, and obtain the function predictor $\hat{f}$ (line 9). For the remaining rounds, we select the best arm greedily based on the estimates (line 11).

---

**Algorithm 3** General Stein's Oracle Single Index Bandit (GSTOR)

---

**Input:** $T$, the probability rate $\delta$, parameters $T_1, \lambda, \tau, W, h$
1: **for** $t = 1$ **to** $T_1$ **do**
2:      Pull an arm $x_t \in \mathcal{X}_t$ uniformly randomly and observe the stochastic reward $y_t$.
3: **end for**
4: Obtain the estimator $\hat{\theta}$ with $\{x_i, y_i\}_{i=1}^{T_1}$ based on Eqn. (1).
5: Get the normalized estimator $\hat{\theta}_0$ as $\hat{\theta}_0 = \hat{\theta}/\|\Sigma_X^{1/2}\hat{\theta}\|_1$.
6: **for** $t = T_1 + 1$ **to** $2T_1$ **do**
7:      Pull an arm $x_t \in \mathcal{X}_t$ uniformly randomly and observe the stochastic reward $y_t$.
8: **end for**
9: Approximate the unknown reward function $\hat{f}(\cdot)$ with $\hat{\theta}_0$ based on Eqn. (2).
10: **for** $t = 2T_1 + 1$ **to** $T$ **do**
11:      Choose the arm $x_t = \arg\max_{x \in \mathcal{X}_t} \hat{f}(x^\top \hat{\theta}_0)$, break ties arbitrary.
12: **end for**

---

### L.2 USEFUL LEMMAS

We first prove that our normalized estimator $\hat{\theta}_0$ holds a similar error bound as the rate in Theorem 3.1.

**Lemma L.1.** (Bound of SIM) *Following the same notation of Theorem 3.1. For any single index model defined in with samples $x_1, \ldots, x_n$ drawn from some distribution $\mathcal{D}$ with covariance matrix $\Sigma_X$. Under Assumption 2.2, 2.3 and the identifiability assumption that $\|\Sigma_X^{1/2}\theta_*\|_1 = 1, \mu^* > 0$, by solving the optimization problem in Eqn. (1) with $\tau = \sqrt{3(\sigma^2 + L_f^2)Mn/\log(2d/\delta)}$ and $\lambda = 0$, with the probability at least $(1 - 2\delta)$ it holds that :*

$$\left\| \frac{\hat{\theta}}{\|\Sigma_X^{1/2}\hat{\theta}\|_1} - \theta_* \right\|_1 = \tilde{O}\left(\frac{d}{\sqrt{n}}\right).$$

*Note this result holds under general distributions $\mathcal{D}$ without the need for a Gaussian design.*

*Proof.* We start by controlling the difference between $\|\Sigma_X^{1/2}\hat{\theta}\|_1$ and $\mu_*$. When the dimension $d$ is low, the eigenvalues of $\Sigma_X$ are bounded, and we denote $C_{\min}$ and $C_{\max}$ are two positive constants such that $C_{\min} \leq \lambda_{\min}(\Sigma_X) \leq \lambda_{\max}(\Sigma_X) \leq C_{\max}$. Notably, for sufficiently large $n$, since $\|\Sigma_X^{1/2}\theta_*\|_2 = 1$, we have

$$
\begin{aligned}
\|\Sigma_X^{1/2}\hat{\theta}\|_1 &= \|\Sigma_X^{1/2}\hat{\theta} - \Sigma_X^{1/2}\mu_*\theta_* + \Sigma_X^{1/2}\mu_*\theta_*\|_1 \\
&\geq \mu_* - \|\Sigma_X^{1/2}(\hat{\theta} - \mu_*\theta_*)\|_1 \\
&\geq \mu_* - \sqrt{d}\|\Sigma_X^{1/2}(\hat{\theta} - \mu_*\theta_*)\|_2 \\
&\geq \mu_* - \sqrt{d}\sqrt{C_{\max}}\|\hat{\theta} - \mu_*\theta_*\|_2 \\
&\geq \mu_* - \sqrt{d}\sqrt{C_{\max}}\|\hat{\theta} - \mu_*\theta_*\|_1 \\
&\geq \mu_* - \tilde{O}\left(\frac{d^{\frac{3}{2}}}{\sqrt{n}}\right) \geq \frac{\mu_*}{2},
\end{aligned}
\tag{13}
$$

holds with probability at least $1 - \delta$, where the triangle inequality gives the first inequality, the second inequality comes from Cauchy-Schwarz, the third inequality is because of $\lambda_{\max}(\Sigma_X) \leq C_{\max}$. The last two inequalities holds with the same probability as indicated by Theorem 3.1. Using the similar reason with the triangle inequlity for the other direction, we have

$$\|\Sigma_X^{1/2}\hat{\theta}\|_1 \leq \mu_* + \tilde{O}\left(\frac{d^{\frac{3}{2}}}{\sqrt{n}}\right) \leq \frac{3\mu_*}{2}, \tag{14}$$

hold with probability at least $1 - \delta$. Combining Eqn. (13) and (14) gives us the result that

$$\left| \|\Sigma_X^{1/2}\hat{\theta}\|_1 - \mu_* \right| \leq \tilde{O}\left(\frac{d^{\frac{3}{2}}}{\sqrt{n}}\right) \leq \frac{\mu_*}{2}. \tag{15}$$

holds with probability at least $1 - 2\delta$. We proceed with

$$
\left\| \frac{\hat{\theta}}{\|\Sigma_X^{1/2}\hat{\theta}\|_1} - \theta_* \right\|_1 = \left\| \frac{\hat{\theta}}{\|\Sigma_X^{1/2}\hat{\theta}\|_1} - \frac{\|\Sigma_X^{1/2}\hat{\theta}\|_1 \theta_*}{\|\Sigma_X^{1/2}\hat{\theta}\|_1} \right\|_1
$$
$$
\leq \frac{\|\hat{\theta} - \mu_*\theta_*\|_1}{\|\Sigma_X^{1/2}\hat{\theta}\|_1} + \frac{|\mu_* - \|\Sigma_X^{1/2}\hat{\theta}\|_1| \|\theta_*\|_1}{\|\Sigma_X^{1/2}\hat{\theta}\|_1}
$$
$$
\leq \frac{\|\hat{\theta} - \mu_*\theta_*\|_1}{\|\Sigma_X^{1/2}\hat{\theta}\|_1} + \frac{|\mu_* - \|\Sigma_X^{1/2}\hat{\theta}\|_1| \|\theta_*\|_1}{\|\Sigma_X^{1/2}\hat{\theta}\|_1}
$$
$$
\leq \frac{\|\hat{\theta} - \mu_*\theta_*\|_1}{\|\Sigma_X^{1/2}\hat{\theta}\|_1} + \frac{|\mu_* - \|\Sigma_X^{1/2}\hat{\theta}\|_1|\sqrt{d}}{C_{\min}\|\Sigma_X^{1/2}\hat{\theta}\|_1}
$$
$$
\leq \tilde{O}\Big(\frac{d}{\sqrt{n}}\Big)\frac{2}{\mu_*} + \tilde{O}\Big(\frac{d^{\frac{3}{2}}}{\sqrt{n}}\Big)\frac{2\sqrt{d}}{C_{\min}\mu_*} = \tilde{O}\Big(\frac{d^2}{\sqrt{n}}\Big),
$$

where the third inequality comes from the fact that $\|\theta_*\|_1 \leq \sqrt{d}\|\theta_*\|_2 \leq \sqrt{d} \cdot \frac{\|\Sigma_X^{1/2}\theta_*\|_2}{C_{\min}} \leq \sqrt{d}\frac{\|\Sigma_X^{1/2}\theta_*\|_1}{C_{\min}} = \frac{\sqrt{d}}{C_{\min}}$, and the last inequality is obtained after combining results of Theorem 3.1 and Eqn. (15). $\qquad\square$

## L.3 KERNEL REGRESSION ERROR BOUND

We slightly abuse the notations for easy presentation in this section. Specifically, we use $\{y_i, x_i\}_{i=1}^n$ to denote the samples for the kernel regression after the parameter estimation and normalization.

**Theorem L.2.** (Full Bound of SIM) *We use $n$ pairs of samples to obtain the normalized estimator $\hat{\theta}_0$ based on Eqn. (1). And we use another $n$ pairs of samples $\{y_i, x_i\}_{i=1}^n$ for the kernel regression with $\hat{\theta}_0$ based on Eqn. (2). Assume that $d^6 = O(n)$. With $M = 2\sqrt{\log(n)}$ and $h = n^{-1/3}$, $X \sim \mathcal{D} = N(\mu_X, \Sigma_X)$, we have*

$$
\mathbb{E}\big|\hat{f}(X^\top\hat{\theta}_0) - f(X^\top\theta_*)\big| = O\Big(\frac{d^{\frac{1}{2}}}{n^{\frac{1}{3}}}\Big),
$$

*where the expectation is taken with respect to $X$ and $\{x_i, y_i\}_{i=1}^n$.*

*Proof.* The core of our analysis relies on decomposing the prediction error into approximation and statistical components. We first bound the deviation between the predicted and true rewards by separating the prediction error into controllable terms using a novel indicator-based partitioning. We then apply Gaussian concentration inequalities to control tail events, and innovatively leverage Stein's method and a perturbation-style argument to precisely control the relationship between $Z$ and $Z^*$.

For notation simplicity, with $i \in [n]$, we denote

$$
Z_i^* = x_i^\top\theta_*, \ Z_i = x_i^\top\hat{\theta}_0, \ Z = x^\top\hat{\theta}_0, \ Z^* = x^\top\theta_*. \tag{16}
$$

Based on Theorem L.1 and the assumption that $d^6 = O(n)$, it holds that

$$
\|\hat{\theta}_0 - \theta_*\|_1 = O(n^{-1/3}). \tag{17}
$$

Notably, $\frac{Z - \mu_X^\top\hat{\theta}_0}{\|\Sigma_X^{1/2}\hat{\theta}_0\|_2} = \frac{(X - \mu_X)^\top\hat{\theta}_0}{\|\Sigma_X^{1/2}\hat{\theta}_0\|_2}$ is a random variable which follows the Gaussian distribution $N(0, 1)$ under our settings given in the assumption, then we get a tail bound for $Z$ as

$$
\mathbb{P}\left( \frac{|Z - \mu_X^\top\hat{\theta}_0|}{\|\Sigma_X^{1/2}\hat{\theta}_0\|_2} \geq t \right) \leq 2\exp\left(-t^2/2\right). \tag{18}
$$

In other words, by letting $t = 2\sqrt{\log n}$ in Eqn. (18), with probability $1 - 2/n^2$, we have $\frac{|Z - \mu_X^\top\hat{\theta}_0|}{\|\Sigma_X^{1/2}\hat{\theta}_0\|_2} \leq 2\sqrt{\log n}$. Since $\|\Sigma_X^{1/2}\hat{\theta}_0\|_2 \leq \|\Sigma_X^{1/2}\hat{\theta}_0\|_1$ holds almost surely, and $\|\Sigma_X^{1/2}\hat{\theta}_0\|_1 = 1$ in our assumption, we continue to have $|Z - \mu_X^\top\hat{\theta}_0| \leq 2\sqrt{\log n}$ with the same high probability.

Next, we separate our $\ell_1$ prediction error into two parts

$$
\begin{aligned}
&\mathbb{E}\left|\hat{f}(Z) - f(Z^*)\right| \\
&= \underbrace{\mathbb{E}\left[\left|\hat{f}(Z) - f(Z^*)\right| \cdot \mathbb{I}_{\{|Z - \mu_X^\top \hat{\theta}_0| \leq W\}}\right]}_{\text{(I)}} + \underbrace{\mathbb{E}\left[\left|\hat{f}(Z) - f(Z^*)\right| \cdot \mathbb{I}_{\{|Z - \mu_X^\top \hat{\theta}_0| > M\}}\right]}_{\text{(II)}}.
\end{aligned}
\tag{19}
$$

For term (II), by our definition of $\hat{f}(\cdot)$ given in Eqn. (2) (it is 0 when the event $\{|Z - \mu_X^\top \hat{\theta}_0| \leq W\}$ holds), and Assumption 2.3 , we have

$$
\mathbb{E}\left[\left|\hat{f}(Z) - f(Z^*)\right| \mathbb{I}_{\{|Z - \mu_X^\top \hat{\theta}_0| > M\}}\right] \leq L_f \mathbb{P}\left(\left|Z - \mu_X^\top \hat{\theta}_0\right| > M\right) \lesssim \frac{1}{n^2}.
$$

For term (I), we further separate it into (III) and (IV), which can be regarded as integrated mean (III) error and approximation error (IV) respectively. After defining the function $g(z) = \mathbb{E}(y|Z = z)$ for $z \in \mathbb{R}$, we continue to have

$$
\text{(I)} \leq \underbrace{\mathbb{E}\left[\left|\hat{f}(Z) - g(Z)\right| \mathbb{I}_{\{|Z - \mu_X^\top \hat{\theta}_0| \leq W\}}\right]}_{\text{(III)}} + \underbrace{\mathbb{E}\left[|g(Z) - f(Z^*)| \mathbb{I}_{\{|Z - \mu_X^\top \hat{\theta}_0| \leq W\}}\right]}_{\text{(IV)}}.
\tag{20}
$$

To handle the term (III), we define $g_0(Z)$ as

$$
g_0(Z) = \frac{\sum_{i=1}^n f(Z_i) K_h(Z - Z_i)}{\sum_{i=1}^n K_h(Z - Z_i)},
$$

and we proceed to control (III) by

$$
\text{(III)} \leq \underbrace{\mathbb{E}\left[\left|\hat{f}(Z) - g_0(Z)\right| \mathbb{I}_{\{|Z - \mu_X^\top \hat{\theta}_0| \leq W\}}\right]}_{\text{(V)}} + \underbrace{\mathbb{E}\left[|g_0(Z) - g(Z)| \mathbb{I}_{\{|Z - \mu_X^\top \hat{\theta}_0| \leq W\}}\right]}_{\text{(VI)}}.
\tag{21}
$$

Combining Eqn. (19), Eqn. (20) and Eqn. (21), we can see that $\ell_1$ prediction error can be bounded by the sum of (II), (IV), (V) and (VI). Next, we will bound these terms separately.

**Step 1** (bound (IV)): According to the data distribution assumption, $Z - \mu_X^\top \hat{\theta}_0$ follows $N(0, \hat{\theta}_0^\top \Sigma_X \hat{\theta}_0)$ and $Z^* - \mu_X^\top \theta_*$ follows $N(0, \theta_*^\top \Sigma_X \theta_*)$. Moreover, for two random variables following normal distributions, there exists the general result that

$$
\begin{aligned}
Z^* &= \mu_X^\top \theta_* + \|\Sigma_X^{1/2} \theta_*\|_2 \left( \frac{\hat{\theta}_0 \Sigma_X \theta_*}{\|\Sigma_X^{1/2} \hat{\theta}_0\|_2 \|\Sigma_X^{1/2} \theta_*\|_2} \frac{(X - \mu_X)^\top \hat{\theta}_0}{\|\Sigma_X^{1/2} \hat{\theta}_0\|_2} + \sqrt{1 - \frac{\hat{\theta}_0 \Sigma_X \theta_*}{\|\Sigma_X^{1/2} \hat{\theta}_0\|_2 \|\Sigma_X^{1/2} \theta_*\|_2}} \zeta \right) \\
&:= \mu_X^\top \theta_* + \|\Sigma_X^{1/2} \theta_*\|_2 \left( \cos\alpha \cdot \frac{Z - \mu_X^\top \hat{\theta}_0}{\|\Sigma_X^{1/2} \hat{\theta}_0\|_2} + \sin\alpha \cdot \zeta \right),
\end{aligned}
$$

where $\alpha$ is a real number within $[0, \pi/2]$, and $\zeta \sim N(0, 1)$ is independent of $Z$. In addition, notice the equality that

$$
\frac{\|a\|_2 \|b\|_2 - \langle a, b \rangle}{\|a\|_2 \|b\|_2} = \frac{-(\|a\|_2 - \|b\|_2)^2 + \|a - b\|_2^2}{2\|a\|_2 \|b\|_2}
$$

and $\left|\|a\|_2 - \|b\|_2\right| \leq \|a - b\|_2$ for any real vectors $a, b \in \mathbb{R}^d$. If we let $\Sigma_X^{1/2} \hat{\theta}_0$ and $\Sigma_X^{1/2} \theta_*$ play the roles of $a$ and $b$ respectively, then by Eqn. 17, it holds that

$$
\begin{aligned}
\sin^2 \alpha &\leq \frac{\|\Sigma_X^{1/2} \hat{\theta}_0 - \Sigma_X^{1/2} \theta_*\|_2^2}{\|\Sigma_X^{1/2} \hat{\theta}_0\|_2 \|\Sigma_X^{1/2} \theta_*\|_2} \leq \frac{C_{\max}^2 \|\hat{\theta}_0 - \theta_*\|_2^2}{\|\Sigma_X^{1/2} \hat{\theta}_0\|_2 \|\Sigma_X^{1/2} \theta_*\|_2} \\
&\leq \frac{dC_{\max}^2 \|\hat{\theta}_0 - \theta_*\|_1^2}{\|\Sigma_X^{1/2} \hat{\theta}_0\|_1 \|\Sigma_X^{1/2} \theta_*\|_1} = dC_{\max}^2 \|\hat{\theta}_0 - \theta_*\|_1^2 = O(dn^{-\frac{2}{3}}),
\end{aligned}
\tag{22}
$$

where the third inequality comes from Cauchy-Schwarz inequality, and the last order equality comes from Eqn. 17. Thus, the single index model can be equivalently written as

$$Y = f(Z^*) + \epsilon, \quad Z^* = \|\Sigma_X^{1/2}\theta_*\|_2 \left(\cos\alpha \cdot \frac{Z - \mu_X^\top\hat{\theta}_0}{\|\Sigma_X^{1/2}\hat{\theta}_0\|_2} + \sin\alpha \cdot \zeta\right) + \mu_X^\top\theta_*.$$

For simplicity, we denote $\tilde{Z}(z) = \cos\alpha \cdot \frac{\|\Sigma_X^{1/2}\theta_*\|_2}{\|\Sigma_X^{1/2}\hat{\theta}_0\|_2}(z - \mu_X^\top\hat{\theta}_0) + \mu_X^\top\theta_*$, $\tilde{Z} = \cos\alpha \cdot \frac{\|\Sigma_X^{1/2}\theta_*\|_2}{\|\Sigma_X^{1/2}\hat{\theta}_0\|_2}(Z - \mu_X^\top\hat{\theta}_0) + \mu_X^\top\theta_*$, and according to the definition of $g(z)$, we have

$$
\begin{aligned}
g(z) &= \mathbb{E}(y|Z = z) = \mathbb{E}\left[f(\tilde{Z}(z) + \|\Sigma_X^{1/2}\theta_*\|_2\sin\alpha \cdot \zeta)|Z = z\right] \\
&= \int_\mathbb{R} f(\tilde{Z}(z) + \|\Sigma_X^{1/2}\theta_*\|_2\sin\alpha \cdot \zeta) \cdot \phi(\zeta)d\zeta,
\end{aligned}
\tag{23}
$$

where $\phi$ is the density of the standard normal distribution. It is obvious that $g(z) \leq L_f$ by Assumption 2.3. To bound (IV), we first use $f(\tilde{Z})$ to approximate $f(Z^*)$ as well as $g(Z)$, then (IV) is bounded as

$$
\begin{aligned}
\mathbb{E}\left[|f(Z^*) - g(Z)|\,\mathbb{I}_{\{|Z-\mu_X^\top\hat{\theta}_0|\}\leq W\}}\right] \leq& \mathbb{E}\left[\left|f(Z^*) - f(\tilde{Z})\right|\mathbb{I}_{\{|Z-\mu_X^\top\hat{\theta}_0|\leq W\}}\right] \\
&+ \mathbb{E}\left[\left|f(\tilde{Z}) - g(Z)\right|\mathbb{I}_{\{|Z-\mu_X^\top\hat{\theta}_0|\leq W\}}\right].
\end{aligned}
\tag{24}
$$

For the first term on the right side of Eqn. (24), by mean value theorem

$$
\begin{aligned}
f(Z^*) - f(\tilde{Z}) &= f(\tilde{Z} + \|\Sigma_X^{1/2}\theta_*\|_2\sin\alpha \cdot \zeta) - f(\tilde{Z}) \\
&= f'(\tilde{Z} + t_1(Z,\zeta)\|\Sigma_X^{1/2}\theta_*\|_2\sin\alpha \cdot \zeta) \cdot \|\Sigma_X^{1/2}\theta_*\|_2\sin\alpha \cdot \zeta,
\end{aligned}
$$

where $t_1(Z,\zeta)$ is a constant between $[0,1]$ depending on $Z$ and $\zeta$. We continue to have

$$
\begin{aligned}
&\mathbb{E}\left[\left|f(Z^*) - f(\tilde{Z})\right|\mathbb{I}_{\{|Z-\mu_X^\top\hat{\theta}_0|\leq W\}}\right] \\
=& \|\Sigma_X^{1/2}\theta_*\|_2\sin\alpha \int_{|Z-\mu_X^\top\hat{\theta}_0|\leq W} \int_\mathbb{R} \left|f'(\tilde{Z} + t_1(Z,\zeta)\|\Sigma_X^{1/2}\theta_*\|_2\sin\alpha \cdot \zeta)\zeta\phi(\zeta)\right| d\zeta dF(Z) \\
\lesssim& L_{f'}\|\Sigma_X^{1/2}\theta_*\|_2\sin\alpha = O(\sqrt{d}n^{-\frac{1}{3}}),
\end{aligned}
\tag{25}
$$

where the inequality is due to Assumption 2.3 that $f'(\cdot)$ is bounded, and the last equality comes from Eqn. (22) and Eqn. 17 that $\|\Sigma_X^{1/2}\theta_*\|_2 \leq \|\Sigma_X^{1/2}\theta_*\|_1 = 1$. For the second term on the right side of Eqn. (24), by the definition of $g(z)$ in Eqn. (23), we have

$$
\begin{aligned}
f(\tilde{Z}) - g(Z) &= f(\tilde{Z}) - \int_\mathbb{R} f(\tilde{Z} + \|\Sigma_X^{1/2}\theta_*\|_2\sin\alpha \cdot \zeta)\phi(\zeta)d\zeta \\
&= \|\Sigma_X^{1/2}\theta_*\|_2\sin\alpha \int_\mathbb{R} f'(\tilde{Z} + t_2(Z,\zeta)\|\Sigma_X^{1/2}\theta_*\|_2\sin\alpha \cdot \zeta)\zeta\phi(\zeta)d\zeta,
\end{aligned}
$$

where $t_2(Z,\zeta)$ is a constant between $[0,1]$ depending on $Z$ and $\zeta$, and this further implies that

$$
\begin{aligned}
&\mathbb{E}\left[\left|f(\tilde{Z}) - g(Z)\right|\mathbb{I}_{\{|Z-\mu_X^\top\hat{\theta}_0|\leq W\}}\right] \\
\leq& \|\Sigma_X^{1/2}\theta_*\|_2\sin\alpha \int_{|Z-\mu_X^\top\hat{\theta}_0|\leq W} \int_\mathbb{R} \left|f'(\tilde{Z} + t_2(Z,\zeta)\|\Sigma_X^{1/2}\theta_*\|_2\sin\alpha \cdot \zeta)\zeta\right|\phi(\zeta)d\zeta dF(Z) \\
=& O(\sqrt{d}n^{-\frac{1}{3}}),
\end{aligned}
\tag{26}
$$

where the last equality is due to Assumption 2.3 that $f'(x)$ is bounded, Eqn. 17 that $\|\Sigma_X^{1/2}\theta_*\|_2 \leq \|\Sigma_X^{1/2}\theta_*\|_1 = 1$ and Eqn. (22).

Combining Eqn. (25) and Eqn. (26) give us that

$$(\text{IV}) = O(\sqrt{d}n^{-\frac{1}{3}}).$$

**Step 2** (bound (V)): For term (V), we have

$$(\text{V}) = \int_{|Z - \mu_X^\top \hat\theta_0| \leq W} \int \mathbb{E}\left[ |\hat f(Z) - g_0(Z)| \Big| Z_1, \ldots, Z_n, Z \right] dF(Z_1, \ldots, Z_n) dF(Z).$$

For any fixed $Z$, we let $B_n(Z)$ to be the event $\{n\mathbb{P}_n(B(Z,h)) > 0\}$, where $\mathbb{P}_n(B(Z,h)) = \frac{1}{n}\sum_{i=1}^n \mathbb{I}_{\{|Z_i - Z| \leq h\}}$, then we further have

$$
\begin{aligned}
\mathbb{E}\left[ |\hat f(Z) - g_0(Z)| \Big| Z_1, \ldots, Z_n, Z \right] &\leq \mathbb{E}^{\frac{1}{2}}\left[ (\hat f(Z) - g_0(Z))^2 \Big| Z_1, \ldots, Z_n, Z \right] \\
&= \mathbb{E}^{\frac{1}{2}}\left[ \left( \frac{\sum_{i=1}^n (y_i - g(Z_i))\mathbb{I}_{\{|Z_i - Z| \leq h\}}}{\sum_{i=1}^n \mathbb{I}_{\{|Z_i - Z| \leq h\}}} \right)^2 \Big| Z_1, \ldots, Z_n, Z \right] \\
&= \left( \frac{\sum_{i=1}^n \mathrm{Var}(Y_i|Z_i)\mathbb{I}_{\{|Z_i - Z| \leq h\}}}{n^2 \mathbb{P}_n(B(Z,h))^2} \right)^{\frac{1}{2}} \\
&\lesssim \frac{1}{n\mathbb{P}_n(B(Z,h))} \mathbb{I}_{B_n(Z)}.
\end{aligned}
$$

For the last inequality, we can further prove that $\mathrm{Var}(Y_i|Z_i) \leq \mathbb{E}(Y_i^2|Z_i) \lesssim 1$. Specifically, by definition, we know that

$$\mathrm{Var}(Y|Z = z) \leq \mathbb{E}(Y^2|Z = z) \leq 2\int_{\mathbb{R}} f^2(\tilde Z(z) + \|\Sigma_X^{1/2}\theta_*\| \sin\alpha \cdot \zeta)\phi(\zeta)d\zeta + 2\sigma^2 \lesssim 1,$$

where the last inequality holds since $f$ is bounded and $\sigma^2$ is finite.

When conditional on $Z$, we have $n\mathbb{P}_n(B(Z,h)) = \sum_{i=1}^n \mathbb{I}_{\{|Z_i - Z| \leq h\}} \sim \mathrm{Binomial}(n, q)$, with $q = \mathbb{P}(Z_1 \in B(Z,h)|Z)$, and $B(Z,h)$ represents the ball centered at $Z$ with the radius $R$. Thus, when conditional on $Z$, we obtain

$$\int \frac{\mathbb{I}_{B_n(Z)}}{n\mathbb{P}_n(B(Z,h))} dF(Z_1, \ldots, Z_n) = \int \frac{\mathbb{I}_{\{n\mathbb{P}_n(B(Z,h))>0\}}}{n\mathbb{P}_n(B(Z,h))} dF(Z_1, \ldots, Z_n) \leq \frac{1}{nq},$$

where the last inequality follows from Lemma 4.1 in Györfi et al. (2006). We further get one upper bound for (V) as

$$(\text{V}) \lesssim \int_{|Z - \mu_X^\top \hat\theta_0| \leq W} \frac{dF(Z)}{n\mathbb{P}(Z_1 \in B(Z,h)|Z)}.$$

As $\{|Z - \mu_X^\top \hat\theta_0| \leq W\}$ is a bounded area, we can choose $a_1, \ldots, a_R$ such that $\{|Z - \mu_X^\top \hat\theta_0| \leq W\}$ is covered by $\cup_{i=1}^R B(a_i, h/2)$ with $R \leq cW/h$ for some constant $c > 0$. Then we finally bound the term (V) as

$$
\begin{aligned}
(\text{V}) &\lesssim \int_{|Z - \mu_X^\top \hat\theta_0| \leq W} \frac{dF(Z)}{n\mathbb{P}(Z_1 \in B(Z,h)|Z)} \leq \sum_{i=1}^R \int \frac{\mathbb{I}_{\{Z \in B(a_i, h/2)\}} dF(Z)}{n\mathbb{P}(Z_1 \in B(Z,h)|Z)} \\
&\leq \sum_{i=1}^R \int \frac{\mathbb{I}_{\{Z \in B(a_i, h/2)\}} dF(Z)}{n\mathbb{P}(Z_1 \in B(a_i, h/2))} \leq \frac{R}{n} \leq \frac{cW}{nh} \lesssim \frac{\sqrt{\log(n)}}{n^{2/3}},
\end{aligned}
\tag{27}
$$

where the last inequality is due to the set that $h \asymp n^{-1/3}$.

**Step 3** (bound (VI)): We first showcase that function $g(z)$ defined in Eqn. (23) is a Lipschitz function, with Lipschitz constant bounded by the order of $\sqrt{d}$: for any $z_1, z_2 \in \mathbb{R}$, by the mean value theorem, we have

$$
\begin{aligned}
|g(z_1) - g(z_2)| &\leq \left| \cos\alpha \frac{\|\Sigma_X^{1/2}\theta_*\|_2}{\|\Sigma_X^{1/2}\hat\theta_0\|_2}(z_2 - z_1) \right| \cdot \\
&\quad \int_{\mathbb{R}} \left| f'(\tilde Z(z_1) + \|\Sigma_X^{1/2}\theta_*\|_2 \sin\alpha \cdot \zeta) + t(\zeta)\cos\alpha \frac{\|\Sigma_X^{1/2}\theta_*\|_2}{\|\Sigma_X^{1/2}\hat\theta_0\|_2}(z_2 - z_1) \right| \phi(\zeta)d\zeta \\
&\leq \frac{\|\Sigma_X^{1/2}\theta_*\|_2}{\|\Sigma_X^{1/2}\hat\theta_0\|_2}|z_2 - z_1| L_{f'} \leq \sqrt{d}|z_2 - z_1| L_{f'} \lesssim \sqrt{d}|z_2 - z_1|,
\end{aligned}
$$

where the second inequality uses the boundness of $f(\cdot)$ and $f'(\cdot)$, and the third inequality utilizes the fact that $\|\Sigma_X^{1/2}\theta_*\|_2 \leq \|\Sigma_X^{1/2}\theta_*\|_1 = 1$, and $\sqrt{d}\|\Sigma_X^{1/2}\hat{\theta}_0\|_2 \geq \|\Sigma_X^{1/2}\hat{\theta}_0\|_1 = 1$. To deal with term (VI), we first bound the difference between $g_0(Z)$ and $g(Z)$,

$$
\begin{aligned}
|g_0(Z) - g(Z)| &= |g_0(Z)\mathbb{I}_{B_n(Z)} - g(Z)\mathbb{I}_{B_n(Z)} - g(Z)\mathbb{I}_{B_n(Z)^c}| \\
&= \left| \frac{\sum_{i=1}^n (g(Z_i) - g(Z))K_h(Z - Z_i)}{\sum_{i=1}^n K_h(Z - Z_i)}\mathbb{I}_{B_n(Z)} + g(Z)\mathbb{I}_{B_n(Z)^c} \right| \\
&\leq \sqrt{d}h + g(Z)\mathbb{I}_{B_n(Z)^c},
\end{aligned}
$$

where the last inequality follows from the Lipschitzness of $g(\cdot)$, which yields that $g$ is a Lipschitz function with the Lipschitz constant bounded by $\sqrt{d}$. We proceed to have

$$
\begin{aligned}
&\mathbb{E}\left[ |g_0(Z) - g(Z)|\mathbb{I}_{|Z - \mu_X^\top \hat{\theta}_0| \leq W} \right] \\
&\leq \sqrt{d}h + \int_{|Z - \mu_X^\top \hat{\theta}_0| \leq W} g(Z)\mathbb{E}(\mathbb{I}_{B_n(Z)^c}|Z)dF(Z) \\
&\leq \sqrt{d}h + L_f \int_{|Z - \mu_X^\top \hat{\theta}_0| \leq W} \left(1 - \mathbb{P}(Z_1 \in B(Z,h)|Z)\right)^n dF(Z) \\
&\leq \sqrt{d}h + L_f \int_{|Z - \mu_X^\top \hat{\theta}_0| \leq W} \exp(-n\mathbb{P}(Z_1 \in B(Z,h)|Z))\frac{n\mathbb{P}(Z_1 \in B(Z,h)|Z)}{n\mathbb{P}(Z_1 \in B(Z,h)|Z)}dF(Z) \\
&\lesssim \sqrt{d}h + \sup_{u \in [0,1]} \{ue^{-u}\} \int_{|Z - \mu_X^\top \hat{\theta}_0| \leq W} \frac{1}{n\mathbb{P}(Z_1 \in B(Z,h)|Z)}dF(Z) \\
&\lesssim O(\sqrt{d}n^{-\frac{1}{3}}),
\end{aligned}
$$

where the third inequality is due to $(1 - x)^n \leq e^{-nx}$ for any $x \in [0,1]$, and the last inequality comes from the fact that $\sup_{u \in [0,1]}\{ue^{-u}\} \leq 1$ and the result in Eqn. (27). We finish the proof of the theorem. $\qquad\square$

## L.4    PROOF OF THEOREM 3.9

First, note that the Gaussian distribution naturally follows Assumption 2.2 with constant $M$. And the proof of Theorem 3.1 only relies on the boundness of the unknown function $f(\cdot)$. Therefore, based on Theorem 3.1 and Lemma L.1, it holds that

$$
\left\|\hat{\theta} - \mu_*\theta_*\right\|_1 \leq 3d \cdot \sqrt{\frac{M(\sigma^2 + L_f^2)\log(2d/\delta)}{T_1}} = \tilde{O}\left(\frac{d}{\sqrt{T_1}}\right), \quad \left\|\hat{\theta}_0 - \mu_*\theta_*\right\|_1 = \tilde{O}\left(\frac{d}{\sqrt{T_1}}\right).
$$

Furthermore, we can deduce that $d^6 = O(T_1)$. Then based on Theorem L.2 for $X \sim \mathcal{D} = N(\mu_X, \Sigma_X)$, we have

$$
\mathbb{E}\left|\hat{f}(X^\top \hat{\theta}_0) - f(X^\top \theta_*)\right| = O\left(\frac{d^{\frac{1}{2}}}{T_1^{\frac{1}{3}}}\right).
$$

Since the arm at each round is randomly sampled from $D$, then we know that for any $t \in [T], k \in [K]$, it holds that

$$
\mathbb{E}\left|\hat{f}(x_{t,k}^\top \hat{\theta}_0) - f(x_{t,k}^\top \theta_*)\right| = O\left(\frac{d^{\frac{1}{2}}}{T_1^{\frac{1}{3}}}\right).
$$

Then for any $t > 2T_1$, we have that

$$
\begin{aligned}
\mathbb{E}\left(f(x_{t,*}^\top \theta_*) - f(x_t^\top \theta_*)\right) &= \mathbb{E}\left(f(x_{t,*}^\top \theta_*) - \hat{f}\left(x_{t,*}^\top \hat{\theta}\right)\right) + \mathbb{E}\left(\hat{f}\left(x_{t,*}^\top \hat{\theta}\right) - f(x_t^\top \theta_*)\right) \\
&\leq \mathbb{E}\left(f(x_{t,*}^\top \theta_*) - \hat{f}\left(x_{t,*}^\top \hat{\theta}\right)\right) + \mathbb{E}\left(\hat{f}\left(x_t^\top \hat{\theta}\right) - f(x_t^\top \theta_*)\right) \\
&\lesssim \left(\frac{d^{\frac{1}{2}}}{T_1^{\frac{1}{3}}}\right).
\end{aligned}
$$

Table 1: Average running time (in seconds) of each method under different link functions in our simulations in Figure 1 and real-data experiments in Table 3.

| Dataset | ESTOR | STOR | GSTOR | UCB-GLM | GLM-TSL | LinUCB | LinTS |
|---|---|---|---|---|---|---|---|
| (1). Linear | 0.69 | 0.20 | 0.79 | – | – | 0.39 | 0.71 |
| (2). Poisson | 0.72 | 0.26 | 0.81 | 131.24 | 364.89 | – | – |
| (3). Square | 0.76 | 0.28 | 0.89 | 249.01 | 707.95 | – | – |
| (4). Five | 0.76 | 0.29 | 0.91 | 1151.08 | 3276.35 | – | – |
| Forest Cover | 4.61 | 1.88 | 5.01 | 662.09 | 1512.92 | 3.64 | 3.39 |
| Yahoo News | 0.43 | 0.16 | 0.52 | 57.80 | 121.35 | 0.28 | 0.42 |

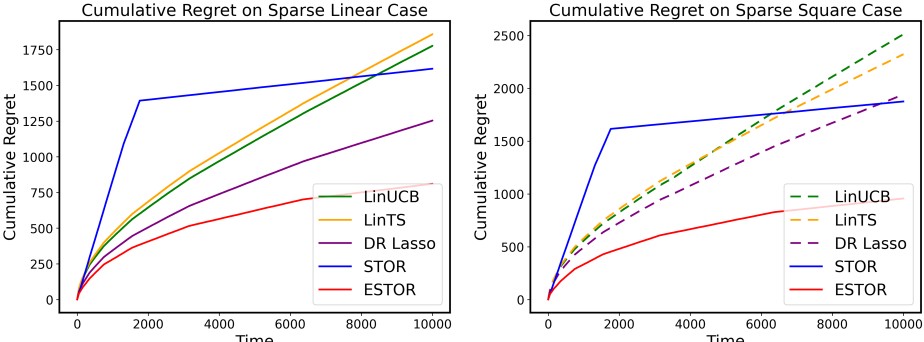

Figure 2: Plot of regret of STOR, ESTOR, LinUCB, LinTS and DR Lasso under the sparse high-dimensional cases (left: identity reward function, right: square reward function).

Therefore, based on the choice of $T_1$ we have that:

$$\mathbb{E}(R_T) = \sum_{t=1}^{2T_1} \mathbb{E}\left[f(x_{t,*}^\top \theta_*) - f(x_t^\top \theta_*)\right] + \sum_{t=2T_1+1}^{T} \mathbb{E}\left[f(x_{t,*}^\top \theta_*) - f(x_t^\top \theta_*)\right]$$

$$\lesssim 2L_f T_1 + \left(\frac{d^{\frac{1}{2}}}{T_1^{\frac{1}{3}}}\right) \cdot T = O\left(d^{\frac{3}{8}} T^{\frac{3}{4}}\right).$$

$\square$

## M   EXPERIMENTAL DETAILS

### M.1   ADDITIONAL DETAILS OF SIMULATIONS

We first report the hyperparameter configuration used in our experiments. We use the false rate $\delta = 0.05$, the dimension $d = 15$ and the number of arms at each round $K = 20$. Each entry of the contextual vector is i.i.d. sampled from a standard normal distribution. In each of the 20 repetitions, the parameter vector $\theta_*$ is generated by independently sampling each entry from a standard normal distribution and then normalizing the vector to have an $l_1$ norm equal to 1, in accordance with our problem setting in Section 2. To ensure a fair comparison, we use the theoretically recommended values for key hyperparameters in each algorithm, such as the exploration rate for UCB-based methods, and parameters $\tau_i$, $T_0$, and $T_1$ for our ESTOR and STOR algorithms. For each method, we conduct two sets of experiments by multiplying the key hyperparameters by 1 and by 2, respectively. Each configuration is run over 20 repetitions, and we report the better regret curve over two different settings for each method. Specifically, for UCB-based methods such as LinUCB and UCB-GLM, the key hyperparameter is the exploration rate scaling the confidence bound. For TS-based methods, including LinTS and GLM-TSL, the key parameter is the variance multiplier of the posterior distribution, which plays a role analogous to the exploration rate in UCB by influencing the spread of the posterior

distribution. For ESTOR, the main hyperparameter is the threshold value $\tau_i$ used in estimating the unknown parameter $\theta_*$. We adopt the theoretical value $\tau_i = \sqrt{3(e_{i-1} - e_{i-2})/\log(2d\log_2(T)/\delta)}$ and evaluate two versions of the algorithm using multipliers 1 and 2 on $\tau_i$, respectively. We set $T_0 = 50$ since it can be any value less than a bound in the theoretical result. For STOR, we consider the exploration phase length $T_1$ as the key hyperparameter. Inspired by our theoretical analysis, we use the formula $T_1 = (dT)^{2/3}\log(2d/\delta)^{1/3}/8$ and apply multipliers of 1 and 2 to this value as well.

We also conducted experiments with GSTOR, but did not include its results in Figure 1, as that figure focuses on the monotone setting, where GSTOR, designed for general reward functions, is comparably less efficient. Instead, we report GSTOR's performance along with all other implemented methods in Table 2. Compared to STOR, GSTOR performs slightly worse in the first three cases, likely because STOR avoids the additional exploration overhead required for reward function estimation, while GSTOR incurs some efficiency loss in that process. However, GSTOR clearly outperforms GLB-based algorithms under model misspecification, demonstrating its robustness across a range of reward functions. Moreover, in the final case, GSTOR outperforms STOR, suggesting that when the reward function is sufficiently complex, explicit estimation may improve overall approximation accuracy and lead to better performance.

We also report the average running time of different methods across the four cases in Figure 1 over 20 repetitions. All simulations were conducted on a machine equipped with the Apple M3 chip. Consistent with our time complexity analysis in section 3.3, our proposed methods are significantly faster than the commonly used GLB algorithms. Specifically, STOR, ESTOR and GSTOR are all hundreds of times faster than UCB-GLM and thousands of times faster than GLM-TSL, demonstrating their strong practical scalability. This efficiency stems from the fact that our methods avoid solving computationally intensive optimization problems at each iteration. Moreover, our methods exhibit stable running times across all settings. In contrast, the running times of UCB-GLM and GLM-TSL vary depending on the reward functions, which is due to the varying difficulty of solving their respective optimization problems. In summary, our methods are not only robust to unknown reward functions but also substantially more efficient than these state-of-the-art GLB algorithms.

Furthermore, we evaluate the sparse high-dimensional setting (Section 3.4) by setting $K = 30$, $d = 60$, and sparsity level $r = 10$, where the indices of non-zero entries are sampled uniformly without replacement. We include the classic sparse linear bandit algorithm, DR Lasso Bandit (Kim & Paik, 2019), for a fair comparison. In the first plot of Figure 2, we consider the linear case with identity reward function; in the second plot, we adopt the same nonlinear square function $f(x) = \text{sign}(x) \cdot x^2 + 2x$ used in our main experiments. Since LinUCB, LinTS, and DR Lasso Bandit are only designed for linear settings and face significant computational challenges under nonlinear rewards, we still fit them with a linear reward model to assess their robustness under model misspecification.

All the hyperparameter configurations are the same as the former simulations on low-dimensional settings. For the regularizer parameter $\lambda$ in Eqn. (1), we leverage its exact theoretical value deduced from Theorem 3.7. We showcase the average cumulative regret over 20 repetitions in Figure 2. Compared to the low-dimensional results in Figure 1(1), our methods ESTOR and STOR outperform LinUCB and LinTS, and ESTOR in particular performs better than DR Lasso Bandit. Under the nonlinear square reward, all three baselines suffer from misspecification and exhibit linear regret growth, while our methods retain a sublinear regret curve with stronger performance. These results demonstrate that our algorithms are not only agnostic to the reward function and highly efficient, but also extend effectively to the sparse high-dimensional regime, maintaining robust and accurate performance.

## M.2 ADDITIONAL DETAILS OF REAL-WORLD EXPERIMENT

For our real-world experiment, we use the Forest Cover Type dataset (Blackard, 1998) from the UCI repository and the benchmark Yahoo Today Module dataset on news article recommendation (Chu et al., 2009). We approximate the arm feature vector distribution by fitting a normal distribution using the estimated mean and covariance matrix from a small subset of the data, since our proposed methods further rely on the distribution of the feature vector. We will show that this approximation works effectively, as our methods demonstrate superior and consistent performance, highlighting their robustness and efficiency in real-world applications. However, we do not provide theoretical guarantees

Table 2: Average final cumulative regret of each method under different link functions in our simulations in Figure 1. For entries in the format a/b, the first value corresponds to the case where the model is correctly specified, while the second value represents performance under model misspecification.

| Dataset | ESTOR | STOR | GSTOR | UCB-GLM | GLM-TSL | LinUCB | LinTS |
|---------|-------|------|-------|---------|---------|--------|-------|
| Linear | 289.37 | 571.62 | 604.45 | – | – | 309.35 | 405.21 |
| Poisson | 993.99 | 2036.94 | 2199.01 | 830.68 | 1148.02 | – | – |
| Square | 482.92 | 1331.41 | 1397.26 | 713.36/1314.54 | 875.12/1692.19 | – | – |
| Five | 198.51 | 611.11 | 519.18 | 136.14/415.05 | 213.87/648.23 | – | – |

Table 3: Results on the Forest Cover Type and the Yahoo news recommendation dataset.

| Dataset | Metric | ESTOR | STOR | GSTOR | UCB-GLM | GLM-TSL | LinUCB | LinTS |
|---------|--------|-------|------|-------|---------|---------|--------|-------|
| Forest | Regret | **844.78** | 1101.28 | 1278.91 | 1497.39 | 2330.80 | 5506.45 | 4081.33 |
| Yahoo | Reward | **349.8** | 302.7 | **344.6** | 255.6 | 248.1 | 221.0 | 219.3 |

under this approximation for real-world applications, as the resulting error would introduce additional terms in the final regret bound. All the algorithmic settings and hyperparameter configurations are identical to the simulations presented above. The Forest Cover Type dataset consists of 581,012 samples with 55 features. The label of each instance denotes a specific type of forest cover. Following the setup in Ding et al. (2021), we assign a binary reward to each data point: a reward of 1 if the point belongs to the first class (Spruce/Fir species), and 0 otherwise. We extract feature vectors from the dataset and partition the data into $K = 32$ clusters, each representing an arm. The reward of a cluster is defined as the proportion of data points within the cluster that have a reward of 1. At every round, we sample a feature vector randomly from each cluster to represent its observation. Table 3 reports the average cumulative regret at the final time step $T = 10,000$, averaged over 10 independent runs. For GLB algorithms UCB-GLM and GLM-TSL, we adopt the logistic bandit setting for both, as this is the default and most common modeling assumption in binary reward scenarios. We report the average cumulative regrets of each method over repetitions in Table 3.

The Yahoo news recommendation dataset comprises over 40 million user visits to the Yahoo Today Module between May 1 and May 10, 2009. In each visit, the user is presented with a news article and chooses to click (reward 1) or not click (reward 0). Both users and articles are represented by feature vectors of dimension 5 with an additional constant feature, constructed via conjoint analysis with a bilinear model (Chu et al., 2009; Yi et al., 2024; 2025). For our experiment, we discard article features and use data from May 1 and May 2, 2009. Due to the heavy click response imbalance, we subsample the data by removing a portion of the non-click (reward 0) events. We set the time horizon to $T = 5,000$, and at each round, randomly select $K = 10$ arms without replacement. For each method, we compute the total reward as the number of clicks accumulated over the time horizon $T$ (higher values indicate better performance). Results are averaged over 10 independent runs and reported in Table 3. The average running time of each algorithm on these two real datasets is also displayed in Table 1.

It is evident that all our proposed algorithms consistently outperform the other methods on both datasets. This strong performance stems from the fact that, in real-world settings, the true underlying link function is unknown and potentially complex. Unlike other methods, our methods do not rely on the explicit knowledge of the link function, again highlighting the advantage of adopting agnostic approaches in practice. In contrast, GLB methods require the reward function to be specified a priori. While the logistic model is commonly used under the binary reward case, this choice is inappropriate in many real-world applications, making such methods susceptible to model misspecification. It is also worth noting that GSTOR demonstrates strong performance on both real-world datasets, particularly on the Yahoo News dataset. This suggests that the underlying reward function in the Yahoo setting may not exhibit a monotonic structure, making GSTOR's flexibility advantageous. In contrast, on the Forest Cover Type dataset, ESTOR and STOR outperform GSTOR, indicating that

the reward function in this environment is likely monotonic. Nonetheless, all three proposed methods consistently outperform the GLB baselines across both datasets.

Moreover, although our methods assume knowledge of the covariate distribution for theoretical analysis, we approximate it using a normal distribution fitted from a small subset of the data in our real data experiments. The strong results obtained under this approximation further demonstrate the robustness and feasibility of our methods in real-world applications.

