# OpenReview forum: "Single Index Bandits: Generalized Linear Contextual Bandits with Unknown Reward Functions"
_ICLR.cc/2026/Conference — ICLR 2026 Poster_

### Official Review · Reviewer_pAj9 · 2025-10-27

**Soundness:** 3
**Presentation:** 2
**Contribution:** 3
**Rating:** 6
**Confidence:** 3

**Summary:**

The paper studies generalized linear bandits (GLBs) with unknown reward functions, referred to as single index bandits (SIBs). This setting is analogous to single index models which generalize generalized linear models by allowing an unknown link function. The presence of an unknown reward function poses substantial theoretical challenges. The authors proposed a family of algorithms for the single index bandits, based on the Stein’s method: STOR and ESTOR for the case where the unknown reward function is monotonically increasing, and GSTOR for the case of general reward functions. The paper provides regret bounds for all proposed algorithms within this new problem setting.

**Strengths:**

The paper makes a clear and original contribution by being the first to extend generalized linear bandits to the setting where the reward function is unknown. The authors carefully discuss the theoretical challenges that arise in this new formulation and emphasize that most of the popular GLB algorithms such as UCB-based methods cannot be directly applied.  To tackle with the challenges, the authors introduce new algorithms based on Stein’s method, which jointly estimate both the underlying parameter and the unknown link function. Notably, Among these, ESTOR achieves a near-optimal regret bound (optimal up to logarithmic factors) under the assumption of a monotone reward function. The paper further extends this approach to the sparse high-dimensional regime and to the general reward function setting. For the latter, the authors propose GSTOR, and prove regret guarantees under a Gaussian design assumption.

**Weaknesses:**

The presentation of the paper could be improved for clarity and intuition. Theorem 3.1 plays a central role across all three proposed algorithms, as the joint learning of both the parameter and the unknown link function is the core challenge of the new problem setup. However, little intuition is given for the minimization problem. It is hard to see how the bound in Theorem 3.1 contributes to the regret analysis that follows. It would be helpful if the authors highlighted where and how the unknown function is estimated within the algorithmic framework. Including a brief proof sketch for Theorem 3.1 and Theorem 3.5 would also improve understanding, especially regarding why the regret bound improves substantially in Theorem 3.5 through epoch scheduling.

In contrast, the sections on STOR and the sparse high-dimensional extension could be presented more concisely, as they are relatively straightforward once ESTOR is clear. The last section on arbitrary reward function is an exciting direction to expand on, but it is presented shortly. The regret bound for the general case is relatively weak, given it is formulated under Gaussian assumptions - It would strengthen the paper to elaborate on the notion of “fundamental unattainability” and to explain more clearly the need/necessity behind the double Etc strategy.

**Questions:**

1.What is the fundamental difficulty in the arbitrary reward function setting that prevents the exponential epoch scheduling technique (used in ESTOR) from being effective?
2.Does the presented upper bound match any known lower bound in this setting? And what about the dependency on K?
3.Could the authors also include the performance of GSTOR in Figure 1 for a more complete empirical comparison across all proposed algorithms?

---

> ### Author Response · Authors · 2025-11-19
> **Thank you for your comments. Please see our responses to your concerns (Part 1)**
>
> Thank you very much for your insightful comments on our work. We are happy to learn that you find our work makes clear and original contributions by tackling a very practical problem, and our estimators and theoretical results are solid. Please see our responses to your comments:
>
> **Weaknesses**
>
> Our work systematically studies the novel single-index bandit (SIB) problem through multiple layers of contribution. Due to space constraints, we could not include detailed proof sketches in the main text; however, we have added a new section titled Summary of Technical Novelty as Appendix C in the submitted revision to clarify the structure and depth of our contributions to resolve your concerns.
>
> For your convenience, we provide detailed answers to your concerns as follows. We have included them in Appendix C as well.
>
> (1). Intuition of Theorem 3.1 and 3.7 with a proof sketch:
>  We use the Stein’s-method-based estimator since Stein’s identity implies that $E[f(X^\top \theta_*) \cdot S(X)] = \mu_* \theta_*$ for some scalar $\mu_* \neq 0$, under mild regularity conditions. Thus, taking a (truncated) average of $y_i \cdot S(x_i)$ yields an estimator that is directionally aligned with $\theta_*$. The truncation function $\phi_\tau$ is essential to control the variance of the estimator in the presence of heavy-tailed rewards, as we show through the bias-variance trade-off in the proof.
>
> The proof of Theorem 3.1 follows a classical comparison argument. We first apply Stein’s identity to show that $E[y_i S(x_i)] = \mu_* \theta_*$, which motivates our estimator. We then analyze the optimization gap between the empirical loss $L(\hat\theta)$ and $L(\mu_* \theta_*)$ by bounding the gradient norm $|\nabla L(\mu_* \theta_*)|_\infty$, where the bias is controlled by truncation and the concentration is handled via Bernstein’s inequality. Finally, applying a first-order Taylor expansion and standard convexity arguments yields the $\ell_2$ and $\ell_1$ estimation bounds. Notably, the estimator admits a closed-form solution in the low-dimensional setting, achieving $\tilde{O}(\sqrt{d/n})$ rate without requiring knowledge of $f(\cdot)$. The proof sketch of Theorem 3.1 has been included in the revised manuscript. For Theorem 3.7, we adapt this argument to the sparse high-dimensional regime by adding an $\ell_1$-penalty to the loss and leveraging subgradient optimality conditions. While the core Stein’s-method-based structure remains intact, we introduce a new support-splitting analysis to decouple estimation over the true support versus its complement. This enables us to obtain sparsity-adaptive rates of $\tilde{O}(\sqrt{s/n})$ in $\ell_2$ norm and $\tilde{O}(s/\sqrt{n})$ in $\ell_1$ norm, again without requiring prior knowledge of the link function.
>
> (2). Missing Explanation of How the Unknown Function is Estimated
> Our approach, based on Stein’s identity, does not require estimating the unknown reward function directly. In fact, the key advantage of our framework lies in circumventing function estimation in the monotonic setting. Specifically, when the link function is assumed to be monotone increasing, the relative ordering of $f(x^\top \theta)$ is preserved by $x^\top \hat\theta$, allowing the algorithm to select actions greedily using $x^\top \hat\theta$ without needing the explicit form of $f(\cdot)$. This leads to significant gains in sample efficiency, as we avoid incurring additional exploration cost for estimating $f$. Notably, this setting still encompasses the entire class of generalized linear bandits (GLB) with canonical exponential family, since their link functions are inherently monotone. Therefore, our estimator implicitly handles the unknown function via structural assumptions and bypasses the need for direct estimation of $f(\cdot)$.
>
> (3). Advantage of ESTOR
> It is well known that classical Explore-then-Commit (EtC) algorithms fail to achieve minimax-optimal regret because parameter estimation is performed using only the initial exploration phase, discarding valuable information from subsequent rounds. To address this inefficiency, we propose ESTOR, an epoch-based variant of STOR that employs exponentially growing epochs. At the beginning of each epoch, ESTOR estimates the parameter using samples from the immediately preceding epoch and then performs greedy selection based on this estimate. This design ensures that the parameter is continually refined using the latest data, overcoming the static-sample limitation of standard EtC. Crucially, our SIM estimator aligns naturally with this structure: because the greedy selection induces a tractable sampling distribution in each epoch, we can apply Stein’s identity to obtain high-quality estimates, enabling sample-efficient learning and allowing ESTOR to achieve the optimal regret bound of $\tilde{O}(\sqrt{T})$.
>
>
> (Please see Part 2 for the rest of our response to the weakness)

---

> ### Author Response · Authors · 2025-11-19
> **Thank you for your comments. Please see our responses to your concerns (Part 2)**
>
> (the rest of our response to the weakness)
>
> Proof Sketch of Theorem 3.5: We first characterize the distribution over arms induced by greedy selection in each epoch and show that samples follow this distribution. Applying Stein’s identity, we bound the estimation error of $\hat\theta_i$ in terms of the sample size from the previous epoch. We then upper bound the per-round regret within each epoch by the corresponding estimation error. Since epoch lengths grow exponentially, the error terms decay geometrically across epochs. Summing the regret over all epochs yields a cumulative regret of $\tilde{O}(dK^{3/2}\sqrt{T})$. We have included a proof sketch of Theorem 3.5 in the submitted revision.
>
> (4). GSTOR under Gaussian Assumptions:
> In the general setting with arbitrary differentiable reward functions, we introduce GSTOR, which employs a double exploration strategy. The first phase is dedicated to estimating the underlying parameter $\theta_*$, while the second focuses on learning the unknown link function $f(\cdot)$. This separation is necessitated by a fundamental challenge: consistent estimation of a general, differentiable function $f$ requires strong assumptions on the covariate distribution. Although the Gaussian design assumption in our analysis may appear strong, we emphasize that it is a mild and standard assumption in this setting. As established in the nonparametric regression literature (Härdle, 2004; Carroll et al., 1997), even in offline settings, uniform estimation guarantees typically assume Gaussian or well-behaved covariate designs. Under only differentiability and bounded derivative assumptions, achieving uniform convergence for estimating $f$ over arbitrary distributions remains a longstanding open problem. Therefore, our double exploration strategy is statistically grounded, and we explicitly highlight this limitation as an open and challenging direction in our conclusion (Line 482).
>
> We would like to clarify that the paper systematically addresses several closely related components, and each of them is developed with depth. The theoretical analyses include complete proofs, the algorithmic designs are fully derived, and the experiments directly validate the theory. Each section is essential: together, they provide a coherent flow from estimator construction to novel algorithm design and then to extensions for sparse high-dimensional and the general function setting. Due to space limitations, in the main paper, we could not include technical details, but all the detailed proof is included in the Appendix. We have made the structure and contributions clearer by adding Appendix C of the uploaded revision to highlight the depth of each component more explicitly.
>
> **Q1. What if the fundamental difficulty in the arbitrary reward function setting**
>
> We thank the reviewer for raising this important question. The core difficulty in the arbitrary reward function setting is that consistent estimation of a general, differentiable link function requires strong distributional assumptions, even in all offline SIM literature. As shown in classical nonparametric regression theory (Härdle, 2004; Carroll et al., 1997), meaningful estimation guarantees typically rely on restrictive assumptions such as Gaussian covariate designs. Under only differentiability and bounded derivative assumptions, obtaining uniform estimation error bounds for general link functions under arbitrary covariate distributions remains an open statistical learning challenge. We highlight this limitation and identify it as an important open direction in our conclusion (Line 482).
>
> Regarding ESTOR, its exponential-epoch design uses samples from the last epoch to estimate the unknown parameter at the beginning of each new epoch. However, since the samples within each epoch are collected greedily based on the current estimate, their distribution may deviate significantly from Gaussian. As a result, the theoretical guarantees for our estimator, derived under Gaussian design assumptions, no longer hold, and this fact may make the epoch-based parameter updates unreliable in this setting. This breakdown prevents us from extending ESTOR’s exponential epoch scheduling to the fully general link function regime. We also would like to emphasize that this general reward-function setting is presented as an extension.
>
> **Q2. Lower bound? Dependence on K?**
>
> The precise dependence of the lower bound on the number of arms $K$ for our problem setting remains unclear since our problem setting is significantly more difficult than GLB, and hence it is unclear whether the rates achieved in the GLB setting are attainable here. We will leave this challenging problem as future work. Finally, as shown in Remark 3.6, our regret bound scales as $\sqrt{\log(K)}$ when $K$ is large, which matches the known lower bound for the simpler linear contextual bandit setting.

---

> ### Author Response · Authors · 2025-11-19
> **Thank you for your comments. Please see our responses to your concerns (Part 3)**
>
> **Q3. Performance of GSTOR in Figure 1 for a more complete empirical comparison**
>
> Thank you for the helpful suggestion. We have implemented GSTOR in all four cases presented in Figure 1. To maintain the visual clarity and accessibility of Figure 1, we report the corresponding performance results for GSTOR in Table 2 of Appendix M.1. For your convenience, we also copy Table 2 of Appendix M.1 below (for entries in the format a/b, the first value corresponds to the case where the model is correctly specified, while the second value represents performance under model misspecification.):
>
> | Dataset  | ESTOR  | STOR   | GSTOR   | UCB-GLM         | GLM-TSL         | LinUCB | LinTS |
> |----------|--------|--------|---------|------------------|------------------|--------|--------|
> | Linear   | 289.37 | 571.62 | 604.45  | --              | --              | 309.35 | 405.21 |
> | Poisson  | 993.99 | 2036.94| 2199.01 | 830.68          | 1148.02         | --     | --     |
> | Square   | 482.92 | 1331.41| 1397.26 | 713.36 / 1314.54| 875.12 / 1692.19| --     | --     |
> | Five     | 198.51 | 611.11 | 519.18  | 136.14 / 415.05 | 213.87 / 648.23 | --     | --     |
>
> Since the four scenarios in Figure 1 all assume monotone reward functions, GSTOR is expected to be less efficient than STOR, which is specifically optimized for such settings. Indeed, GSTOR performs slightly worse than STOR in the first three cases, likely due to the additional exploration required for reward function estimation. In contrast, STOR sidesteps this cost by relying only on directional parameter estimation. However, GSTOR clearly outperforms GLB-based baselines under model misspecification, demonstrating its robustness across a broad class of unknown reward functions. Notably, in the final (non-monotone) case, GSTOR outperforms STOR, suggesting that when the reward function becomes sufficiently complex, explicitly estimating it, as GSTOR does, may lead to improved approximation and better performance. We have put all results and conclusions in our revised manuscript.
>
> Thank you very much for your time and valuable insights. Please let us know if our response has decently resolved your concerns. And we are more than happy to take any additional questions from you.

---

> > ### Comment · Reviewer_pAj9 · 2025-11-26
> >
> > Thanks for the efforts in putting together the proof sketch and setting up new experiments. I see the challenges in the SIB compared to GLB, as well as the challenges of extending it to arbitrary reward functions. I feel a bit conservative of the power of the GSTOR method, given the results above. I would like to keep my score as it is.

---

> ### Author Response · Authors · 2025-11-26
>
> Thank you very much for your thoughtful review. We are glad to hear that the challenges of SIB compared to GLB are now clearer. We agree that the regret bound of GSTOR has room for improvement; however, strengthening the bound under the general SIB setting with arbitrary reward functions is inherently difficult. Therefore, we explicitly highlight this limitation as an important direction for future work in the Conclusion section. Given that the paper already presents several novel contributions, we treat this general case as an extension rather than the core focus of the current work. And we believe our work will trigger many potential future directions as SIB is a new problem in the bandit literature.
>
> Thank you again for your feedback. Please let us know if you have any other questions and we would be more than happy to address them.

---

### Official Review · Reviewer_vpkp · 2025-10-29

**Soundness:** 3
**Presentation:** 3
**Contribution:** 2
**Rating:** 4
**Confidence:** 3

**Summary:**

This paper introduces and addresses the Single Index Bandit problem, a generalization of the Generalized Linear Bandit (GLB) framework. In a SIB, the expected reward is an unknown, potentially non-linear function $f$ of a linear predictor $x^\top \theta_*$, in contrast to GLBs which assume $f$ is known.  The authors propose a family of efficient algorithms based on a novel estimator derived from Stein's method, which avoids the need for complex optimization or a known link function.

**Strengths:**

1. The work directly addresses a critical and often unrealistic limitation (known link function) in the extensive GLB literature, making bandit algorithms significantly more robust to model misspecification.
2. The paper is generally well-written and structured. The motivation is compelling, and the challenges of the problem are clearly explained, especially in contrasting with existing GLB and general contextual bandit approaches.

**Weaknesses:**

1. The analysis crucially relies on the assumption that the context vectors (arms) are i.i.d. from a fixed distribution. This is a significant limitation, as a large body of bandit literature deals with adversarially chosen arm sets. Why this assumption is needed?
2. The worst-case regret bound of ESTOR exhibits a $K^{3/2}$ dependence on the number of arms, a limitation not present in prior work on heavy-tailed GLBs. Why this term exists? Is there any lower bound?
3. This paper assumes $||\theta_*||=1$, which is less common in contextual linear bandits. Why the norm bound on the parameter is needed?

**Questions:**

see the weaknesses.

---

> ### Author Response · Authors · 2025-11-19
> **Thank you for your comments. Please see our responses to your concerns (Part 1)**
>
> Thank you very much for your constructive comments and careful review of our work. We’re more than happy to hear that you find our work tackles a critical and unrealistic limitation of existing GLB literature with compelling motivation, and our paper is well-written. Please see our response to your concerns:
>
> **W1. Assumptions are strict**
>
> As noted in our Conclusion section, we acknowledge that assuming i.i.d. contexts from a fixed distribution may seem restrictive compared to adversarially chosen arms, which are common in generalized linear bandits (GLBs). However, as we carefully elaborate in Appendix B (Related Works), the single index model (SIM)/single index bandit (SIB) is fundamentally more challenging than the generalized linear model (GLM)/generalized linear bandit (GLB) setting, both statistically and algorithmically. Notably, even in the offline statistical learning literature, all existing SIM works require stronger distributional assumptions, such as standard Gaussian contexts or i.i.d. entries from a known distribution, to obtain any form of non-asymptotic theoretical guarantee. For example:
> - Thrampoulidis et al. (2015), Plan & Vershynin (2016), and others assume standard Gaussian contexts.
> - Fan et al. (2023) also require i.i.d. context entries and assume known distributional form and finite forth moments.
> - In the monotone SIM setting, works like Balabdaoui et al. (2019) still only achieve n^{1/3} convergence under strong assumptions on both noise and covariates.
>
> These facts underscore that no existing work can handle adversarially chosen arms in the offline SIM setting, even offline. Extending SIM analysis to adversarial contexts remains an open and technically formidable problem in statistical learning, let alone the even more challenging bandit setting. In contrast, our i.i.d. context assumption is already weaker than all prior SIM literature, and enables us to tackle the more difficult online bandit setting where the unknown reward function must be learned in a sequential and data-limited regime.
>
> Finally, while GLBs support adversarial arms, they fundamentally assume a known link function, making the setting much more tractable. Notably, all techniques developed for GLBs fail to extend to SIBs where the link is unknown.
>
> As a conclusion, our assumption is appropriate and currently necessary for any meaningful regret guarantee in the single index bandit setting.
>
> **W2. The worst-case regret bound of ESTOR exhibits $K^{3/2}$ dependence**
>
> Thank you for your careful review. The $K^{3/2}$ dependence arises from the need to uniformly control the second moment of the Stein score function of our estimator across epochs, and the magnitude of the score function can be large, which introduces additional complexity compared to generalized linear bandits with sub-Gaussian or heavy‑tailed noise. We respectfully feel it is not appropriate to directly compare our bound with prior GLB results, as our problem setting is substantially more challenging, and all existing techniques to handle GLBs do not apply here, as mentioned in the related work section in Appendix B.
>
> Moreover, the precise dependence of the lower bound on the number of arms $K$ for our problem setting remains unclear since our problem setting is significantly more difficult than GLB, leaving open the question whether the rates achieved in the GLB setting are attainable here. We will leave this challenging problem as future work. Finally, as shown in the Remark 3.6, our regret bound scales as $\sqrt{\log(K)}$ when $K$ is large, which matches the known lower bound for the simpler linear contextual bandit setting.

---

> ### Author Response · Authors · 2025-11-19
> **Thank you for your comments. Please see our responses to your concerns (Part 2)**
>
> **W3. The paper assumes norm = 1**
>
> Thank you for the insightful question. As noted in Line 132, we have to assume $\||\theta_*\|| = 1$ (but not $\||\theta_*\|| \leq 1$) to ensure model identifiability. In the single-index model/bandit setting, where the reward function is unknown, there is inherent ambiguity due to the scaling symmetry between $\theta_*$ and the link function $f(\cdot)$. For example, the pair $(\theta_*, f(z)) = (\theta_1, z)$ is equivalent to $(\theta_*, f(z)) = (2\theta_1, z/2)$ and both lead to the same observed rewards. Without fixing the norm of $\theta_*$, the model is not identifiable, as infinitely many parameter-function pairs generate the same behavior. On the contrary, the generalized linear bandit setting typically assumes $\||\theta_*\|| \leq 1$, as the link function is known and fixed, and hence the identifiability issue does not arise there.
>
> Therefore, this is a standard assumption in all SIM literature (Han, 1987; Härdle, 2004; Carroll et al., 1997; Thrampoulidis et al., 2015; Plan & Vershynin, 2016; Na et al., 2019; Neykov et al., 2016; Fan et al., 2023). This assumption is not restrictive, but rather necessary to define the target parameter uniquely under the SIM/SIB setting.
>
> We truly appreciate your thoughtful feedback and sincerely hope our response has thoroughly addressed your concern. We would be grateful if you could reconsider your assessment of our work in light of these clarifications provided and the great contributions of our work. Please feel free to let us know if you have any other concerns, and we are more than happy to address them.

---

> > ### Comment · Reviewer_vpkp · 2025-11-26
> >
> > Thank you for the authors’ response. I no longer have further concerns and have raised my score.

---

> > > ### Author Response · Authors · 2025-11-26
> > >
> > > Thank you very much for your careful review and insightful feedback. Please feel free to let us know if you have any other questions, and we are happy to engage in further discussion at any time.

---

### Official Review · Reviewer_Fk6c · 2025-10-31

**Soundness:** 3
**Presentation:** 2
**Contribution:** 2
**Rating:** 4
**Confidence:** 4

**Summary:**

This paper introduces Single Index Bandits (SIB), extending generalized linear bandits (GLBs) to settings where the reward function $f$ is unknown. Note that GLBs require the link function, and misspecification in the link function can lead to linear regret. To address this, the authors propose a novel Stein's method-based estimator that achieves minimax-optimal error rates $O\sqrt{d/n}$ without knowing $f$,  requiring only $O(nd)$ time complexity. For monotonically increasing $f$, they develop STOR (explore-then-commit), achieving $O(T^{2/3})$ regret and ESTOR (epoch-based), achieving near-optimal $O(\sqrt{T})$ regret. The framework extends to sparse high-dimensional settings, replacing $d$ with sparsity $s$ in regret bounds. For general non-monotonic $f$, GSTOR employs kernel regression under Gaussian design, achieving $O(T^{3/4})$ regret. The Stein's estimator cleverly exploits $E[f(X^\intercal \theta^{\star})S(X)] = E[f'(X^\intercal \theta^{\star})]\theta^{\star}$ through integration by parts, using truncation to control variance. Experiments on synthetic and real datasets (Forest Cover, Yahoo News) demonstrate substantial improvements over GLB methods, with ESTOR/STOR running 100-1000x faster than UCB-GLM/GLM-TSL.

**Strengths:**

1. The paper tackles a fundamental limitation of GLBs - the known reward function (link function) assumption. By using an the Stein's method approach, they address this issue. The resulting estimator is both theoretically optimal (achieving minimax rates) and computationally efficient (closed-form solution requiring no optimization).
2. The paper provides two algorithms, STOR, a simpler variant with uniform exploration, and ESTOR which is an epoch-based algorithm that balances exploration-exploitation. Finally, GSTOR, stated in the Appendix for non-monotonic functions, is the algorithm that tackles non-monotonic reward functions.
3. Synthetic experimentation shows good performance against baselines.

**Weaknesses:**

1. To me, one of the main weaknesses of the paper is that it tries to pack too much stuff into the paper without going deep into one section.
2. Following the previous comment, none of the theorems, technical novelty has been explained in detail.
3. The sparse high-dimensional experiments only test linear rewards (not truly testing the unknown f capability), GSTOR evaluation is limited.

**Questions:**

1. The technical novelty is not clear to me. I actually went over the proof of Theorem 3.1 in the appendix. It uses the standard Holder's and Cauchy-Schwarz inequality. Relies on Lemma C.3, which follows directly from Steim's inequality and Bernstein's inequality. Where is the key technical challenge in this?
2. One thing I am slightly confused about is that, to get rid of the knowledge of the link function, the authors mainly rely on the estimator and the quadratic loss function in eq (1), and that data is sampled from a fixed distribution D. Is this enough? Or are there some hidden assumptions I am missing?
3. While theoretically sound, the STOR algorithm (which is an Explore-then-Commit algorithm) is quite impractical. Observe that, combined with Theorem 3.1, the first stage requires $O(dT^{2/3}$ samples. which is very large. Basically, to build the estimator itself, it takes two-thirds of the samples. Now, if you look into a stochastic bandit setting with no structure, the ETC algorithm takes $O\sqrt{T}$ samples. Check Chapter 6 of https://tor-lattimore.com/downloads/book/book.pdf.
4. The sparse high-dimensional setting only tests linear rewards; as such, testing against a synthetic complex f will enhance the paper.

---

> ### Author Response · Authors · 2025-11-19
> **Thank you for your comments. Please see our responses to your concerns (Part 1)**
>
> Thank you very much for your detailed questions and meticulous review of our work. We’re pleased to hear that you find our work tackles a fundamental limitation of GLB and our proposed estimator is novel, theoretically optimal, and computationally efficient with great experimental results. Please see our response to your concerns:
>
> **W1. The paper packs too much stuff**
>
> Thank you very much for your feedback. Our work systematically studies the novel single-index bandit (SIB) problem through multiple layers of contribution. We begin by proposing a new estimator for the SIM with provable optimal error rates under both low-dim and high-dim settings, followed by a warm-up algorithm using the explore-then-commit (EtC) strategy. Building on this, we introduce ESTOR, an efficient epoch-based algorithm achieving the optimal regret of order $\tilde{O}(\sqrt{T})$. We then extend our estimator and algorithms to the high-dimensional sparse regime, showing that both the estimation and regret guarantees scale properly with the sparsity level $s$. Finally, we study the challenging general case with non-monotone reward functions and provide comprehensive experiments on both synthetic and real data to validate our methods. Due to space constraints, we couldn't include proof sketches in the original main text; however, we have added proof sketches and a new section titled Summary of Technical Novelty (Appendix C) in the submitted revision to clarify the depth of our contributions.
>
> We respectfully note that the breadth of our work reflects the solid technical foundation we aim to establish for this new problem class. As the first research effort to provide rigorous theoretical results for SIBs, we believe that addressing multiple facets of the problem is important and necessary. Each section tackles a technically difficult aspect beyond all existing literature. While the paper covers several components, we ensure that each contribution is solid, self-contained and builds on the previous sections. We kindly hope the reviewer can appreciate the significance of our work, and the revised manuscript can make the structure and depth of our contributions clear to you.
>
> **W2, Q1: Technical novelty of our works**
>
> We respectfully clarify that while the proof of Theorem 3.1 builds upon standard tools such as Hölder’s and Cauchy-Schwarz inequalities, as do most modern theoretical results, the technical novelty already lies in the novel design of the estimator and algorithms themselves instead of the use of exotic inequalities.
>
> Our estimator in Eqn. (1), based on truncating $y_i \cdot S(x_i)$, is new to the literature. It is computationally efficient, does not require knowledge of the reward function, and crucially, achieves the optimal estimation rate under mild assumptions. This is non-trivial in the SIM, where existing estimators require more structure (e.g. Gaussian design) or lack computational efficiency, as highlighted in Related Works (Appendix B). Moreover, we extend this estimator to the high-dimensional sparse regime with a simple $\ell_1$ regularizer, and it still achieves the minimax-optimal rate without requiring knowledge of sparsity level s. This already results in a technically significant and practically useful result.
>
> The analysis of ESTOR involves additional technical challenges. We design a novel epoch-based approach where each round uses only past data collected in the last epoch for parameter estimation. This is typically problematic, as greedy sampling leads to highly biased covariate distributions. As a consequence, existing offline estimators, especially for single-index models, typically require restrictive distributional assumptions (e.g., Gaussian design) and will break down under our setting (see Appendix B). In contrast, our proposed estimator remains statistically valid under this adaptive data collection scheme because it operates on score functions and does not rely on specific feature distributions. This compatibility is crucial for achieving the optimal regret bound of order $\tilde{O}(\sqrt{T})$, and to our knowledge, such analysis has not appeared in prior literature.
>
> We hope this clarifies that while the proofs are clean, the technical difficulty lies in proposing a unified, efficient, and theoretically grounded approach to a fundamentally new bandit setting. Our problem formulation is new, the estimator is novel, and the epoch-based algorithm and analysis are carefully tailored to align with the estimator under adaptive data, which go beyond existing estimation methods. These components work together to deliver theoretical guarantees that are unattainable using existing techniques. We respectfully believe this represents a technically significant contribution, where the simplicity of the analysis reflects the strength and elegance of the method, but not a lack of depth. We have added a new Summary of Technical Novelty section (Appendix C) to help readers better navigate these contributions.

---

> ### Author Response · Authors · 2025-11-19
> **Thank you for your comments. Please see our responses to your concerns (Part 2)**
>
> **W3, Q4: The sparse high-dimensional experiments only test linear rewards**
>
> Thank you for your valuable feedback. In response to your suggestion, we have included a new experiment in the sparse high-dimensional setting that evaluates performance under a complex nonlinear reward function. Specifically, in addition to the linear case, we now test our methods using the complex square function $f(x) = \text{sign}(x) \cdot x^2 + 2x$, as used in our main experiments. To benchmark performance, we also include DR Lasso Bandit introduced in [R1] as a representative and classic sparse linear bandit baseline. Since DR Lasso and other linear methods (e.g., LinUCB, LinTS) are only designed for linear settings and face great computational challenges under nonlinear reward function, we fit them under a linear reward model to assess their robustness under model misspecification. The results, now shown in Figure 2 in Appendix M, demonstrate that our methods maintain sublinear regret under nonlinear rewards, whereas the baselines exhibit linear-type growth under model misspecification. We have incorporated this new experiment and corresponding discussion into the revised submission. Thank you very much for helping improve the quality of our work.
>
> [R1]. Doubly-robust lasso bandit, Gi-Soo Kim and Myunghee Cho Paik, NeurIPS 2019.
>
> **Q2. Are there hidden assumptions missing?**
>
> Thank you for the thoughtful question. We confirm that for Theorem 3.1, all assumptions are explicitly stated in the paper. The estimator only relies on Assumptions 2.2 and 2.3, and that the data is sampled from a known fixed distribution. These are standard and mild assumptions, and no hidden conditions are used. This reflects the strength of our estimator: it is computationally efficient, does not require knowledge of the link function, and achieves the optimal error rate under minimal assumptions. As we discuss in Appendix B, our assumptions are strictly weaker than those in prior SIM literature. For example, classical approaches often assume Gaussian or i.i.d. covariates with bounded fourth moments, which we avoid entirely. Therefore, our work significantly advances SIM-based estimation under milder and more realistic conditions.
> For the ESTOR algorithm, the regret analysis further relies on Assumption 3.3, which is standard and easy to satisfy (as discussed in Remark 3.4). Again, we confirm all necessary assumptions are clearly stated in each theorem statement. We believe this fact further highlights the significance, rigor, and technical novelty of our work beyond existing contextual bandit and offline SIM literature.
>
> **Q3. STOR is impractical with longer exploration stage**
>
> Thank you very much for raising this insightful issue. We respectfully clarify that choosing the $n=O(T^{2/3})$ for the exploration phase in our STOR algorithm is both standard and theoretically optimal in the contextual bandit setting. Specifically, with the exploration data size $n$, our estimation already attains the minimax error bound $\|\hat\theta - \theta_*\| = \tilde{O}(1/\sqrt{n})$ in terms of both $\ell_1$ and $\ell_2$ norms (Wainwright 2019), and the final cumulative regret is of the form:
> $$R(T) = n + \frac{T - n}{\sqrt{n}}.$$
> This is minimized when we have the exploration length $n = T^{2/3}$, leading to the regret rate $\tilde{O}_T(T^{2/3})$. Choosing $n = T^{1/2}$, for example, would result in a worse regret of $\tilde{O}_T(T^{3/4})$. Thus, allocating roughly two-thirds of the samples to exploration is not inefficient, but necessary to reach possibly the best theoretical guarantee with the classic EtC framework.
>
> Notably, the contextual bandit is considerably more challenging than the classical MAB, and hence the EtC typically follows a different scaling than the $\sqrt{T}$ exploration budget in stochastic MABs. This is also reflected in prior contextual bandit work. For example:
>
> - [R2] uses $O(T^{2/3})$ exploration for sparse linear and low-rank matrix bandits.
> - [R3] adopts $O(T^{2/3})$ exploration to achieve the best trade-off between exploration and exploitation in high-dimensional settings without restrictive conditions.
> - [R4] shows that exploration length must exceed $T^{1/2}$ to align with the estimator’s statistical rate and achieve the optimal bound.
>
> In summary, taking $T^{2/3}$ samples for exploration is a principled and well-justified choice for our problem setting and broader contextual bandit literature, and ensures that STOR achieves as close to the best possible theoretical regret as one can obtain with the explore-then-commit framework.
>
> [R2]. A simple unified framework for high dimensional bandit problems, Li et al., ICML 2022
>
> [R3]. High-dimensional sparse linear bandits, Hao et al., NeurIPS 2020
>
> [R4]. High-dimensional contextual bandit problem without sparsity, Komiyama et al., NeurIPS 2023
>
> We sincerely appreciate your valuable feedback. Please feel free to let us know if there’s anything else we can elaborate on.

---

> ### Author Response · Authors · 2025-11-27
> **Thank you for your reviews. Please let us know if any additional clarification would be helpful.**
>
> Thank you very much for your thoughtful comments and insightful feedbacks. We have provided a detailed response that carefully addresses all of your concerns, including the points on the breadth of our work, technical novelty, hidden assumptions, exploration length, and the high-dimensional nonlinear experiments. In our replies, we note that the breadth of the paper reflects our goal of systematically developing the first comprehensive theoretical and algorithmic foundation for the single-index bandit problem, with each component addressing a key technical aspect with novel and solid analysis, and we add detailed clarification in Appendix C due to the space limit. We then highlight the substantial novelty of our estimator and algorithms which are new beyond the existing literature, as well as the solid and original theoretical results we establish to deliver guarantees that are unattainable using any existing technique. We also confirm that no implicit assumptions are used in our analysis, clarify that the chosen exploration length is actually optimal and efficient for our structured contextual bandit setting, with support from both theoretical analysis and prior work on contextual bandits. Moreover, we include an additional experiment on a complex nonlinear reward function in the sparse high-dimensional regime to further demonstrate the robustness of our methods over baselines.
>
> We sincerely hope that the clarifications and additional materials in our rebuttal and revised submission sufficiently resolve the issues raised, and we would be grateful if you could kindly reconsider your assessment of our work in light of these explanations and great contributions of our work. Please feel free to let us know if you have any further questions or if additional clarification would be helpful, and we would be more than happy to continue the discussion.

---

### Official Review · Reviewer_2RuZ · 2025-11-07

**Soundness:** 3
**Presentation:** 3
**Contribution:** 3
**Rating:** 6
**Confidence:** 3

**Summary:**

This paper investigates generalized linear bandits with unknown reward functions. The authors propose three algorithms that progressively handle increasingly general link functions. STOR addresses monotone functions using an ETC strategy, ESTOR achieves near-optimal regret through an epoch-based framework, and GSTOR extends the approach to arbitrary functions under Gaussian design. The paper explores sparse high-dimensional settings and leverages Stein’s identity to estimate model parameters without prior knowledge of the reward function. Extensive experiments validate the effectiveness of the proposed methods.

**Strengths:**

1) This paper tackles a challenging and relatively unexplored generalized linear bandit setting in which the link function is unknown.
2) It offers a hierarchical perspective on assumptions and algorithms, distinguishing between monotone and general link functions, to illustrate the trade-off between assumptions and regret.
3) The proposed Stein’s estimator is new and conceptually clear, enabling efficient estimation of the underlying parameter without explicitly modeling the link function.

**Weaknesses:**

1) The monotonicity assumption is key to achieving near-optimal regret but remains restrictive. It would be helpful to discuss whether weaker conditions, e.g., local monotonicity or Lipschitz continuity, could still lead to meaningful results.
2) The non-monotone case achieves $O(T^{3/4})$ rate is suboptimal. The dependence on the number of arms $K$ might be suboptimal.
3) The analysis of sparsity is good but assumes knowledge of the true sparsity level.

**Questions:**

1) How critical is the Gaussian design assumption in GSTOR? Could the analysis extend to sub-Gaussian or other distributions, or does it rely on properties unique to the Gaussian setting?
2) Are the dependencies on $T, d, K$ optimal?
3) Can the sparse variants adapt to unknown sparsity levels? How sensitive are the algorithms to misspecification of the sparsity parameter?
4) Are there specific settings where realizability-based contextual bandits can outperform the single-index bandits?

---

> ### Author Response · Authors · 2025-11-19
> **Thank you for your comments. Please see our responses to your concerns (Part 1)**
>
> Thank you for your valuable comments on our work. We are happy to know you find our work tackles a challenging and unexplored problem with a new and conceptually sound estimator. Please see our responses to your questions as follows:
>
> **W1. The monotonicity assumption is restrictive**
>
> Thank you very much for the insightful question. As we explained in line 87, it is a well-known result that generalized linear models in the canonical exponential family can directly imply that the underlying reward function $f(\cdot)$ is monotone increasing (McCullagh, 2019). Therefore, all existing works on GLBs assuming the canonical form, e.g. Filippi et al. 2010 (Eqn. (2)), implicitly have the same monotonicity assumption as well. Our assumption thus still directly subsumes those classical GLB works. Moreover, in the offline statistical literature, the monotone single-index model is the most thoroughly developed and widely applied variant, as we have discussed in Appendix B (Balabdaoui et al. 2019a, Balabdaoui et al. 2019b, Dai et al. 2022). Given that our work is the first to extend single-index models to the contextual bandit setting, this popular monotone setting is a natural and principled starting point. We agree that extending our analysis to weaker assumptions such as local monotonicity or Lipschitz continuity is an important and challenging direction. In fact, these conditions remain underexplored even in the offline SIM statistical learning literature. We therefore view this as a promising and non-trivial avenue for future work, both for offline and online estimation.
>
> **W2. Q2: The non-monotone case is sub-optimal and the dependence on $T, d, K$**
>
> Thank you for your careful review. For the non-monotone setting, we acknowledge that our regret bound may not be optimal. As noted in Line 419, the minimax lower bound for this general SIB setting remains open and is left as a very challenging future work. Due to the significant difficulty of SIMs compared to GLBs, it is unclear whether the lower bounds from GLBs can carry over.
>
> Secondly, we would like to emphasize the main algorithm in our work is ESTOR, which achieves the optimal rate of order $\sqrt{T}$ in terms of $T$, and the time horizon $T$ is the most dominant term over $d$ and $K$ in regret analysis. Regarding the dependence on the number of arms $K$, we further show that the regret of ESTOR scales as $\sqrt{\log(K)}$ when $K$ is large in Appendix J. This matches the known lower bound for the simpler linear contextual bandit setting (as shown in Li et al. 2019). Moreover, the precise dependence of the lower bound on the dimension d and the number of arms K for our problem setting remains unclear, leaving open the question of whether the rates achieved in the GLB setting are attainable here. We will leave this challenging problem as future work. As the first work to provide solid theoretical results for SIBs, we believe our contribution serves as an important starting point for this challenging problem class, and naturally opens up many intriguing future directions that remain largely unexplored.
>
> **W3, Q3: The analysis of sparsity depends on $s$**
>
> Thank you for your constructive comment. We would like to clarify that the implementation and error bound analysis of our Stein’s-method-based estimator under sparsity in Theorem 3.7 does not rely on the knowledge of $s$ at all. In fact, our bounds in both $\ell_1$ and $\ell_2$ norms match the minimax optimal rates (Wainwright, 2019), as they achieve the same rates as sparse Lasso regression, up to logarithmic factors, and Lasso regression is a special case of our broader problem class.
>
> For Algorithm 2 (ESTOR), which is the main algorithm in our work, we would like to emphasize that none of the hyperparameter choices in Corollary 3.8 depend on $s$. And ESTOR does not require any prior knowledge of $s$ to achieve the stated cumulative regret bound of order $\tilde O(sK^{3/2}\sqrt{T}) = \tilde O_T(\sqrt{T})$.
>
> For Algorithm 1 (STOR), the value of $s$ is used only to set the exploration phase length $T_1$. Even if we remove this dependence, we can still prove a regret bound of order $\tilde{O}(sT^{2/3})$, where the exponent of $s$ changes from 2/3 to 1. We have clarified this detail further in the revised manuscript. Thank you very much for helping improve our work.

---

> ### Author Response · Authors · 2025-11-19
> **Thank you for your comments. Please see our responses to your concerns (Part 2)**
>
> **Q1. How critical is the Gaussian design to GSTOR?**
>
> We thank the reviewer for raising this important point. In the fully general setting where the reward function is unknown and potentially non-monotone, we rely on kernel regression to estimate the function. Under only a differentiability assumption with a bounded derivative, obtaining meaningful estimation guarantees under arbitrary reward functions is extremely challenging. As shown in classical nonparametric regression literature (Härdle, 2004; Carroll et al. 1997), existing theoretical guarantees typically require restrictive distributional assumptions such as the Gaussian design adopted in our analysis. Without such assumptions, even offline estimation of a general differentiable function remains an open and difficult problem, and deriving regret guarantees for bandits in this regime is a promising future direction. Note we also highlight this limitation and mention it as a challenging open direction in our Conclusion section (line 482).
>
> We also emphasize that this general reward-function setting is presented as an extension. One of our main contributions is the ESTOR algorithm, which achieves the optimal regret order $\tilde{O}_T(\sqrt{T})$ under the monotonicity assumption.
>
> **Q4. Specific settings where realizability-based contextual bandits outperform the single-index bandits?**
>
> Thank you for this excellent question. As we discuss in Appendix B related work section (Lines 761–788) and clarify below, under the single-index bandit (SIB) setting with an unknown link function, realizability-based contextual bandits cannot obtain meaningful regret guarantees, even in principle. This is due to fundamental incompatibilities between (i) what realizability-based algorithms require from offline regression oracles and (ii) what is statistically achievable for single-index models (SIMs) based on existing literature.
>
> In short, realizability-based contextual bandits rely on strong regression oracles that can estimate the full reward function under mild distributional assumptions. However, in the single index model setting with unknown reward functions, no known estimator satisfies these requirements unless one assumes restrictive designs (e.g., Gaussian design) and hence can't be used. Therefore, realizability-based methods cannot provide meaningful regret guarantees in our setting and are currently incompatible with SIB-based analysis.
>
> Thank you very much for your time and valuable insights. Please let us know if our response has decently resolved your concerns. And we are more than happy to take any additional questions from you.

---

### Author Response · Authors · 2025-11-29
**Summary of the Rebuttal and Discussions**

Dear AC, SAC and PC,

Thank you for your service to the community and for your dedication to orchestrating and overseeing the review process during this challenging period. We are grateful to all reviewers for their constructive feedback and insightful comments. They consistently acknowledge the key strengths of our work:

- Compelling problem setting: Our paper tackles a crucial and highly challenging problem: generalized linear bandits without prior knowledge of the reward function, known as the single-index bandit (SIB) problem. The motivation is compelling, and the challenges of the problem are clearly explained
- Technical breadth and depth: We provide a comprehensive study on the SIB problem, beginning with a novel estimator, then designing an efficient algorithm under unknown increasing reward functions, further extending the theory to the sparse high-dimensional regime, and finally to general reward functions. We also provide extensive empirical results to support our findings.
- Novel and solid analysis: Reviewers note the novelty, rigor, and significance of our theoretical contributions, which go beyond what existing literature can achieve.
- Efficient estimator: Reviewers note our estimator is new and conceptually clear.

---

**Summary of author-reviewer discussion before the incident**: During the initial reviewer–author discussion phase, we received follow-up comments from Reviewer vpkp and Reviewer pAj9 before the unfortunate incident occured. Reviewer vpkp acknowledged that all of their concerns had been fully resolved and raised their rating from 4 to 6, and Reviewer pAj9 confirmed that they had no remaining concerns and maintained their score of 6. These exchanges all took place before the reported anonymity leak incident (we believe the comments with timestamps are still visible), and we respectfully think their comments reflect their genuine, unbiased assessments of our work, unaffected by the later anonymity breach.

For Reviewer 2RuZ and Reviewer Fk6c, we have provided a detailed and thorough rebuttal addressing all of their points. We haven't heard back from them in the discussion phase, and due to the incident they are unable to reply.

---

We summarize the key points of our rebuttal to Reviewer 2RuZ and Reviewer Fk6c below:

1. For Reviewer Fk6c, we carefully responded to their concerns regarding the breadth of the paper, technical novelty, potential hidden assumptions, the choice of exploration length, and the high-dimensional nonlinear experiments. In our replies, we note that the breadth of the paper reflects our goal of systematically developing the first comprehensive theoretical and algorithmic foundation for the single-index bandit problem, with each component addressing a key technical aspect with novel and solid analysis, and we add detailed clarification in Appendix C due to the space limit. We then highlight the substantial novelty of our estimator and algorithms which are new beyond the existing literature, as well as the solid and original theoretical results we establish to deliver guarantees that are unattainable using any existing technique. We also confirm that no implicit assumptions are used in our analysis, clarify that the chosen exploration length is actually optimal and efficient for our structured contextual bandit setting, with support from both theoretical analysis and prior work on contextual bandits. Moreover, we include an additional experiment on a complex nonlinear reward function in the sparse high-dimensional regime to further demonstrate the robustness of our methods over baselines. Together, these clarifications address all concerns raised by the reviewer.

2. For Reviewer 2RuZ, our rebuttal thoroughly addresses all of their concerns. We clarified the monotonicity assumption is natural and commonly used, and highlighted the optimality and novelty of our algorithms, especially ESTOR. We also provided detailed explanations regarding the huge challenges of the non-monotone setting, how our analysis does not require knowledge of the sparsity level $s$, the role of the Gaussian design assumption in existing literature, and why realizability-based contextual bandits are fundamentally incompatible with single-index models.

We have uploaded a revised version of the paper incorporating the reviewers’ comments. And we would greatly appreciate it if you could review our detailed rebuttal and the updated manuscript when forming a well-informed assessment of our work. Thank you again for your significant effort in upholding the quality and integrity of the review process. Please feel free to let us know if you have any other concerns about our work, and we are more than happy to address them.

---

### Meta-Review · Area_Chair_UAkr · 2026-01-05

**Summary:**

This paper investigates the Single Index Bandit (SIB) problem. SIB addresses the limitation of Generalized Linear Bandits (GLBs), which typically requires a known reward function. The authors propose a novel estimator based on Stein's method and develop a family of algorithms. First, they propose STOR and ESTOR for settings where the unknown reward function is monotonically increasing. Second, this work extends these methods to sparse high-dimensional settings as well as general non-monotonic reward functions.



In the following, I will list the concerns of each reviewer.

- **Reviewer 2RuZ**: This reviewer gave a positive assessment of the paper. The primary concerns involved the assumptions used, specifically the restrictiveness of the monotonicity assumption and the Gaussian design assumption for GSTOR. Furthermore, 2RuZ questioned the optimality of the results and inquired about the algorithm's sensitivity to misspecification of the sparsity parameter
- __Reviewer Fk6c__: This reviewer expressed an overall negative assessment of the paper. The primary concerns centered on the paper’s technical novelty, level of challenge, and clarity. Reviewer Fk6c argued that the STOR algorithm is impractical due to its large sample complexity. In addition, the reviewer found the experimental evaluation insufficient, pointing out that in the sparse high-dimensional setting, the experiments were limited to linear reward functions.
- **Reviewer vpkp**: This reviewer initially gave a lower score but raised the score during the rebuttal period. Their initial concerns centered on the restrictive assumptions, specifically that context vectors must be i.i.d. from a fixed distribution and the requirement that $\|\theta_{\star}\|=1$. Reviewer vpkp also pointed out that the $K^{3/2}$ dependence on the number of arms of the regret bound of ESTOR in the worst case  is a limitation.
- **Reviewer pAj9**: This reviewer gave a positive assessment. Their first concern was that the presentation could be improved for better clarity and intuition. Secondly, they questioned the optimality of the provided regret bounds . pAj9 also suggested that the authors should conduct a more complete empirical comparison.

**Reviewer Concerns:**

- **Reviewer 2RuZ**: The authors acknowledged that while the monotonicity and Gaussian design assumptions are somewhat restrictive, they are natural and standard in literature.  Furthermore, the authors clarified that although the regret bounds may not be minimax optimal with respect to dimension $d$ and number of arms $K$, the algorithm achieves the optimal rate of order $\sqrt{T}$ in terms of $T$. They also clarified that the sparsity analysis does not rely on prior knowledge of the sparsity level $s$.  In summary, it appears that the majority of this reviewer's concerns have been effectively resolved.
- __Reviewer Fk6c__:  The authors provided comprehensive responses addressing concerns regarding the paper's breadth, technical novelty, potential hidden assumptions, exploration length, and high-dimensional nonlinear experiments. They included a detailed clarification of the paper's technical novelty in appendix. Regarding the exploration length, the authors justified the use of a larger sample size for the exploration phase. They argued that because contextual bandits are considerably more challenging than classical Multi-Armed Bandits (MAB), this exploration budget is theoretically necessary and is consistent with prior literature [1] [2].
- **Reviewer vpkp**: During the rebuttal period, the authors addressed Reviewer vpkp's concerns. Consequently, the reviewer raised their score from $4$ to $6$.
- **Reviewer pAj9**: Following the authors' response, Reviewer pAj9 confirmed they had no further concerns and maintained their positive score of $6$.



__References__

[1] A simple unified framework for high dimensional bandit problems. ICML, 2022.

[2] High-dimensional sparse linear bandits. NeurIPS, 2020.

**Reviewer Scores:**

Reviewers 2RuZ and Fk6c did not participate in the post-rebuttal discussion phase.



**Reviewer 2RuZ**:  This reviewer initially provided a positive assessment (Score: 6). Upon reviewing the authors' rebuttal, which provided detailed clarifications on the monotonicity assumption and the optimality of the regret bounds, I believe the reviewer's concerns have been effectively addressed. Therefore, the initial positive score remains justified.



__Reviewer Fk6c__: This reviewer initially leaned towards a negative assessment (Score: 4). However, after reviewing the authors' response and the rebuttal summary, I find that the majority of the reviewer's concerns have been resolved. Specifically, the authors provided strong justifications for the technical novelty (clarified in the new Appendix C) and the exploration batch size, demonstrating that the latter is theoretically optimal for this specific setting. The author also provided an additional experiment to validate their methods.

---

### Decision · Program_Chairs · 2026-01-26

Accept (Poster)